# BEHAVIOR LEARNING (BL)

**Zhenyao Ma** [*]
Xiamen University

**Yue Liang**
University of Tübingen

**Dongxu Li**
Xi'an Jiaotong University

## ABSTRACT

Inspired by behavioral science, we propose *Behavior Learning* (BL), a novel general-purpose machine learning framework that learns interpretable and identifiable optimization structures from data, ranging from single optimization problems to hierarchical compositions. It unifies predictive performance, intrinsic interpretability, and identifiability, with broad applicability to scientific domains involving optimization. BL parameterizes a compositional utility function built from intrinsically interpretable modular blocks, which induces a data distribution for prediction and generation. Each block represents and can be written in symbolic form as a utility maximization problem (UMP), a foundational paradigm in behavioral science and a universal framework of optimization. BL supports architectures ranging from a single UMP to hierarchical compositions, the latter modeling hierarchical optimization structures that offer both expressiveness and structural transparency. Its smooth and monotone variant (IBL) guarantees identifiability under mild conditions. Theoretically, we establish the universal approximation property of both BL and IBL, and analyze the M-estimation properties of IBL. Empirically, BL demonstrates strong predictive performance, intrinsic interpretability and scalability to high-dimensional data. Code: `github.com/MoonYLiang/Behavior-Learning`; installable via `pip install blnetwork`.

## 1 INTRODUCTION

Scientific research often grapples with phenomena that resist precise formalization (Anderson, 1972; Mitchell, 2009), including human and social domains (Simon, 1955; Arthur, 2009). Such phenomena are difficult to predict and even harder to falsify through theory alone. Interpretable machine learning (Interpretable ML) (Molnar, 2020), with its powerful approximation capabilities and built-in transparency, offers a promising alternative for modeling such phenomena. Yet a long-standing tension remains unresolved: model predictive performance and intrinsic interpretability often trade off—a challenge commonly known as the *performance–interpretability trade-off* (Arrieta et al., 2020). High-performing models such as deep neural networks (LeCun et al., 2015) typically lack transparency, while intrinsically interpretable models struggle to capture complex nonlinear patterns.

Some efforts have been made to mitigate the performance–interpretability trade-off, and the main existing approaches can be broadly grouped into four categories. (i) Additive models (Caruana et al., 2015; Hastie, 2017; Nori et al., 2019; Agarwal et al., 2021; Chang et al., 2021). (ii) Concept-based models (Alvarez Melis & Jaakkola, 2018; Kim et al., 2018; Koh et al., 2020). (iii) Rule- and score-based systems (Ustun & Rudin, 2016; Angelino et al., 2018) . (iv) Shape-constrained neural networks (You et al., 2017) . Recent additional interpretable modeling frameworks include Kraus et al. (2024); Liu et al. (2024b); Plonsky et al. (2025). These approaches demonstrate varied strengths.

However, two fundamental limitations remain, restricting their scientific applicability. (i) *Insufficient alignment with scientific theories*. Most approaches focus on extending existing machine learning methods to achieve interpretability, rather than developing a scientifically grounded framework (e.g., based on optimization problems or differential equations). This often hinders alignment with scientific theories and limits the ability to extract scientific knowledge from learned models (Roscher et al., 2020; Bereska & Gavves, 2024; Longo et al., 2024). (ii) *Non-uniqueness of interpretations*.

---

[*]Correspondence to: zhenyaoma@stu.xmu.edu.cn

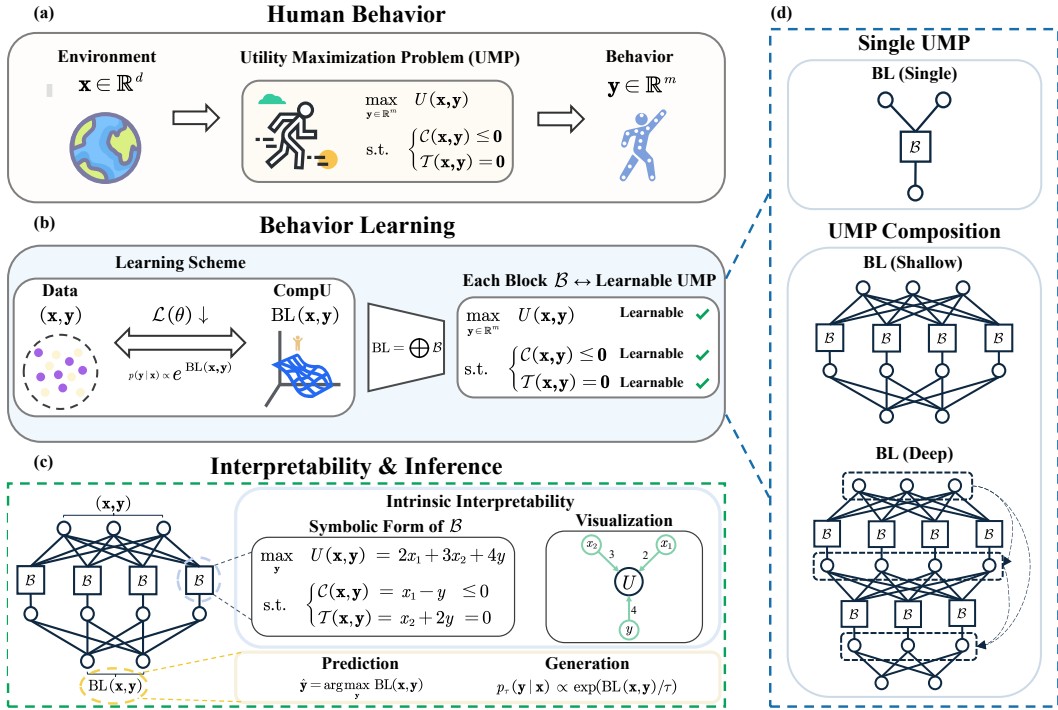

Figure 1: **Behavior Learning (BL).** (a) Human behavior modeled as a UMP. (b) Learning scheme of BL, where CompU denotes the compositional utility function. (c) BL offers intrinsic interpretability (via symbolic form as an optimization problem), identifiability (via unique parameterization), and inference capability. (d) Three architectural variants of BL, from single UMP to deep compositions.

Most models are *non-identifiable*—their interpretations are not uniquely determined by observable predictions in a mathematical sense (Ran & Hu, 2017; Méloux et al., 2025). As a result, such models cannot support reliable estimation of ground-truth parameters (Newey & McFadden, 1994; Van der Vaart, 2000), and may even lack Popperian falsifiability (Popper, 2005), ultimately limiting their scientific credibility. These limitations naturally raise a key question: can we design an interpretable ML framework that mitigates the performance–interpretability trade-off while being scientifically grounded and identifiable?

Inspired by behavioral science, we propose **Behavior Learning (BL)**: *a general-purpose machine learning framework that learns interpretable and identifiable (hierarchical) optimization structures from data*. It unifies high predictive performance, intrinsic interpretability, and identifiability, with broad applicability to scientific domains involving optimization. As illustrated in Figure 1, BL builds on one of the most fundamental paradigms in behavioral science—utility maximization—which posits that human behavior arises from solving a *utility maximization problem* (UMP) (Samuelson, 1948; Debreu, 1959; Mas-Colell et al., 1995). Motivated by this paradigm, BL learns interpretable optimization structures from data. It models responses ($y$) as drawn from a probability distribution induced by a UMP or a composition of multiple interacting UMPs. This distribution is parameterized by a compositional utility function $\mathrm{BL}(\mathbf{x}, \mathbf{y})$, constructed from intrinsically interpretable modular blocks $\mathcal{B}(\mathbf{x}, \mathbf{y})$. Each block is a learnable penalty-based formulation that represents an **optimization problem** (UMP), which can be written in symbolic form and offers transparency comparable to linear regression.

BL admits *hierarchical structure*, mainly in three architectural variants: BL(Single), defined by a single block; BL(Shallow), a moderately layered composition of blocks; and BL(Deep), a deep hierarchical composition of multiple blocks. The latter two model, and can be symbolically interpreted as, **hierarchical optimization structures**. All variants are trained end-to-end to induce a conditional Gibbs distribution for prediction and generation. By refining the penalty functions in each block into smooth and monotone forms, we develop *Identifiable BL* (IBL), the identifiable variant of BL. Under mild conditions, IBL guarantees unique intrinsic interpretability. This property ensures

the scientific credibility of its explanations and further supports recovery of the ground-truth model under appropriate conditions.

While motivated by behavioral science, *BL is not domain-specific*. It applies broadly to any scientific domain where observed outcomes arise as solutions to optimization problems—such as macroeconomics (Ramsey, 1928; Ljungqvist & Sargent, 2018), statistical physics (Gibbs, 1902; Landau & Lifshitz, 2013), or evolutionary biology (Wright et al., 1932; Fisher, 1999). This generality is supported by a key theoretical insight (Theorem 2.2): any optimization problem can be equivalently written as a UMP. This makes BL a general-purpose modeling framework for data-driven inverse optimization (Ahuja & Orlin, 2001) across diverse scientific disciplines.

We study BL both theoretically and empirically. Theoretically, we show that both BL and IBL admit universal approximation under mild assumptions (Section 2.2). For IBL, we further establish its M-estimation properties (Section 2.3), including identifiability, consistency, and universal consistency. Empirically, we evaluate BL across three tasks. Standard prediction tasks (Section 3.1) demonstrate its strong predictive performance. A qualitative case study (Section 3.2) illustrates its intrinsic interpretability. Prediction on high-dimensional inputs (Section 3.3) further demonstrates its scalability to high-dimensional data. Due to limited space, further discussion and related work are deferred to Appendix B and Appendix C, respectively. We also provide guidance on how to scientifically explain BL(Deep) and architectural details in Appendix A and Appendix D, respectively.

Overall, our key contributions are threefold. (i) We propose Behavior Learning (BL), a novel general-purpose machine learning framework inspired by behavioral science, which unifies high predictive performance, intrinsic interpretability, identifiability, and scalability. (ii) For scientific research, BL offers a scientifically grounded, interpretable, and identifiable machine learning approach for modeling complex phenomena that defy precise formalization. BL applies broadly to scientific disciplines involving optimization. (iii) At the paradigm level, BL learns from data the optimization structure of either a single optimization problem or a hierarchical composition of problems through distributional modeling, contributing a new methodology to data-driven inverse optimization.

## 2 BEHAVIOR LEARNING (BL)

### 2.1 UTILITY MAXIMIZATION PROBLEM (UMP)

The modeling of human behavior, particularly in behavioral science and decision theory, often begins with the assumption that observed outcomes arise from a latent optimization process. A canonical formulation of this idea is the Utility Maximization Problem (UMP) (Mas-Colell et al., 1995), in which an agent selects actions $\mathbf{y} \in \mathcal{Y}$ in response to contextual features $\mathbf{x} \in \mathcal{X}$ by solving:

$$\max_{\mathbf{y}\in\mathcal{Y}} U(\mathbf{x},\mathbf{y}) \quad \text{s.t.} \quad \mathcal{C}(\mathbf{x},\mathbf{y}) \leq 0, \ \ \mathcal{T}(\mathbf{x},\mathbf{y}) = 0 \tag{1}$$

Here, $U(\cdot)$ denotes a subjective utility function encoding the agent's internal preferences or goals. The inequality constraint $\mathcal{C}(\cdot)$ captures resource constraints, while the equality constraint $\mathcal{T}(\cdot)$ encodes either endogenous belief consistency or exogenous conservation laws.

The UMP can be recast as a cost–benefit framework, where the agent trades off utility gains against constraint violations. Formally, under mild regularity conditions, it admits an unconstrained penalty reformulation at the level of local optimality (Han & Mangasarian, 1979), as formalized below.

**Theorem 2.1** (Local Exact Penalty Reformulation for UMP). *Let $\mathcal{X} \subset \mathbb{R}^{d_x}$ and $\mathcal{Y} \subset \mathbb{R}^{d_y}$ be nonempty compact sets, and let $U : \mathcal{X} \times \mathcal{Y} \to \mathbb{R}$, $\mathcal{C} : \mathcal{X} \times \mathcal{Y} \to \mathbb{R}^m$, and $\mathcal{T} : \mathcal{X} \times \mathcal{Y} \to \mathbb{R}^p$ be $C^1$. Assume that for a given $\mathbf{x} \in \mathcal{X}$, the Han–Mangasarian constraint qualification holds at any strict local maximizer $\mathbf{y}^\star$ of the UMP. Then there exist $\lambda_0 > 0$, $\lambda_1 \in \mathbb{R}^m_{++}$, and $\lambda_2 \in \mathbb{R}^p_{++}$ such that $\mathbf{y}^\star$ is a local maximizer of*

$$\max_{\mathbf{y}\in\mathcal{Y}} \lambda_0\, \phi\big(U(\mathbf{x},\mathbf{y})\big) - \lambda_1^\top \rho\big(\mathcal{C}(\mathbf{x},\mathbf{y})\big) - \lambda_2^\top \psi\big(\mathcal{T}(\mathbf{x},\mathbf{y})\big). \tag{2}$$

*Here $\phi : \mathbb{R} \to \mathbb{R}$ is strictly increasing and $C^1$, $\rho(z) := \max\{z, 0\}$, and $\psi(z) := |z|$.*

The proof is provided in Appendix E.1. This unconstrained reformulation offers greater tractability for both theoretical analysis and model training.

While motivated by behavioral modeling, the UMP formulation is not domain-specific. It applies to any setting where observed outcomes are solutions to optimization problems. This is because any optimization problem can be equivalently formulated as a UMP. We state this in the following result, while the formal statement and proof are provided in Appendix E.1.

**Theorem 2.2** (Universality of UMP). *Any optimization problem of the form* $\max_{\mathbf{y} \in \mathcal{Y}} f(\mathbf{x}, \mathbf{y})$ *or* $\min_{\mathbf{y} \in \mathcal{Y}} f(\mathbf{x}, \mathbf{y})$, *subject to equality and inequality constraints, is equivalent to a UMP.*

## 2.2 BL ARCHITECTURE

Figure 1(b–d) illustrates the architecture of BL. We consider samples $(\mathbf{x}, \mathbf{y}) \sim \mathcal{D}$, where $\mathbf{x} \in \mathbb{R}^d$ denotes contextual features and $\mathbf{y}$ is the response, represented as $(\mathbf{y}^{\mathrm{disc}}, \mathbf{y}^{\mathrm{cont}}) \in \mathcal{Y}_{\mathrm{disc}} \times \mathbb{R}^{m_c}$, capturing its hybrid structure. Responses are assumed to be stochastically generated by solving multiple interacting UMPs, each with a penalty-based formulation, which together compose a compositional utility function $\mathrm{BL}(\mathbf{x}, \mathbf{y})$. On this basis, we model the data using a conditional Gibbs distribution (Gibbs, 1902) parameterized by $\mathrm{BL}_\Theta(\mathbf{x}, \mathbf{y})$:

$$p_\tau(\mathbf{y} \mid \mathbf{x}; \Theta) = \frac{\exp\big(\mathrm{BL}_\Theta(\mathbf{x}, \mathbf{y})/\tau\big)}{Z_\tau(\mathbf{x}; \Theta)}, \quad Z_\tau(\mathbf{x}; \Theta) = \int_{\mathcal{Y}} \exp\big(\mathrm{BL}_\Theta(\mathbf{x}, \mathbf{y}')/\tau\big) \, d\mathbf{y}' \quad (3)$$

Here the temperature parameter $\tau > 0$ controls the randomness of the response. As $\tau \to 0$, the distribution in equation 3 converges to a Dirac measure supported on $\arg\max_{\mathbf{y}} \mathrm{BL}(\mathbf{x}, \mathbf{y})$, thereby recovering the deterministic best response obtained by solving the composed UMPs.

**Model Structure of** $\mathrm{BL}(\mathbf{x}, \mathbf{y})$. To represent the composition of multiple UMPs, we build $\mathrm{BL}(\mathbf{x}, \mathbf{y})$ by composing fundamental modular blocks $\mathcal{B}(\mathbf{x}, \mathbf{y})$. Each block provides a penalty-based formulation of a single UMP, and together they yield the overall compositional utility function. Motivated by Theorem 2.1, we parameterize $\mathcal{B}(\mathbf{x}, \mathbf{y})$ as

$$\mathcal{B}(\mathbf{x}, \mathbf{y}; \theta) := \lambda_0^\top \phi\big(U_{\theta_U}(\mathbf{x}, \mathbf{y})\big) - \lambda_1^\top \rho\big(\mathcal{C}_{\theta_C}(\mathbf{x}, \mathbf{y})\big) - \lambda_2^\top \psi\big(\mathcal{T}_{\theta_T}(\mathbf{x}, \mathbf{y})\big) \quad (4)$$

where $\theta := (\lambda_0, \lambda_1, \lambda_2, \theta_U, \theta_C, \theta_T)$ denotes the complete set of learnable parameters. Following Theorem 2.1, $\phi$ is an increasing function; $\rho$ penalizes inequality violations; and $\psi$ captures symmetric deviations. Each block can be written as a well-defined UMP.

We then compose $\mathrm{BL}(\mathbf{x}, \mathbf{y})$ from multiple $\mathcal{B}$-blocks through hierarchical composition to improve its representational power for optimization structures, yielding three main architectural variants, as illustrated in Figure 1(d).

1. BL(Single) applies a single instance of $\mathcal{B}(\mathbf{x}, \mathbf{y})$ as defined in equation 4, without any additional layers. It can be viewed as learning a single UMP, and offers maximal interpretability.

2. BL(Shallow) uses $\mathcal{B}(\mathbf{x}, \mathbf{y})$ as the fundamental modular block to construct a shallow network. It introduces one or two intermediate layers of computation. Each layer $\mathbb{B}_\ell$ stacks multiple parallel $\mathcal{B}_{\ell,i}$ blocks to produce a vector in $\mathbb{R}^{d_\ell}$, i.e., $\mathbb{B}_\ell(\mathbf{x}, \mathbf{y}; \theta_\ell) := [\mathcal{B}_{\ell,1}(\mathbf{x}, \mathbf{y}; \theta_{\ell,1}), \ldots, \mathcal{B}_{\ell,d_\ell}(\mathbf{x}, \mathbf{y}; \theta_{\ell,d_\ell})]^\top$. The output of $\mathbb{B}_\ell$ is directly fed into the next $\mathbb{B}_{\ell+1}$, and only the final output is passed through a learnable affine transformation.

3. BL(Deep) extends the BL(Shallow) architecture to more than two layers, enabling richer hierarchical compositions of UMPs while maintaining the same recursive structure. As before, only the final output is affine transformed.

The overall structure of BL(Shallow) and BL(Deep) can be expressed in a unified form, where the shallow case corresponds to $L \leq 2$ and the deep case to $L > 2$:

$$\mathrm{BL}(\mathbf{x}, \mathbf{y}) := \mathbf{W}_L \cdot \mathbb{B}_L\big(\cdots \mathbb{B}_2(\mathbb{B}_1(\mathbf{x}, \mathbf{y})) \cdots\big) \quad (5)$$

**Learning Objective.** The response $\mathbf{y}$ may contain both discrete and continuous components. For discrete responses, we directly apply cross-entropy (Kullback & Leibler, 1951) on $\mathbf{y}^{\mathrm{disc}}$. For continuous responses, since the compositional utility function is analogous to an energy function (LeCun et al., 2006), we employ denoising score matching (Vincent, 2011) on $\mathbf{y}^{\mathrm{cont}}$. The final objective combines the two with nonnegative weights $\gamma_{\mathrm{d}}, \gamma_{\mathrm{c}}$:

$$\mathcal{L}(\theta) = \gamma_{\mathrm{d}} \, \mathbb{E}\big[-\log p_\tau(\mathbf{y}^{\mathrm{disc}} \mid \mathbf{x})\big] + \gamma_{\mathrm{c}} \, \mathbb{E}\big\|\nabla_{\tilde{\mathbf{y}}^{\mathrm{cont}}} \log p_\tau(\tilde{\mathbf{y}}^{\mathrm{cont}} \mid \mathbf{x}) + \sigma^{-2}(\tilde{\mathbf{y}}^{\mathrm{cont}} - \mathbf{y}^{\mathrm{cont}})\big\|^2 \quad (6)$$

**Implementation Details.** Here, we describe the key implementation choices for the general form of BL, taken as defaults unless otherwise noted. Further details are provided in Appendix D.3.

- Function Instantiation. Following equation 4, we instantiate the function $\mathcal{B}(\mathbf{x}, \mathbf{y})$ as

$$\mathcal{B}(\mathbf{x}, \mathbf{y}) = \lambda_0^\top \tanh\big(\mathbf{p}_u(\mathbf{x}, \mathbf{y})\big) - \lambda_1^\top \mathrm{ReLU}\big(\mathbf{p}_c(\mathbf{x}, \mathbf{y})\big) - \lambda_2^\top \big|\mathbf{p}_t(\mathbf{x}, \mathbf{y})\big| \tag{7}$$

  where $\mathbf{p}_u, \mathbf{p}_c, \mathbf{p}_t$ are polynomial feature maps of bounded degree, providing interpretable representations of utility, inequality, and equality terms, respectively. The bounded $\tanh$ reflects the principle of diminishing marginal utility (Jevons, 2013), a commonly assumed principle in behavioral science, while $\mathrm{ReLU}$ and $|\cdot|$ introduce soft penalties for constraint violations.

- Polynomial Maps. In BL(Single), the structure of polynomial maps is optional. In BL(Shallow) and BL(Deep), each $\mathcal{B}$-block employs affine transformations as its polynomial maps, with higher-degree and interaction terms omitted by default for computational efficiency.

- Skip Connections. For deep variants, skip connections can be optionally introduced to improve representational efficiency.

More **detailed architectural descriptions** for this section are provided in Appendix D.

**Theoretical Guarantees.** Under the given architecture, the BL framework has universal approximation power: it can approximate any continuous conditional distribution arbitrarily well, provided that BL has sufficient capacity, as stated below. The proof is given in Appendix E.2.

**Theorem 2.3** (Universal Approximation of BL). *Let $\mathcal{X} \subset \mathbb{R}^d$ and $\mathcal{Y} \subset \mathbb{R}^m$ be compact sets, and let $p^\star(\mathbf{y} \mid \mathbf{x})$ be any continuous conditional density such that $p^\star(\mathbf{y} \mid \mathbf{x}) > 0$ for all $(\mathbf{x}, \mathbf{y}) \in \mathcal{X} \times \mathcal{Y}$. Then for any $\tau > 0$ and $\varepsilon > 0$, there exists a finite BL architecture (with depth and width depending on $\varepsilon$) and a parameter $\theta^\star$ such that the Gibbs distribution in equation 3 satisfies*

$$\sup_{\mathbf{x} \in \mathcal{X}} \mathrm{KL}\big(p^\star(\cdot \mid \mathbf{x}) \,\|\, p_\tau(\cdot \mid \mathbf{x}; \theta^\star)\big) < \varepsilon. \tag{8}$$

**Interpretability.** Alongside its expressive power, BL also exhibits strong intrinsic interpretability. **(i)** Each $\mathcal{B}$-block can be expressed in **symbolic form** as an optimization problem (UMP): the $\tanh$ term defines the objective, the $\mathrm{ReLU}$ term corresponds to an inequality constraint, and the absolute-value term corresponds to an equality constraint. Thus, BL(Single) can be directly expressed as a symbolic UMP, whereas deeper architectures can be interpreted as compositions of UMPs, with each block retaining interpretability. **(ii)** The **polynomial basis** ensures a level of **transparency comparable to linear regression**, as both objectives and constraints can be represented as linear combinations of polynomial features. It can further be visualized as a computational graph (Figure 6), in which each input's influence on every $\mathcal{B}$-block is traceable through compositional pathways. **(iii)** BL(Deep) composes $\mathcal{B}$-blocks in a layered manner, forming a **hierarchical optimization structure**. Interpretation proceeds in a bottom-up fashion, where the relation between any two consecutive layers can be viewed as aggregation or coarse-grained observation. Overall, the interpretive pathway is: *raw input features → micro-level optimization blocks → macro-level aggregation or coarse-grained behavioral constructs → macro-level optimization systems*. Appendix A provides a detailed description of this interpretation procedure. **(iv)** BL also offers multiple architectural degrees of freedom that provide flexibility but simultaneously affect the resulting interpretability. In deep variants, skip connections introduce cross-layer dependency structures that are modeled in statistical physics (Yang & Schoenholz, 2017). Replacing polynomial maps with affine transformations preserves the underlying optimization semantics but reduces symbolic granularity, yielding a more qualitative rather than symbolic interpretation of each block. **(v)** BL can be interpreted as a single UMP when the final layer contains only one $\mathcal{B}$-block, since all lower-layer structures aggregate into a unified optimization problem. When the final layer contains multiple $\mathcal{B}$-blocks, BL corresponds to a linear trade-off among multiple optimization problems.

## 2.3 IDENTIFIABLE BEHAVIOR LEARNING (IBL)

Beyond prediction and interpretability, the BL framework supports a third fundamental goal: *the identification of ground-truth parameters*, which in turn endows BL with the capacity for scientifically credible modeling. We refer to this setting as **Identifiable Behavior Learning (IBL)**. In the

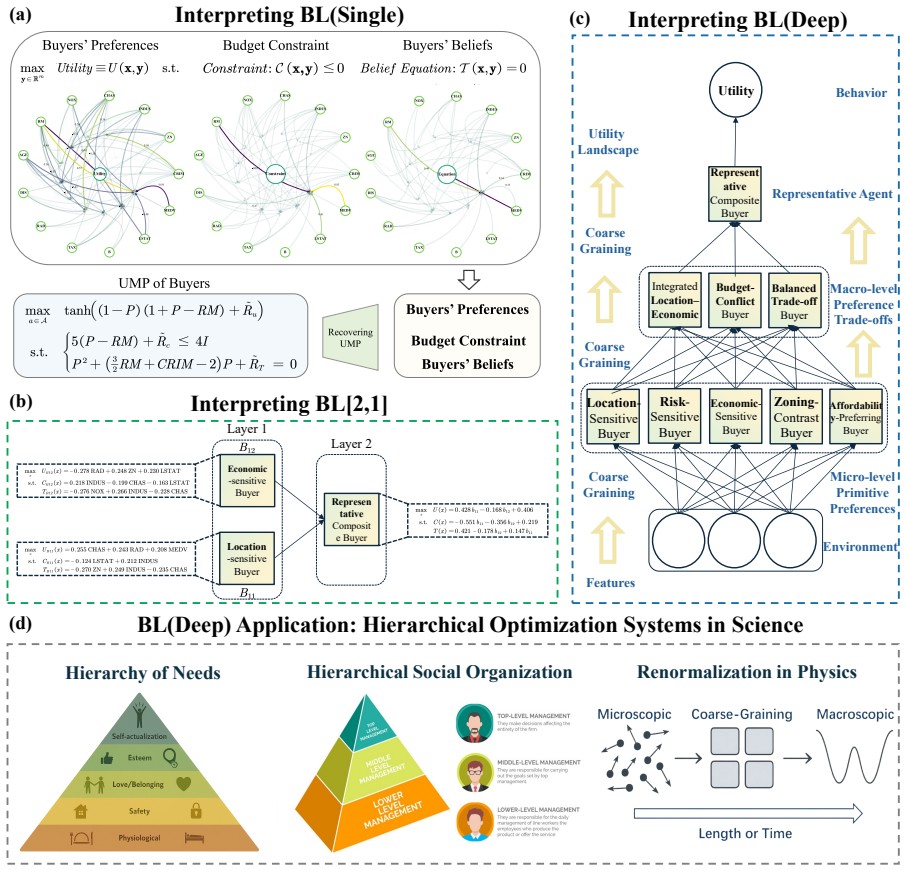

Figure 2: **(a)** Visualization and symbolic form of BL(Single) trained on the *Boston Housing* dataset, modeling the UMP ($\max U$   s.t. $\mathcal{C} \leq 0,\ \mathcal{T} = 0$) of a representative buyer in Boston housing (details in Section 3.2). Top: computational graphs of the polynomials inside the three penalty functions—$\tanh$ (preference), $\mathrm{ReLU}$ (budget), and $|\cdot|$ (belief). Each graph is respectively centered on $\tanh^{-1}(U)$, $\mathcal{C}$, and $\mathcal{T}$ from left to right, with surrounding nodes representing input features. Directed edges (shown only if coefficient $\geq 0.3$) indicate how each feature contributes to the corresponding term. Bottom: approximate symbolic formulation of the trained BL model as a UMP. **(b)** The BL[2,1] architecture. Layer 1 identifies two key micro-level preference types: the *Economic-sensitive Buyer* and the *Location-sensitive Buyer*. Layer 2 aggregates these two components into an effective representative buyer. **(c)** The BL(Deep) [5,3,1] architecture. Layer 1 recovers five distinct micro-level housing preference types. Layer 2 identifies three macro-level trade-off types capturing different ways these primitive preferences interact. Layer 3 aggregates them into the overall representative buyer. Table 8 provides detailed descriptions of each type. BL(Deep) provides a hierarchical explanation consistent with the coarse-graining principle (Kadanoff, 1966) in statistical physics, reconstructing the full micro-to-macro optimization hierarchy. In addition, the preference and trade-off patterns uncovered by BL(Deep) are well documented in the classical economics literature (see Table 9). **(d)** BL can be applied to a broad class of hierarchical optimization structures in science, including hierarchical need structures, hierarchical social–organizational structures, and renormalization-style coarse-grained structures in physics.

IBL setting, we define the modular block as

$$\mathcal{B}^{\mathrm{id}}(\mathbf{x}, \mathbf{y}; \theta) := \lambda_0^\top \phi^{\mathrm{id}}\big(U_{\theta_U}(\mathbf{x}, \mathbf{y})\big) - \lambda_1^\top \rho^{\mathrm{id}}\big(\mathcal{C}_{\theta_C}(\mathbf{x}, \mathbf{y})\big) - \lambda_2^\top \psi^{\mathrm{id}}\big(\mathcal{T}_{\theta_T}(\mathbf{x}, \mathbf{y})\big) \qquad (9)$$

Unlike BL, which uses general nonlinearities, the IBL architecture imposes stricter structural constraints: $\phi^{\mathrm{id}}$ and $\rho^{\mathrm{id}}$ are strictly increasing, while $\psi^{\mathrm{id}}$ is symmetric and strictly increasing in $|\cdot|$. In addition, all three functions are $C^1$. These properties ensure that each UMP block stays responsive

and adjusts smoothly to objectives and constraints. In practice, we instantiate equation 9 as

$$\mathcal{B}^{\mathrm{id}}(\mathbf{x}, \mathbf{y}) = \lambda_0^\top \tanh\big(\mathbf{p}_u(\mathbf{x}, \mathbf{y})\big) - \lambda_1^\top \operatorname{softplus}\big(\mathbf{p}_c(\mathbf{x}, \mathbf{y})\big) - \lambda_2^\top \big(\mathbf{p}_t(\mathbf{x}, \mathbf{y})\big)^{\odot 2} \tag{10}$$

where $(\cdot)^{\odot 2}$ denotes elementwise square.

We design IBL in three architectural forms. Similar to BL, the *IBL(Single)* directly uses $\mathcal{B}^{\mathrm{id}}(\mathbf{x}, \mathbf{y})$ as the compositional utility function. The *IBL(Shallow)* and *IBL(Deep)* variants are defined recursively as

$$\mathrm{IBL}(\mathbf{x}, \mathbf{y}) := \mathbf{W}_L^\circ \cdot \mathbb{B}_L^{\mathrm{id}}\big(\cdots \mathbb{B}_2^{\mathrm{id}}\big(\mathbb{B}_1^{\mathrm{id}}(\mathbf{x}, \mathbf{y})\big) \cdots\big), \quad L \geq 1 \tag{11}$$

where $\mathbb{B}_\ell^{\mathrm{id}}$ stacks multiple parallel blocks $\mathcal{B}_{\ell,i}^{\mathrm{id}}(\mathbf{x}, \mathbf{y})$, and $\mathbf{W}_L^\circ$ is a learnable affine transformation without bias. All other design choices follow the BL setting.

**Theoretical Foundation.** IBL admits favorable properties for ground-truth identification. We begin by establishing identifiability, which is fundamental for statistical inference. We first state our key assumption (see Assumption E.1 for details).

**Assumption 2.1.** *Let $\bar{\Psi}$ denote the quotient space of atomic parameters. We assume that the map $\bar{\Psi} \to \mathbb{R}^{\mathcal{X} \times \mathcal{Y}}, \bar{\psi} \mapsto g_{\bar{\psi}}$, is **injective**, and that any finite set of distinct atoms is **linearly independent**. We further restrict attention to **minimal representations** with no duplicate atoms and a fixed canonical ordering.*

**Theorem 2.4** (Identifiability of IBL)**.** *Under Assumption E.1, the architectures IBL(Single), IBL(Shallow), and IBL(Deep) are identifiable in the parameter quotient space $\bar{\Theta}$.*

**Theorem 2.5** (Loss Identifiability of IBL)**.** *The IBL model is parameterized by $\theta \in \Theta$. Suppose $\Theta$ is compact. Then under Assumption E.1, the population loss $\mathcal{L}$ defined in equation 6 satisfies:*

- *If $\gamma_c > 0$, it admits a **unique** minimizer in the quotient space $\bar{\Theta}$;*

- *If $\gamma_c = 0$, it admits a **unique** minimizer in the scale-invariant quotient space $\widetilde{\Theta}$.*

Theorems 2.4 and 2.5 together establish the identifiability of IBL. Theorem 2.4 shows that if two IBL models of the same structure induce the same compositional utility, then their parameters coincide up to an equivalence class. Theorem 2.5 further extends this result to loss-based identifiability. These results jointly imply that IBL admits a unique parameter estimate up to an equivalence class, and thus yields intrinsic interpretability that is unique up to the same class.

Building on identifiability, Theorem 2.6 establishes the statistical consistency of IBL: under compactness of the parameter space, the learned parameters converge in probability to a minimizer of the population loss as the sample size $n \to \infty$. If the model is correctly specified, the estimator further *converges to the ground-truth parameter, recovering the true underlying model*, thereby endowing IBL with the potential to recover the ground-truth model.

**Theorem 2.6** (Consistency of IBL)**.** *Let $\Xi$ denote the relevant parameter quotient space: $\Xi = \bar{\Theta}$ if $\gamma_c > 0$, and $\Xi = \widetilde{\Theta}$ if $\gamma_c = 0$. Let $\hat{\theta}_n \in \arg\min_{\theta \in \Theta} \mathcal{M}_n(\theta)$ denote the empirical minimizer, and let $\theta^\bullet \in \arg\min_{\theta \in \Theta} \mathcal{M}(\theta)$ denote the population minimizer. Then under the conditions of Theorem E.5,*

$$\hat{\theta}_n \xrightarrow{p} \theta^\bullet \quad in \ \Xi, \qquad \mathcal{M}(\hat{\theta}_n) \xrightarrow{p} \mathcal{M}(\theta^\bullet).$$

*Moreover, if the model is correctly specified (i.e., the data distribution is realized by some $\theta^\star \in \Theta$), then $\theta^\bullet = \theta^\star$ in $\Xi$, and thus $\hat{\theta}_n \xrightarrow{p} \theta^\star$.*

Correct specification is a strong and often unrealistic assumption. Fortunately, the IBL framework—like BL—also enjoys a **universal approximation guarantee** (Theorem E.6). Building on this result, we further establish the **universal consistency** of IBL: *even under misspecification, IBL is capable of recovering the ground-truth model* with sufficiently large sample sizes.

**Theorem 2.7** (Universal Consistency of IBL)**.** *Under the conditions of Theorem E.7, for any admissible data-generating distribution $p^\dagger$ satisfying the regularity assumptions of Theorem E.6, the IBL posterior sequence $\{p_{\hat{\theta}_n}\}$ satisfies*

$$\sup_{x \in \mathcal{X}} \mathrm{KL}\big(p^\dagger(\cdot \mid \mathbf{x}) \,\|\, p_{\hat{\theta}_n}(\cdot \mid \mathbf{x})\big) \xrightarrow{p} 0,$$

*i.e., the learned conditional distributions $\{p_{\hat{\theta}_n}\}$ converge in KL to $p^\dagger$ uniformly over $\mathbf{x}$.*

Specifically, this result implies that, even under model misspecification, the learned predictive distribution $p_{\hat{\theta}_n}$, parameterized by the IBL model, converges uniformly in KL to the true conditional distribution $p^\dagger$, provided that the capacity of the IBL architecture grows with the sample size $n$.

Formal statements and proofs of all theorems in this part are deferred to Appendix E.3.

## 3 EXPERIMENTS

In this section, we conduct four groups of experiments to systematically evaluate the capabilities of BL. Due to space constraints, details are provided in Appendix F.

### 3.1 STANDARD PREDICTION TASKS

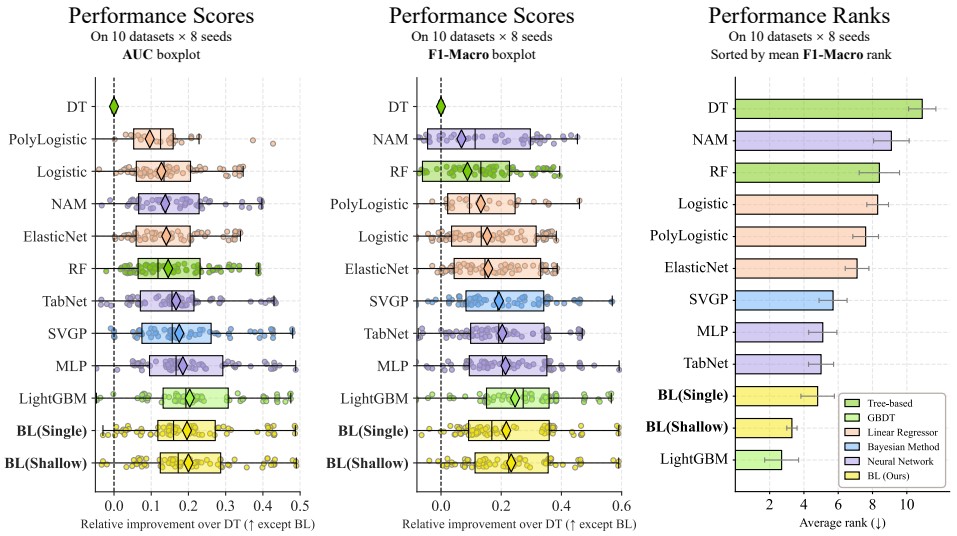

Figure 3: Predictive performance of BL and baselines. Left/Middle: relative AUC and F1-Macro gains over DT, sorted by mean (excluding BL). Right: mean F1-Macro ranks (↓ better). BL achieves first-tier performance in both metrics. Its variants rank second and third in mean F1-Macro rank, with BL(Shallow) showing no statistically significant difference from state-of-the-art models.

*Is BL accurate enough for standard prediction tasks?* In this part, we evaluate the predictive performance of BL on *10 datasets* (Table 2), covering diverse sample sizes, feature dimensions, and scientific domains. For fair comparison, we consider two BL variants—BL(Single) and BL(Shallow)—and compare them against *10 baseline models* (Table 3) drawn from five methodological families: neural networks, tree-based models, gradient boosting methods, Bayesian methods, and linear regressors. All methods share a unified preprocessing and tuning pipeline.

**Predictive Performance.** Figure 3 shows that BL attains first-tier predictive performance overall, achieving the best results among intrinsically interpretable models. Notably, BL(Shallow) surpasses MLP, highlighting that BL delivers interpretability without sacrificing performance.

### 3.2 INTERPRETING BL: A CASE STUDY

*How can BL be interpreted in practice?* This part presents a case study using the *Boston Housing* dataset, where we train a supervised BL(Single) model with a degree-2 polynomial basis, a BL[2,1] model (i.e., a two-layer BL with two B-blocks in the first layer and one in the second layer), and a BL(Deep) model with a [5,3,1] architecture to predict median home values. We illustrate how the internal structure of BL can be interpreted as explicit optimization problems and their hierarchical versions, accompanied by complementary visualizations. Further details are provided in Appendix F.3 and F.6.

**Symbolic Form of BL(Single) as a UMP.**  As shown in Figure 2, the trained BL(Single) model can be interpreted as the UMP of a *representative buyer* in the Boston Housing market, comprising a single objective, inequality, and equality term. Each term is represented by an estimated quadratic polynomial. For parsimony, we extract approximate symbolic expressions by retaining only the monomials with the largest (2–5) absolute coefficients, while collecting the remaining terms (including constants) into a residual term $\tilde{R}$. For example, the utility term can be written as:

$$\mathbf{p}_u = -0.56 \cdot P^2 - 0.6 \cdot \mathrm{RM} + 0.57 \cdot \mathrm{RM} \cdot P + \tilde{R}_u \approx (1 - P)(1 + P - \mathrm{RM}) + \tilde{R}_u$$

We similarly simplify the budget and belief terms to recover an approximate UMP for the buyer. The full symbolic form is illustrated at the bottom of Figure 2.

**Interpreting BL(Single) via Model Visualization.**  Visualizations of each term's polynomial reveal how features constitute the UMP. Three insights emerge from the visualizations in Figure 2. (i) *Median housing price (MEDV)* and *average number of rooms (RM)* are dominant across all terms—MEDV negatively affects utility in a near-quadratic form, while RM modulates its marginal effect. (ii) *Proportion of lower-income residents (LSTAT)* features prominently in the budget constraint, reflecting implicit resource limitations. (iii) *Crime rate (CRIM)* appears only in the belief term, suggesting that buyers treat it as influencing others' behavior rather than their own preferences.

**Interpreting BL(Deep).**  (1) Figure 2 (b) illustrates the optimization problems learned by the BL[2,1] model. Layer 1 identifies two micro-level preference types: an *Economic-sensitive Buyer*, whose utility and constraint terms load primarily on ZN (Large-lot residential share) and LSTAT (Proportion of lower-income residents); and a *Location-sensitive Buyer*, driven mainly by CHAS (Charles River indicator) and RAD (Highway accessibility). Layer 2 aggregates these basic preferences, yielding an effective "representative buyer" that integrates the two preference types. (2) Figure 2 (c) presents the internal structure of the BL[5,3,1] model. In Layer 1, BL recovers five distinct *micro-level preference* types characterizing heterogeneous patterns in the housing market. Layer 2 identifies three macro-level representative agents, each capturing a different *macro-level trade-off* among the basic preferences. Layer 3 then aggregates these components into a single high-level mechanism, yielding the overall representative buyer. Table 8 provides detailed descriptions of each type. (3) Beyond interpretability, we find that each preference pattern and trade-off recovered by BL(Deep) aligns with established findings in the economics literature (see Table 9). This indicates that BL successfully reconstructs underlying scientific knowledge.

## 3.3 Prediction on High-Dimensional Inputs

*Is BL scalable to high-dimensional inputs?* We evaluate BL against the *energy-based MLP* (E-MLP) baseline across network depths $d \in \{1, 2, 3\}$, with all models implemented without skip connections. Experiments are conducted on *four datasets* spanning both image and text domains, and are evaluated using *six metrics*: in-distribution accuracy, calibration metrics (ECE and NLL), and OOD robustness metrics (AUROC, AUPR, and FPR@95). For OOD evaluation, we adopt symmetric ID↔OOD splits, using MNIST (LeCun et al., 2002) and Fashion-MNIST (Xiao et al., 2017) as one pair, and AG News and Yelp Polarity (Zhang et al., 2015) as another. E-MLP and BL are controlled to have comparable parameters.

**Scalability on High-Dimensional Inputs.**  Figure 4 and Table 1 present results for BL and E-MLP across network depths. On image datasets, the two models exhibit comparable in-distribution accuracy, while BL generally achieves stronger out-of-distribution detection performance on Fashion-MNIST at similar accuracy levels. On text datasets, BL consistently improves ID accuracy over E-MLP across depths. However, OOD detection behavior varies by dataset: BL outperforms E-MLP on Yelp, whereas E-MLP shows better OOD discrimination on AG News. BL also achieves better calibration metrics (ECE and NLL; Table 13).

**Downward Shift of the Pareto Frontier.**  Table 11 reports the parameter counts of BL and E-MLP across four tasks, and Tables 12 summarize their runtimes. The two models have highly comparable parameter sizes. Across datasets, BL exhibits slightly higher training time than E-MLP. Combining these results with their comparable predictive performance and the intrinsic interpretability of BL, in contrast with the black-box E-MLP, indicates that BL achieves a *downward shift* of the Pareto frontier.

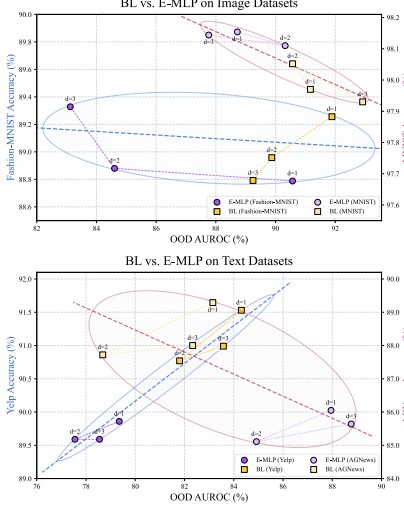

Figure 4: Comparison of BL and E-MLP on image and text datasets; $d$ denotes model depth.

Table 1: ID accuracy and OOD AUROC (%) on image and text datasets. BL and E-MLP are evaluated at depths 1–3 with matched parameter counts, both without skip connections. Top-two per column are blue and red.

| | **Image Datasets** | | | |
| **Model** | MNIST | | Fashion-MNIST | |
| | Accuracy | OOD AUROC | Accuracy | OOD AUROC |
| E-MLP (depth=1) | 98.15 ± 0.07 | 88.72 ± 1.36 | 88.79 ± 0.29 | 90.57 ± 1.39 |
| BL (depth=1) | 97.97 ± 0.18 | 91.17 ± 2.68 | 89.26 ± 0.22 | 91.89 ± 0.71 |
| E-MLP (depth=2) | 98.11 ± 0.08 | 90.32 ± 1.74 | 88.88 ± 0.26 | 84.61 ± 2.56 |
| BL (depth=2) | 98.05 ± 0.12 | 90.57 ± 2.49 | 88.96 ± 0.39 | 89.87 ± 2.48 |
| E-MLP (depth=3) | 98.14 ± 0.11 | 87.76 ± 2.55 | 89.33 ± 0.25 | 83.13 ± 1.90 |
| BL (depth=3) | 97.93 ± 0.27 | 92.92 ± 1.69 | 88.79 ± 0.25 | 89.24 ± 4.18 |
| | **Text Datasets** | | | |
| **Model** | AG News | | Yelp | |
| | Accuracy | OOD AUROC | Accuracy | OOD AUROC |
| E-MLP (depth=1) | 86.05 ± 0.15 | 87.95 ± 2.72 | 89.86 ± 0.09 | 79.35 ± 0.86 |
| BL (depth=1) | 89.29 ± 0.12 | 83.15 ± 3.29 | 91.53 ± 0.02 | 84.31 ± 0.74 |
| E-MLP (depth=2) | 85.11 ± 0.29 | 84.92 ± 5.40 | 89.59 ± 0.22 | 77.55 ± 2.16 |
| BL (depth=2) | 87.72 ± 0.64 | 78.68 ± 5.09 | 90.77 ± 0.19 | 81.80 ± 1.06 |
| E-MLP (depth=3) | 85.64 ± 0.55 | 88.77 ± 1.66 | 89.59 ± 0.53 | 78.55 ± 2.90 |
| BL (depth=3) | 88.00 ± 0.30 | 82.33 ± 3.39 | 90.99 ± 0.17 | 83.58 ± 0.62 |

## 3.4 Constraint Enforcement Test: High-Dimensional Energy Conservation

To evaluate whether the learnable penalty terms in BL enforce near-hard constraints under finite temperature, we isolate the penalty mechanism and test it on a high-dimensional energy-conservation constraint. The utility term is removed, allowing violations to be analyzed as functions of temperature $\tau$ and penalty scale $\lambda$ (details in Appendix F.5).

**Constraint enforcement.** Figure 5 shows that BL achieves near-hard constraint enforcement under finite temperature and penalty scaling. Violations decrease substantially as $\tau$ decreases or $\lambda$ increases. At around $\lambda = 25$ and $\tau = 0.01$, the 64-dimensional energy-conservation constraint is enforced within $10^{-2}$ error. Curves remain mostly smooth and monotone in 64 dimensions, indicating stable Langevin sampling and effective penalty enforcement.

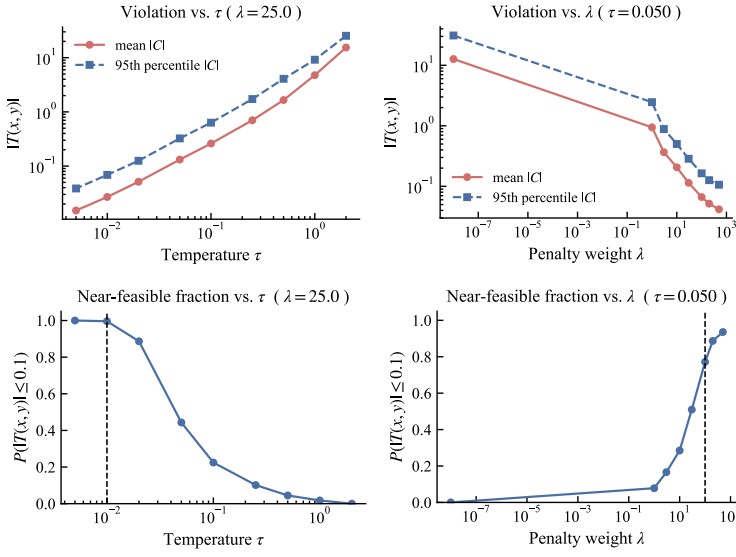

Figure 5: Constraint enforcement test of the BL penalty block on an energy-conservation constraint. The figure reports violation statistics $|T(x,y)|$ when varying the temperature $\tau$ (left side of panel) and the penalty weight $\lambda$ (right side of panel).

## 4 ACKNOWLEDGEMENTS

We would like to thank Prof. Dr. Philipp Hennig, Shu Liu, and Prof. Dr. Sen Geng for their helpful discussions and valuable suggestions. We are also grateful to participants of the Xi'an Jiaotong University seminar for their constructive feedback.

We acknowledge the public computational resources provided by the University of Tübingen.

Finally, we sincerely thank all anonymous reviewers for their insightful comments. In particular, we appreciate Reviewer sGAR for the highly constructive advice. The energy conservation constraint experiment was added following a suggestion from Reviewer sGAR.

If any errors remain, they are solely our responsibility.

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

## A    Scientific Explanation of BL(Deep)

BL(Deep) provides a form of interpretability that is consistent with hierarchical optimization structures. In BL, each layer performs a coarse-graining of the optimization structure implemented by the layer below. An intuitive analogy is a corporate organizational hierarchy: lower-layer managers solve their own local optimization problems, while higher-layer managers aggregate and coordinate the outcomes of many such lower-layer problems to achieve broader organizational objectives. BL(Deep) follows the same principle—higher layers summarize, reorganize, and coordinate the solutions formed at lower layers.

This perspective aligns with many scientific domains characterized by multi-level complexity, including (i) the formation of representative behavioral agents in behavioral sciences, and (ii) renormalization in statistical physics, where fine-scale interactions are compressed into effective coarse-scale potentials.

We describe the explanation procedure below. To build intuition, let us first consider a generic hierarchical optimization structure—this may refer to a multi-layer organizational structure composed of individual agents, or a multi-scale physical system composed of interacting particles.

### Step 1: Bottom-layer interpretation.

Each bottom-layer block is an optimization problem that directly receives inputs from the environment. These blocks correspond to *micro-level behavioral mechanisms*, such as the decision rules of individual agents performing environment-facing tasks in an organization, or the motion laws governing a single particle in statistical physics. Examining these bottom-layer blocks reveals the fundamental optimization principles followed by all units that directly interact with the environment.

### Step 2: Layer-wise coarse-graining and micro-to-macro aggregation.

Blocks in the next layer aggregate the outputs of lower-layer optimization problems through a new optimization step, producing a *coarse-grained behavioral summary*. Each higher-level block represents the effective optimization system that emerges from the interactions among many lower-level units, thereby capturing macro-level regularities distilled from micro-level mechanisms.

This micro-to-macro transition is consistent with many well-established scientific principles, including:

- (i) **Aggregation and coordination**: in hierarchical organizations, the outputs of lower-level agents are aggregated, reallocated, and coordinated by higher-level agents to achieve improved organizational objectives.

- (ii) **Coarse-grained observation**: in hierarchical behavioral systems, individual agents are grouped into categories that share characteristic optimization patterns; in statistical physics, many particles collectively form systems whose coarse-grained behavior is governed by effective potentials induced by microscopic interactions.

### Step 3: Bottom-up reconstruction.

A global explanation is obtained by tracing the hierarchy upward, following the model's micro-to-macro abstraction path: raw input features → micro-level optimization blocks → macro-level aggregation and coordination or coarse-grained behavioral constructs → macro-level optimization system.

At each layer, we inspect the characteristics of each block and its associated optimization objective, as well as how these optimization problems evolve across layers. This reveals how each higher layer aggregates, coordinates, or coarse-grains the outputs of the layer below. Together, these observations yield a compact multi-scale interpretation in which BL is understood as a hierarchical optimization structure.

## B    Discussion

In what follows, we discuss the limitations and future directions of Behavior Learning from the perspectives of theoretical foundations, architecture, and applications.

**Scalability of theoretical assumptions.** The identifiability-related statistical theorems constitute the core theoretical pillars of IBL, ensuring uniqueness of the interpretability and supporting its scientific credibility. Although these results hold under mild conditions, their behavior in large-scale, highly over-parameterized architectures remains less well understood. This highlights the need for systematic investigations into the robustness, potential failure modes, and empirical boundaries of these guarantees when applied to modern large-scale learning systems.

**Choice of basis functions.** Polynomial basis functions enhance expressivity while preserving symbolic interpretability in BL (Single). However, high-order polynomials may introduce optimization instability, exacerbate sensitivity to initialization and normalization, and complicate training dynamics. Future work may explore alternative basis families—such as trigonometric, spline-based, or neural basis functions—and develop conditioning or normalization strategies that improve numerical stability without sacrificing interpretability.

**Interpretable generative modeling.** BL integrates several training techniques from energy-based models while retaining intrinsic interpretability, enabling interpretable generative modeling for vision (e.g., image or video generation) and language (e.g., large language models). Extending BL to explicitly generative architectures in which outputs correspond directly to human-understandable and scientifically meaningful blocks represents a compelling direction. Such extensions could yield generative systems with greater transparency, controllability, and scientific credibility compared to traditional black-box models.

**Hybrid architectures for partial interpretability.** A promising direction for future work is to develop hybrid architectures that integrate BL with black-box models in a principled way to achieve partial interpretability. Three avenues are particularly worth exploring: (i) Feature-level integration. Black-box neural networks can serve as high-capacity feature extractors, while BL operates on the resulting learned representations to impose structured, optimization-based semantics. (ii) Decision-critical integration. BL blocks may be inserted specifically at high-risk or decision-critical components of the model, substantially reducing the interpretability and reliability risks associated with purely black-box architectures. (iii) Mechanism-level integration. Because BL provides an optimization-driven inductive bias aligned with many real-world mechanisms, selectively applying BL to the parts of the system where such inductive bias is essential may yield models that better capture the underlying ground-truth processes while retaining the flexibility of deep networks, thereby improving generalization performance.

**BL for scientific and social-scientific modeling.** BL represents data as a composition of optimization problems, closely resonating with modeling paradigms in the natural and social sciences. Its competitive performance, intrinsic interpretability, and statistical rigor position BL as a promising framework for scientific machine learning. Future research may apply BL to domains such as statistical physics, evolutionary biology, computational neuroscience, and climate dynamics, as well as behavioral science, economics, sociology, and political science—particularly in settings involving complex, partially formalized, or cognitively meaningful structures.

## C  RELATED WORK

### C.1  INTERPRETABILITY

Interpretability has become increasingly vital in machine learning (Lipton, 2018; Molnar, 2020), especially for scientific domains (Doshi-Velez & Kim, 2017; Roscher et al., 2020). Ensuring interpretability fosters transparency and reproducibility, and may further provide insights into underlying scientific principles. The ideal form of interpretability is **intrinsic interpretability**, in which a model's structure or parameters are directly understandable to humans. However, intrinsic interpretability is challenging to achieve in some widely used high-capacity models such as deep neural networks (LeCun et al., 2015). This has motivated **post-hoc interpretability** methods (Ribeiro et al., 2016; Lundberg & Lee, 2017), which seek to explain a pre-trained black-box model. While more broadly applicable, such explanations are often considered less suitable for scientific research (Rudin, 2019), as they may compromise stability and faithfulness to the model's decision process.

**Performance–Interpretability Trade-off.** The limited intrinsic interpretability observed in high-capacity models has long been recognized as a central challenge. This is commonly framed as the *performance–interpretability trade-off* (Rudin, 2019; Arrieta et al., 2020), which posits a tension between predictive performance and intrinsic interpretability. High-performing models such as deep neural networks often lack transparency, whereas intrinsically interpretable models struggle to capture complex nonlinear patterns. Several efforts have sought to mitigate the performance–interpretability trade-off, which can be broadly categorized into four groups. (i) Additive models. Classical GAMs (Hastie, 2017), modern GA2Ms/EBMs (Caruana et al., 2015; Nori et al., 2019), and neural variants such as NAM (Agarwal et al., 2021) and NODE-GAM (Chang et al., 2021) preserve interpretability by decomposing predictions into main effects and low-order interactions. (ii) Concept-based models. Concept Bottleneck Models (Koh et al., 2020), TCAV (Kim et al., 2018), and SENN (Alvarez Melis & Jaakkola, 2018) map inputs into human-interpretable latent concepts and use them as intermediate predictors. (iii) Rule- and score-based systems. SLIM (Ustun & Rudin, 2016) and CORELS (Angelino et al., 2018) generate transparent scoring functions or rule lists with provable optimality guarantees. (iv) Shape-constrained networks. Deep Lattice Networks (You et al., 2017) and related monotonic architectures impose monotonicity and calibration constraints to encode domain priors while retaining flexibility.

**Limitations in Scientifically Credible Modeling.** The above approaches demonstrate strengths, yet two fundamental limitations restrict their applicability in scientific research. First, most methods are tool-centric modifications of machine learning architectures rather than frameworks grounded in scientific theory (e.g., optimization, dynamical systems, conservation laws). As recent surveys emphasize (Roscher et al., 2020; Karniadakis et al., 2021; Allen et al., 2023; Bereska & Gavves, 2024; Longo et al., 2024; Mersha et al., 2024), genuine scientific insight requires models linked to mechanistic principles, yet many interpretability techniques remain detached from such principles. Second, these approaches are typically non-identifiable (Ran & Hu, 2017; Méloux et al., 2025), meaning that multiple distinct parameterizations can explain the same data. This lack of uniqueness undermines their reliability for recovering ground-truth mechanisms and, in statistical terms, complicates consistency guarantees. As a result, the trained model may fail to converge to the true data-generating process as sample size increases (Newey & McFadden, 1994; Van der Vaart, 2000).

**Relation to BL.** BL also mitigates the performance–interpretability trade-off. Unlike prior methods, it is principle-driven and scientifically grounded, learning interpretable latent optimization structures directly from data. The framework applies broadly to domains where outcomes arise as solutions to (explicit or latent) optimization problems. It is also identifiable: its smooth and monotone variant, Identifiable Behavior Learning (IBL), guarantees identifiability under mild conditions, ensuring the scientific credibility of its explanations and supporting recovery of the ground-truth model under appropriate conditions.

## C.2 DATA-DRIVEN INVERSE OPTIMIZATION

Inverse optimization (IO) (Ahuja & Orlin, 2001; Chan et al., 2025) is a core paradigm for learning latent optimization problems from observed data. Traditional IO aims to construct objectives or constraints that exactly rationalize a small set of deterministic decisions. In contrast, data-driven IO (Keshavarz et al., 2011; Aswani et al., 2018) focuses on statistically recovering the underlying problem from large-scale, noisy observational data. Inverse optimal control (IOC) (Kalman, 1964; Freeman & Kokotovic, 1996) extends this paradigm to dynamic settings, seeking to infer sequential decision processes from expert trajectories. Within *machine learning*, inverse reinforcement learning (IRL) (Ng et al., 2000; Wulfmeier et al., 2015) and inverse constrained reinforcement learning (ICRL) (Malik et al., 2021; Liu et al., 2024a) are prominent instances of data-driven IOC: Typically, IRL assumes fixed constraints and learns a reward function, whereas ICRL reverses this role. Both require repeatedly solving for (near-)optimal policies and matching with expert demonstrations—incurring high computational cost. In the *behavioral sciences*, particularly economics, numerous studies can be viewed as instances of the data-driven IO paradigm. Foundational work (McFadden, 1972; Dubin & McFadden, 1984; Hanemann, 1984; Berry et al., 1993) and related studies typically posits theoretically grounded, parametric utility maximization problems (UMPs) and estimates their structural parameters from observed behavior.

**Relation to BL.** The BL framework also falls under the paradigm of data-driven inverse optimization but differs notably from prior related work in both machine learning and behavioral science. Compared with IRL and ICRL, BL does not rely on matching expert-demonstrated policies with the aim of improving task-specific performance. Instead, it is proposed as a general-purpose, scientifically grounded, and intrinsically interpretable framework that operates via low-cost end-to-end training with a hybrid CE–DSM objective. It jointly learns a utility functions and constraints—a direction that has received little attention in IRL and ICRL (Park et al., 2020; Jang et al., 2023; Liu & Zhu, 2024). Meanwhile, in behavioral science, related work typically formulates distinct utility maximization models under varying assumptions for specific decision contexts, and estimate their parameters accordingly. However, to the best of our knowledge, no existing work proposes a structure-free framework for learning UMPs that generalizes across contexts. BL fills this gap with a structure-free, data-driven approach that does not rely on fixed UMP structures.

### C.3 Energy-based Models (EBMs)

Energy-based models (EBMs) (LeCun et al., 2006) are a prominent data-driven IO scheme, rooted in the principle of energy minimization from statistical physics. They learn an energy function $E_\theta(x, y)$ that parameterizes the compatibility between inputs and outputs, inducing a Gibbs distribution $p_\theta(y \mid x) \propto \exp\{-E_\theta(x, y)\}$ that favors outcomes corresponding to low-energy solutions. In practice, this energy function is almost always instantiated by high-capacity neural networks, endowing the learned landscape with strong expressive power but also a black-box nature. Training EBMs typically relies on objectives that circumvent the intractable partition function, with classical approaches including contrastive divergence (Hinton, 2002), persistent contrastive divergence (Tieleman, 2008), and noise-contrastive estimation (Gutmann & Hyvärinen, 2010). A particularly influential line of work is score matching (Hyvärinen & Dayan, 2005) and its denoising variant (DSM) (Vincent, 2011), which have underpinned breakthroughs in score-based generative modeling (Song & Ermon, 2019; 2020) and laid the foundation for modern diffusion methods (Song et al., 2020).

**Relation to BL.** BL and EBMs exhibit a principled correspondence: BL is grounded in behavioral science and rooted in utility maximization, while EBMs are grounded in statistical physics and based on energy minimization. BL adopts several training techniques common to EBMs, such as Gibbs distribution modeling and denoising score matching (DSM). However, the two frameworks differ substantially in model structure. EBMs primarily focus on generative quality and typically employ black-box neural networks to learn an opaque energy function with little regard for interpretability. In contrast, BL is built on the utility maximization problem (UMP) and its equivalence to penalty formulations, yielding a principled and scientifically grounded framework. Its architecture is composed of intrinsically interpretable blocks, each of which can be explicitly expressed in symbolic form as a UMP—a foundational paradigm in behavioral science and a universal optimization framework. These properties enable BL to jointly achieve high predictive performance, intrinsic interpretability, and identifiability, thereby supporting scientifically credible modeling that extends beyond mere generative capability.

## D Architecture Details

### D.1 Learning Scheme Details

**Input and output of the BL function.** We formulate BL as a direct mapping from input–output pairs to compositional utility representations:

$$\mathrm{BL}: \ \mathcal{X} \times \mathcal{Y} \to \mathbb{R}^{d_{\mathrm{out}}}, \qquad (x, y) \mapsto \mathrm{BL}(x, y) \in \mathbb{R}^{d_{\mathrm{out}}},$$

where the output dimension $d_{\mathrm{out}}$ is chosen according to the modeling choice. This formulation intentionally allows BL to return either a scalar or a vector for each $(x, y)$; the following cases are most common:

- Scalar per candidate (pointwise evaluation). Set $d_{\mathrm{out}} = 1$. Here $\mathrm{BL}(x, y) \in \mathbb{R}$ is a scalar compositional utility evaluated for the single candidate $y$. This view is natural for continuous $y$ (regression or density estimation) or when one prefers to evaluate candidates individually.

- Vectorized over a finite candidate set. If $\mathcal{Y} = \{y_1, \ldots, y_m\}$ is finite, one can choose $d_{\text{out}} = m$ and define the vector-valued output by stacking evaluations over the candidate set:

$$\mathrm{BL}(x) := \begin{bmatrix} \mathrm{BL}(x, y_1) \\ \vdots \\ \mathrm{BL}(x, y_m) \end{bmatrix} \in \mathbb{R}^m.$$

  This vectorized form is convenient for classification: it evaluates all class candidates at once and yields a single compositional utility vector per $x$.

- Flexibility and equivalence. The scalar and vector modes are compatible: the vectorized form is simply a batch of pointwise evaluations. Conversely, a scalar pointwise evaluator can be used to assemble a vector by repeated calls over a candidate set. The choice between pointwise (scalar) and vectorized outputs is therefore an engineering choice that trades off computational efficiency and convenience.

Given a dataset $\mathcal{D} = \{(x_i, y_i)\}_{i=1}^n$, training and inference may use either mode: vectorized computation where feasible (e.g., small finite $\mathcal{Y}$), or pointwise evaluation when $\mathcal{Y}$ is large or continuous.

**Conditional Gibbs model.** Let $(x, y) \sim \mathcal{D}$ with $x \in \mathbb{R}^d$ and $y = (y^{\text{disc}}, y^{\text{cont}}) \in \mathcal{Y}_{\text{disc}} \times \mathbb{R}^{m_c}$ (discrete, continuous, or hybrid). BL induces a conditional Gibbs distribution with temperature $\tau > 0$:

$$p_\tau(y \mid x) = \frac{\exp\{\mathrm{BL}(x, y)/\tau\}}{Z_\tau(x)}, \qquad Z_\tau(x) = \int_{\mathcal{Y}} \exp\{\mathrm{BL}(x, y')/\tau\} \, dy'.$$

For discrete $\mathcal{Y} = \{y_1, \ldots, y_m\}$, if we choose the vector-output formulation, we define

$$\mathrm{BL}(x) := \begin{bmatrix} \mathrm{BL}(x, y_1), \ldots, \mathrm{BL}(x, y_m) \end{bmatrix} \in \mathbb{R}^m,$$

so that the conditional distribution reduces to a softmax over this compositional utility vector:

$$p_\tau(y = k \mid x) = \mathrm{softmax}_k\left(\tfrac{1}{\tau}\mathrm{BL}(x)\right).$$

Behaviorally, $\tau$ encodes *noisy rationality*; as $\tau \to 0$, $p_\tau(\cdot \mid x)$ concentrates on $\arg\max_y \mathrm{BL}(x, y)$, corresponding to the deterministic optimal choice implied by the learned model.

**Supervised, unsupervised, and generative uses.** BL accommodates multiple regimes. (i) Supervised: take $x$ as input and $y$ as label. For discrete $y$, one may either (a) adopt the vector-output formulation, where $\mathrm{BL}(x) \in \mathbb{R}^m$ yields a compositional utility vector over all classes and the likelihood is given by a softmax, or (b) adopt the scalar-output formulation, where $\mathrm{BL}(x, y)$ is evaluated separately for each candidate and then normalized across classes. For continuous $y$, BL naturally operates in the scalar-output mode, treating $\mathrm{BL}(x, y) \in \mathbb{R}$ as a compositional utility field. (ii) Unsupervised / generative: model a marginal $p(y) \propto \exp\{\mathrm{BL}(y)/\tau\}$ (empty $x$) or a joint $p(x, y) \propto \exp\{\mathrm{BL}(x, y)/\tau\}$; sampling the Gibbs distribution yields a generator.

**Learning objective.** Since the response $\mathbf{y}$ may contain both discrete and continuous components, we estimate $\theta$ by minimizing a type-specific risk:

$$\mathcal{L}(\theta) = \gamma_d \, \mathbb{E}\left[-\log p_\tau(y^{\text{disc}} \mid x)\right] + \gamma_c \, \mathbb{E}\left\|\nabla_{\tilde{y}^{\text{cont}}} \log p_\tau(\tilde{y}^{\text{cont}} \mid x) + \sigma^{-2}(\tilde{y}^{\text{cont}} - y^{\text{cont}})\right\|^2,$$

where the first term is cross-entropy on the discrete component and the second is denoising score matching (DSM) on the continuous component with $\tilde{y}^{\text{cont}} = y^{\text{cont}} + \varepsilon$, $\varepsilon \sim \mathcal{N}(0, \sigma^2 I)$. Set $(\gamma_d, \gamma_c) = (1, 0)$ for purely discrete outputs, $(0, 1)$ for purely continuous outputs, and $(> 0, > 0)$ for hybrids.

## D.2 MODEL STRUCTURE DETAILS

In the main text we adopted a compact notation for BL; here we present an equivalent, more explicit matrix/vector formulation that makes dimensions, linear maps, and the per-head parameterizations explicit, which is useful for formal proofs and for implementation details.

**Fixed bases and head pre-activations.** For a block input $z$ (specified below), let

$$m_u(z) \in \mathbb{R}^{d_u}, \qquad m_c(z) \in \mathbb{R}^{d_c}, \qquad m_t(z) \in \mathbb{R}^{d_t}$$

denote fixed basis (e.g., monomial) vectors. Learnable linear maps produce head pre-activations:

$$u(z) := M_u\, m_u(z) + b_u \in \mathbb{R}^{r_u}, \quad c(z) := M_c\, m_c(z) + b_c \in \mathbb{R}^{r_c}, \quad t(z) := M_t\, m_t(z) + b_t \in \mathbb{R}^{r_t},$$

with $M_u \in \mathbb{R}^{r_u \times d_u}$, $M_c \in \mathbb{R}^{r_c \times d_c}$, $M_t \in \mathbb{R}^{r_t \times d_t}$ and optional biases $b_\bullet$.

**Single BL block.** A single modular block is

$$\mathcal{B}(z) = \lambda_0^\top\, \phi\big(u(z)\big) - \lambda_1^\top\, \rho\big(c(z)\big) - \lambda_2^\top\, \psi\big(t(z)\big), \tag{12}$$

where $\lambda_0 \in \mathbb{R}^{r_u}$, $\lambda_1 \in \mathbb{R}^{r_c}$, $\lambda_2 \in \mathbb{R}^{r_t}$ are learnable weights, and $\phi, \rho, \psi$ act coordinatewise with the roles specified in Theorem 2.1 (increasing $\phi$ for utility, penalty $\rho$ for inequality violations, symmetric $\psi$ for equalities). Identifying

$$U_{\theta_U}(x, y) = u\big(z = (x, y)\big), \quad \mathcal{C}_{\theta_C}(x, y) = c\big(z = (x, y)\big), \quad \mathcal{T}_{\theta_T}(x, y) = t\big(z = (x, y)\big),$$

substituting into equation 12 recovers the main-text parameterization in equation 4.

**Layer of parallel blocks.** A layer $\mathbb{B}_\ell$ stacks $d_\ell$ parallel copies of equation 12 with (possibly) distinct parameters $\theta_{\ell,i}$:

$$\mathbb{B}_\ell(z_\ell) := \begin{bmatrix} \mathcal{B}_{\theta_{\ell,1}}(z_\ell) \\ \vdots \\ \mathcal{B}_{\theta_{\ell,d_\ell}}(z_\ell) \end{bmatrix} \in \mathbb{R}^{d_\ell}.$$

We adopt the standard layered (feedforward) form:

$$z_1 := (x, y), \qquad z_{\ell+1} := \mathbb{B}_\ell(z_\ell) \quad (\ell = 1, \dots, L-1),$$

so that each layer's input is simply the previous layer's output. This is the canonical feedforward architecture.

Optionally, one may allow each layer to explicitly access the original inputs:

$$z_1 := (x, y), \qquad z_{\ell+1} := \mathbb{B}_\ell\big((x, y),\, z_\ell\big).$$

To improve trainability one may also use residual connections:

$$z_{\ell+1} := z_\ell + \mathbb{B}_\ell(z_\ell).$$

**Shallow/Deep composition and final affine readout.** For depth $L \geq 1$, the BL compositional utility is produced by a final learnable affine transformation of the top layer:

$$\mathrm{BL}(x, y) = W_L\, \mathbb{B}_L\big(z_L\big) + b_L, \tag{13}$$

with $W_L \in \mathbb{R}^{1 \times d_L}$ for scalar output or $W_L \in \mathbb{R}^{m \times d_L}$ for vector output, and bias $b_L$ of matching dimension. The cases $L = 1$ (with $d_1 = 1$), $L \leq 2$, and $L > 2$ correspond to BL(Single), BL(Shallow), and BL(Deep), respectively, exactly as described in the main text.

### D.3 Implementation Details

#### D.3.1 Function Instantiation

**Default instantiation.** In practice, we instantiate equation 4 with the specific choice $(\phi, \rho, \psi) = (\tanh, \mathrm{ReLU}, |\cdot|)$:

$$\mathcal{B}(x, y; \theta) = \lambda_0^\top \tanh\big(U_{\theta_U}(x, y)\big) - \lambda_1^\top \mathrm{ReLU}\big(\mathcal{C}_{\theta_C}(x, y)\big) - \lambda_2^\top \big|\mathcal{T}_{\theta_T}(x, y)\big|. \tag{14}$$

Here $\lambda_0, \lambda_1, \lambda_2$ are learnable nonnegative weights. The bounded $\tanh$ captures saturation effects and diminishing returns in the utility head (Jevons, 2013), while ReLU and $|\cdot|$ impose asymmetric (one-sided) and symmetric (two-sided) penalties for inequality and equality violations.

**Variants and simplifications.** Several variants of equation 14 are often useful:

- Identity utility head. Set $\phi = \mathrm{id}$ so the utility head uses raw polynomials:

$$\mathcal{B} = \lambda_0^\top U_{\theta_U} - \lambda_1^\top \mathrm{ReLU}(\mathcal{C}_{\theta_C}) - \lambda_2^\top |\mathcal{T}_{\theta_T}|.$$

- Smooth penalty alternatives. Replace $\mathrm{ReLU}$ with softplus to yield smooth inequality penalties, or replace $|\cdot|$ with Huber or squared penalties to modulate sensitivity near zero for equality terms.

- Dropping heads. The framework is modular, so one may omit heads depending on the task:
  - No $T$ head: ignores symmetric deviations, yielding a constrained maximization with only inequality penalties.

  - No $C$ head: if the $T$ head is retained, the model reduces to a maximization problem with only equality constraints; if $T$ is also removed, it becomes a fully unconstrained maximization.

  - No $U$ head: produces a pure (soft-)constraint model focusing on feasibility.

  Strikingly, removing both $U$ and $T$ leaves only piecewise-linear $\mathrm{ReLU}$ penalties; when followed by a final affine readout, the resulting architecture becomes highly similar to a standard MLP—suggesting that MLPs may be viewed as a closely related special instance within the broader BL framework.

### D.3.2 POLYNOMIAL FEATURE MAPS AND LINEAR REDUCTIONS

We adopt a pragmatic default: use low-degree polynomial maps for single-block models to maximize interpretability, and use affine (degree-1) maps inside blocks for shallow/deep stacks to control parameter growth and compute. Below we state the instantiations and give the final block formulas used in experiments.

**BL(Single) — polynomial instantiation.** Let $m_D(x, y)$ denote a fixed basis of monomials up to total degree $D$ (e.g. $D \leq 2$):

$$m_D(x, y) = \begin{bmatrix} x, & y, & \mathrm{vec}(xx^\top), & \mathrm{vec}(xy^\top), & \mathrm{vec}(yy^\top), \dots \end{bmatrix}^\top.$$

Parameterize each map as a linear map on this basis:

$$U_{\theta_U}(x, y) = M_U\, m_D(x, y) + b_U, \quad \mathcal{C}_{\theta_C}(x, y) = M_C\, m_D(x, y) + b_C, \quad \mathcal{T}_{\theta_T}(x, y) = M_T\, m_D(x, y) + b_T,$$

with learnable matrices $M_\bullet$ and biases $b_\bullet$. The block becomes

$$\mathcal{B}(x, y; \theta) = \lambda_0^\top \phi(M_U m_D + b_U) - \lambda_1^\top \rho(M_C m_D + b_C) - \lambda_2^\top \psi(M_T m_D + b_T).$$

**BL (Shallow/Deep) — linear-by-layer instantiation.** For stacked architectures (Shallow/Deep) we use affine maps inside each block to keep per-layer complexity low:

$$U_{\theta_U}(x, y) = A_U\, [x; y] + b_U, \quad \mathcal{C}_{\theta_C}(x, y) = A_C\, [x; y] + b_C, \quad \mathcal{T}_{\theta_T}(x, y) = A_T\, [x; y] + b_T,$$

with learnable $A_\bullet$ and $b_\bullet$. The corresponding block is

$$\mathcal{B}(x, y; \theta) = \lambda_0^\top \phi(A_U[x; y] + b_U) - \lambda_1^\top \rho(A_C[x; y] + b_C) - \lambda_2^\top \psi(A_T[x; y] + b_T).$$

**On-demand higher-order terms.** If diagnostics or domain knowledge indicate underfitting, we optionally augment the affine maps with selected higher-order terms or interactions. Concretely, this is done by appending a small set of monomials (e.g. $x_i y_j$, $x_i^2$, $y_k^2$) to the input vector $[x; y]$ and re-estimating the same affine maps $A_\bullet$. This targeted augmentation preserves the base affine parameterization, increases expressivity only where required, and keeps both computational and statistical costs modest while retaining interpretability.

### D.3.3 SKIP CONNECTIONS

Skip connections are optional in our implementation. When beneficial, we often consider two patterns tailored to BL: a DenseNet-style (concatenative) variant and a ResNet-style (additive) variant.

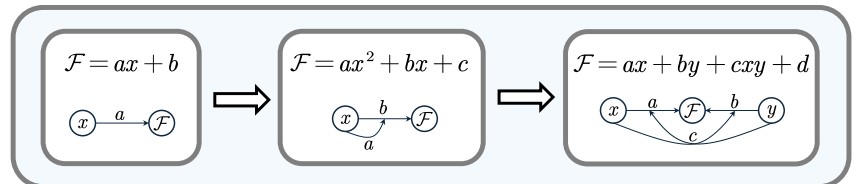

Figure 6: Visualization of polynomial feature maps as computation graphs, where nodes represent variables or outputs and edges represent their effects. The left panel illustrates the linear form $\mathcal{F} = ax + b$, in which the single edge $x \to \mathcal{F}$ directly encodes the marginal effect of $x$ on $\mathcal{F}$. The middle panel shows the quadratic form $\mathcal{F} = ax^2 + bx + c$, where $x$ not only has a direct edge $x \to \mathcal{F}$ but also acts on its own edge ("$x \to \mathcal{F}$"), thereby modifying the strength of its self-effect through a higher-order contribution. The right panel depicts the interaction form $\mathcal{F} = ax + by + cxy + d$, where $y$ has an edge $y \to \mathcal{F}$ and, in addition, $x$ acts on this edge ("$y \to \mathcal{F}$"), thereby modulating the strength of $y$'s contribution to $\mathcal{F}$. Symmetrically, $y$ may act on the edge ("$x \to \mathcal{F}$"), so that each variable can reshape the other's effect through the interaction term.

**Dense skip connections (DenseNet-style, concatenation).**   This variant feeds each layer with the concatenation of all preceding representations, mirroring DenseNet (Huang et al., 2017). Let

$$z_1 := [\,x;\, y\,], \qquad s_1 := \mathbb{B}_1(z_1) \in \mathbb{R}^{d_1}.$$

For $\ell \geq 2$,

$$z_\ell := [\,x;\, y;\, s_1;\, \ldots;\, s_{\ell-1}\,], \qquad s_\ell := \mathbb{B}_\ell(z_\ell) \in \mathbb{R}^{d_\ell}.$$

The final compositional utility is read out as

$$\mathrm{BL}(x, y) \;=\; W_L\, s_L + b_L.$$

*Pros.* By exposing all earlier block outputs explicitly as inputs to later blocks, dense skips preserve a transparent feature trail: one can trace which intermediate $\mathcal{B}$-block outputs enter downstream computations and the final affine readout. This often improves feature reuse and yields favorable interpretability at the block level.

**Residual skip connections (ResNet-style, addition).**   This variant adds an identity (or projected) shortcut to each layer, as in ResNet (He et al., 2016). Define

$$z_1 := [\,x;\, y\,], \qquad s_1 := \mathbb{B}_1(z_1) \in \mathbb{R}^{d_1},$$

and for $\ell \geq 2$,

$$s_\ell := \mathbb{B}_\ell(s_{\ell-1}) \;+\; \Pi_\ell\, s_{\ell-1}, \qquad \Pi_\ell \in \mathbb{R}^{d_\ell \times d_{\ell-1}},$$

where $\Pi_\ell$ is the identity if $d_\ell = d_{\ell-1}$, or a bias-free learnable projection otherwise. The readout is again

$$\mathrm{BL}(x, y) \;=\; W_L\, s_L + b_L.$$

**Skip Connections and Interpretability.**   Skip connections introduce explicit cross-layer dependency structures, a form widely studied in statistical physics and other scientific domains. Such structures enhance scientific interpretability by making long-range influences transparent. In behavioral and organizational sciences, they capture situations in which lower-level agents directly affect higher-level decision makers without routing through intermediate layers. In physics, microscopic parameters can exert direct effects on macroscopic behaviors across multiple scales. Architecturally, ResNet-style skip connections model linear cross-layer dependencies, whereas DenseNet-style connections realize concatenative (information-replicating) dependencies. These mechanisms provide flexible yet interpretable pathways for representing hierarchical interactions.

# E   PROOFS OF THEOREMS

## E.1   UTILITY MAXIMIZATION PROBLEM (UMP)

**Theorem 2.1** (Local Exact Penalty Reformulation for UMP).

*Let $\mathcal{X} \subset \mathbb{R}^{d_x}$ and $\mathcal{Y} \subset \mathbb{R}^{d_y}$ be nonempty compact sets, and let $U : \mathcal{X} \times \mathcal{Y} \to \mathbb{R}$, $\mathcal{C} : \mathcal{X} \times \mathcal{Y} \to \mathbb{R}^m$, and $\mathcal{T} : \mathcal{X} \times \mathcal{Y} \to \mathbb{R}^p$ be $C^1$. Consider the Utility Maximization Problem (UMP)*

$$\max_{\mathbf{y} \in \mathcal{Y}} U(\mathbf{x}, \mathbf{y}) \quad s.t. \quad \mathcal{C}(\mathbf{x}, \mathbf{y}) \leq 0, \quad \mathcal{T}(\mathbf{x}, \mathbf{y}) = 0. \tag{15}$$

*Assume there exists a feasible point $\mathbf{y}^\star \in \mathrm{int}(\mathcal{Y})$ which is a strict local maximizer of equation 15 and the Han–Mangasarian constraint qualification (2.1) holds at $\mathbf{y}^\star$ (in the notation of Han & Mangasarian (1979)). Let $\phi : \mathbb{R} \to \mathbb{R}$ be strictly increasing and $C^1$, and define $\rho(z) := \max\{z, 0\}$ and $\psi(z) := |z|$ (componentwise on $\mathbb{R}^m$ and $\mathbb{R}^p$). Then there exist $\lambda_0 > 0$, $\lambda_1 \in \mathbb{R}^m_{++}$, and $\lambda_2 \in \mathbb{R}^p_{++}$ such that $\mathbf{y}^\star$ is a local maximizer of*

$$\max_{\mathbf{y} \in \mathcal{Y}} \lambda_0 \, \phi\big(U(\mathbf{x}, \mathbf{y})\big) - \lambda_1^\top \rho\big(\mathcal{C}(\mathbf{x}, \mathbf{y})\big) - \lambda_2^\top \psi\big(\mathcal{T}(\mathbf{x}, \mathbf{y})\big). \tag{16}$$

*Proof.* Fix $\mathbf{x} \in \mathcal{X}$ and abbreviate

$$g(\mathbf{y}) := \mathcal{C}(\mathbf{x}, \mathbf{y}) \in \mathbb{R}^m, \qquad h(\mathbf{y}) := \mathcal{T}(\mathbf{x}, \mathbf{y}) \in \mathbb{R}^p.$$

Feasibility of $\mathbf{y}^\star$ means $g(\mathbf{y}^\star) \leq 0$ componentwise and $h(\mathbf{y}^\star) = 0$.

**Step 1: Convert to a constrained local minimization problem in the ambient space.** Pick any $\lambda_0 > 0$ and define

$$f(\mathbf{y}) := -\lambda_0 \, \phi\big(U(\mathbf{x}, \mathbf{y})\big). \tag{17}$$

Since $\phi$ is strictly increasing, for any $\mathbf{y}_1, \mathbf{y}_2$ we have $U(\mathbf{x}, \mathbf{y}_1) > U(\mathbf{x}, \mathbf{y}_2)$ if and only if $f(\mathbf{y}_1) < f(\mathbf{y}_2)$. Hence $\mathbf{y}^\star$ is a strict local maximizer of equation 15 if and only if $\mathbf{y}^\star$ is a strict local minimizer of

$$\min_{\mathbf{y} \in \mathcal{Y}} f(\mathbf{y}) \quad s.t. \quad g(\mathbf{y}) \leq 0, \quad h(\mathbf{y}) = 0. \tag{18}$$

Now use the interior-point assumption $\mathbf{y}^\star \in \mathrm{int}(\mathcal{Y})$: there exists $\varepsilon_0 > 0$ such that $B_{\varepsilon_0}(\mathbf{y}^\star) \subset \mathcal{Y}$. Therefore, the notion of strict local minimizer over $\mathcal{Y}$ at $\mathbf{y}^\star$ coincides with the ambient-space notion: for any function $F$, there exists $\varepsilon \in (0, \varepsilon_0]$ such that

$$F(\mathbf{y}^\star) < F(\mathbf{y}) \quad \forall \mathbf{y} \in \big(\mathcal{Y} \cap B_\varepsilon(\mathbf{y}^\star)\big) \setminus \{\mathbf{y}^\star\}$$

if and only if

$$F(\mathbf{y}^\star) < F(\mathbf{y}) \quad \forall \mathbf{y} \in B_\varepsilon(\mathbf{y}^\star) \setminus \{\mathbf{y}^\star\}.$$

Hence $\mathbf{y}^\star$ is a strict local minimizer of the constrained problem equation 18 in the ambient-space sense.

Moreover, by assumption, the triple $(f, g, h)$ is continuously differentiable on a neighborhood of $\mathbf{y}^\star$.

**Step 2: Embed vector weights into a norm and build a Han–Mangasarian penalty.** Define the positive part $g_+(\mathbf{y}) \in \mathbb{R}^m$ componentwise by $(g_+(\mathbf{y}))_i := \max\{g_i(\mathbf{y}), 0\}$. Let $\lambda_1 \in \mathbb{R}^m_{++}$ and $\lambda_2 \in \mathbb{R}^p_{++}$ be arbitrary for the moment, and define a norm on $\mathbb{R}^{m+p}$ by the weighted $\ell_1$-norm

$$\|(u, v)\|_\lambda := \lambda_1^\top |u| + \lambda_2^\top |v|, \qquad (u, v) \in \mathbb{R}^m \times \mathbb{R}^p, \tag{19}$$

where $|\cdot|$ is componentwise absolute value. Since $\lambda_1, \lambda_2$ have strictly positive entries, $\|\cdot\|_\lambda$ is indeed a norm.

Choose the scalar penalty function $Q : [0, \infty) \to [0, \infty)$ as $Q(t) = t$. Then $Q$ satisfies the penalty regularity condition (1.3) in Han & Mangasarian (1979), in particular $Q'(0+) = 1 > 0$. Define for $\alpha \geq 0$ the penalty function

$$P(\mathbf{y}, \alpha) := f(\mathbf{y}) + \alpha \, Q\big(\big\|\big(g_+(\mathbf{y}), h(\mathbf{y})\big)\big\|_\lambda\big). \tag{20}$$

Expanding equation 20 using equation 19 and $Q(t) = t$ yields

$$P(\mathbf{y}, \alpha) = -\lambda_0 \, \phi\big(U(\mathbf{x}, \mathbf{y})\big) + \alpha \Big[\lambda_1^\top g_+(\mathbf{y}) + \lambda_2^\top |h(\mathbf{y})|\Big]. \tag{21}$$

Since $\rho(g(\mathbf{y})) = g_+(\mathbf{y})$ and $\psi(h(\mathbf{y})) = |h(\mathbf{y})|$ componentwise, equation 21 can be rewritten as

$$P(\mathbf{y}, \alpha) = -\lambda_0 \, \phi\big(U(\mathbf{x}, \mathbf{y})\big) + \alpha \Big[\lambda_1^\top \rho\big(g(\mathbf{y})\big) + \lambda_2^\top \psi\big(h(\mathbf{y})\big)\Big]. \tag{22}$$

**Step 3: Apply Han–Mangasarian Theorem 4.4.** By Steps 1–2, the functions $f, g, h$ are $C^1$ on a neighborhood of $\mathbf{y}^\star$, $\mathbf{y}^\star$ is a strict local minimizer (ambient-space sense) of the constrained problem equation 18, and the Han–Mangasarian constraint qualification (2.1) holds at $\mathbf{y}^\star$. Therefore, by (Han & Mangasarian, 1979, Thm. 4.4), there exists $\bar\alpha \geq 0$ such that for every $\alpha \geq \bar\alpha$, $\mathbf{y}^\star$ is a local minimizer of $P(\cdot, \alpha)$.

**Step 4: Return to local maximization and ensure strictly positive vector weights.** Choose any

$$\alpha > \max\{\bar\alpha, 0\}, \tag{23}$$

so in particular $\alpha > 0$. Define the penalized maximization objective

$$\widetilde{F}(\mathbf{y}) := -P(\mathbf{y}, \alpha) = \lambda_0\, \phi\big(U(\mathbf{x}, \mathbf{y})\big) - \alpha\, \lambda_1^\top \rho\big(g(\mathbf{y})\big) - \alpha\, \lambda_2^\top \psi\big(h(\mathbf{y})\big).$$

Since $\mathbf{y}^\star$ is a local minimizer of $P(\cdot, \alpha)$, it is a local maximizer of $\widetilde{F}$. Finally set

$$\lambda_1' := \alpha\, \lambda_1 \in \mathbb{R}^m_{++}, \qquad \lambda_2' := \alpha\, \lambda_2 \in \mathbb{R}^p_{++}.$$

Then $\widetilde{F}(\mathbf{y})$ equals

$$\lambda_0\, \phi\big(U(\mathbf{x}, \mathbf{y})\big) - (\lambda_1')^\top \rho\big(\mathcal{C}(\mathbf{x}, \mathbf{y})\big) - (\lambda_2')^\top \psi\big(\mathcal{T}(\mathbf{x}, \mathbf{y})\big),$$

which is precisely the objective in equation 16. Hence $\mathbf{y}^\star$ is a local maximizer of equation 16 over $\mathcal{Y}$. Since $\mathbf{y}^\star \in \mathrm{int}(\mathcal{Y})$, this is equivalent to local maximality in the ambient space sense. This completes the proof. $\qquad\square$

**Theorem 2.2** (Universality of UMP). *Let $\mathcal{X}$ and $\mathcal{Y}$ be arbitrary nonempty sets. Let $f : \mathcal{X} \times \mathcal{Y} \to \mathbb{R}$ be an objective and let*

$$\{g_i\}_{i \in I_\leq}, \quad \{\tilde g_k\}_{k \in I_\geq}, \quad \{h_j\}_{j \in J}$$

*be (possibly empty, countable, or uncountable) families of real–valued constraint functions on $\mathcal{X} \times \mathcal{Y}$. For each fixed $\mathbf{x} \in \mathcal{X}$, consider the optimization problem*

$$\sup_{\mathbf{y} \in \mathcal{Y}} f(\mathbf{x}, \mathbf{y}) \quad s.t. \quad g_i(\mathbf{x}, \mathbf{y}) \leq 0 \; (i \in I_\leq), \; \tilde g_k(\mathbf{x}, \mathbf{y}) \geq 0 \; (k \in I_\geq), \; h_j(\mathbf{x}, \mathbf{y}) = 0 \; (j \in J). \tag{24}$$

*Define (with the convention $\sup \varnothing := -\infty$ and maxima taken in the extended reals)*

$$U(\mathbf{x}, \mathbf{y}) := f(\mathbf{x}, \mathbf{y}), \qquad \mathcal{C}(\mathbf{x}, \mathbf{y}) := \max\Big\{ 0, \; \sup_{i \in I_\leq} g_i(\mathbf{x}, \mathbf{y}), \; \sup_{k \in I_\geq} \big(-\tilde g_k(\mathbf{x}, \mathbf{y})\big) \Big\},$$

$$\mathcal{T}(\mathbf{x}, \mathbf{y}) := \max\Big\{ 0, \; \sup_{j \in J} |h_j(\mathbf{x}, \mathbf{y})| \Big\}.$$

*Then for every $\mathbf{x} \in \mathcal{X}$, problem equation 24 is equivalent to the utility–maximization problem*

$$\sup_{\mathbf{y} \in \mathcal{Y}} U(\mathbf{x}, \mathbf{y}) \quad s.t. \quad \mathcal{C}(\mathbf{x}, \mathbf{y}) \leq 0, \qquad \mathcal{T}(\mathbf{x}, \mathbf{y}) = 0, \tag{25}$$

*in the sense that the feasible sets of equation 24 and equation 25 coincide; hence the optimal values coincide, and whenever maximizers exist, the argmax sets coincide. For minimization problems, replace $U$ by $-f$.*

*Proof.* Fix $\mathbf{x} \in \mathcal{X}$. Let

$$F(\mathbf{x}) := \Big\{ \mathbf{y} \in \mathcal{Y} : \; g_i(\mathbf{x}, \mathbf{y}) \leq 0 \; \forall i \in I_\leq, \; \tilde g_k(\mathbf{x}, \mathbf{y}) \geq 0 \; \forall k \in I_\geq, \; h_j(\mathbf{x}, \mathbf{y}) = 0 \; \forall j \in J \Big\}$$

denote the feasible set of equation 24, and let

$$\hat F(\mathbf{x}) := \Big\{ \mathbf{y} \in \mathcal{Y} : \; \mathcal{C}(\mathbf{x}, \mathbf{y}) \leq 0, \; \mathcal{T}(\mathbf{x}, \mathbf{y}) = 0 \Big\}$$

denote the feasible set of equation 25. We prove that $F(\mathbf{x}) = \hat F(\mathbf{x})$.

*(i) $F(\mathbf{x}) \subseteq \hat F(\mathbf{x})$.* Let $\mathbf{y} \in F(\mathbf{x})$. Then $g_i(\mathbf{x}, \mathbf{y}) \leq 0$ for all $i \in I_\leq$, hence

$$\sup_{i \in I_\leq} g_i(\mathbf{x}, \mathbf{y}) \leq 0.$$

Similarly, $\tilde{g}_k(\mathbf{x}, \mathbf{y}) \geq 0$ for all $k \in I_\geq$ implies $-\tilde{g}_k(\mathbf{x}, \mathbf{y}) \leq 0$ for all $k$, hence

$$\sup_{k \in I_\geq} \left( -\tilde{g}_k(\mathbf{x}, \mathbf{y}) \right) \leq 0.$$

Moreover, $h_j(\mathbf{x}, \mathbf{y}) = 0$ for all $j \in J$ implies $|h_j(\mathbf{x}, \mathbf{y})| = 0$ for all $j \in J$, hence

$$\sup_{j \in J} |h_j(\mathbf{x}, \mathbf{y})| \leq 0 \qquad \text{(with the convention } \sup \varnothing = -\infty \text{)}.$$

By definition,

$$\mathcal{C}(\mathbf{x}, \mathbf{y}) = \max\left\{ 0, \; \sup_{i \in I_\leq} g_i(\mathbf{x}, \mathbf{y}), \; \sup_{k \in I_\geq} \left( -\tilde{g}_k(\mathbf{x}, \mathbf{y}) \right) \right\} = 0,$$

and

$$\mathcal{T}(\mathbf{x}, \mathbf{y}) = \max\left\{ 0, \; \sup_{j \in J} |h_j(\mathbf{x}, \mathbf{y})| \right\} = 0.$$

Thus $\mathbf{y} \in \hat{F}(\mathbf{x})$.

*(ii)* $\hat{F}(\mathbf{x}) \subseteq F(\mathbf{x})$. Let $\mathbf{y} \in \hat{F}(\mathbf{x})$. Set

$$A := \sup_{i \in I_\leq} g_i(\mathbf{x}, \mathbf{y}), \qquad B := \sup_{k \in I_\geq} \left( -\tilde{g}_k(\mathbf{x}, \mathbf{y}) \right), \qquad S := \sup_{j \in J} |h_j(\mathbf{x}, \mathbf{y})|.$$

Then

$$\mathcal{C}(\mathbf{x}, \mathbf{y}) = \max\{0, A, B\} \leq 0.$$

Since $0 \leq \max\{0, A, B\}$ always holds, we have $\max\{0, A, B\} = 0$, and in particular $A \leq 0$ and $B \leq 0$. Using the basic property of the supremum, for every $i \in I_\leq$ we have

$$g_i(\mathbf{x}, \mathbf{y}) \leq \sup_{i \in I_\leq} g_i(\mathbf{x}, \mathbf{y}) = A \leq 0,$$

and for every $k \in I_\geq$ we have

$$-\tilde{g}_k(\mathbf{x}, \mathbf{y}) \leq \sup_{k \in I_\geq} \left( -\tilde{g}_k(\mathbf{x}, \mathbf{y}) \right) = B \leq 0,$$

i.e., $\tilde{g}_k(\mathbf{x}, \mathbf{y}) \geq 0$.

Next, $\mathcal{T}(\mathbf{x}, \mathbf{y}) = 0$ means

$$0 = \mathcal{T}(\mathbf{x}, \mathbf{y}) = \max\{0, S\},$$

hence $S \leq 0$. Since $|h_j(\mathbf{x}, \mathbf{y})| \geq 0$ for every $j \in J$ and $|h_j(\mathbf{x}, \mathbf{y})| \leq S \leq 0$, it follows that

$$|h_j(\mathbf{x}, \mathbf{y})| = 0 \quad \text{for all } j \in J,$$

equivalently $h_j(\mathbf{x}, \mathbf{y}) = 0$ for all $j \in J$. Therefore $\mathbf{y} \in F(\mathbf{x})$.

Combining (i) and (ii) yields $F(\mathbf{x}) = \hat{F}(\mathbf{x})$. Since $U(\mathbf{x}, \mathbf{y}) = f(\mathbf{x}, \mathbf{y})$ (and for minimization problems one may equivalently optimize $-f$), the two problems optimize the same objective over the same feasible set. Consequently, their optimal values coincide, and whenever maximizers exist, their argmax sets coincide. $\qquad\square$

### E.2   BL ARCHITECTURE

**Theorem 2.3** (Universal Approximation of BL). *Let $\mathcal{X} \subset \mathbb{R}^d$ and $\mathcal{Y} \subset \mathbb{R}^m$ be compact sets, and let $p^\star(\mathbf{y} \mid \mathbf{x})$ be any continuous conditional density such that $p^\star(\mathbf{y} \mid \mathbf{x}) > 0$ for all $(\mathbf{x}, \mathbf{y}) \in \mathcal{X} \times \mathcal{Y}$. Then for any $\tau > 0$ and $\varepsilon > 0$, there exists a finite BL architecture (with some depth and width depending on $\varepsilon$) and a parameter $\theta^\star$ such that the Gibbs distribution*

$$p_\tau(\mathbf{y} \mid \mathbf{x}; \theta^\star) = \frac{\exp\left( BL_{\theta^\star}(\mathbf{x}, \mathbf{y})/\tau \right)}{\int_{\mathcal{Y}} \exp\left( BL_{\theta^\star}(\mathbf{x}, \mathbf{y}')/\tau \right) d\mathbf{y}'} \tag{26}$$

*satisfies*

$$\sup_{\mathbf{x} \in \mathcal{X}} \mathrm{KL}\left( p^\star(\cdot \mid \mathbf{x}) \,\|\, p_\tau(\cdot \mid \mathbf{x}; \theta^\star) \right) < \varepsilon. \tag{27}$$

*Proof.* **Step 0 (bounded log-density).** Define $f(\mathbf{x}, \mathbf{y}) := \log p^\star(\mathbf{y} \mid \mathbf{x})$. Since $p^\star$ is continuous and strictly positive on the compact set $\mathcal{X} \times \mathcal{Y}$, it attains a positive minimum and finite maximum. Hence $f \in C(\mathcal{X} \times \mathcal{Y})$ and is bounded.

**Step 1 (the BL block contains a one-hidden-layer $\tanh$ network).** Recall the elementary block

$$\mathcal{B}(\mathbf{x}, \mathbf{y}; \theta) := \lambda_0^\top \tanh\big(\mathbf{p}_u(\mathbf{x}, \mathbf{y})\big) - \lambda_1^\top \mathrm{ReLU}\big(\mathbf{p}_c(\mathbf{x}, \mathbf{y})\big) - \lambda_2^\top \big|\mathbf{p}_t(\mathbf{x}, \mathbf{y})\big|. \tag{28}$$

Set $\lambda_1 = \mathbf{0}$ and $\lambda_2 = \mathbf{0}$. Choose $\mathbf{p}_u(\mathbf{x}, \mathbf{y})$ to be affine in $[\mathbf{x}; \mathbf{y}]$, i.e. $\mathbf{p}_u(\mathbf{x}, \mathbf{y}) = W[\mathbf{x}; \mathbf{y}] + b \in \mathbb{R}^k$ for some $k \in \mathbb{N}$. Then

$$\mathcal{B}(\mathbf{x}, \mathbf{y}; \theta) = \lambda_0^\top \tanh\big(W[\mathbf{x}; \mathbf{y}] + b\big), \tag{29}$$

which is a standard one-hidden-layer $\tanh$ network on the compact domain $\mathcal{X} \times \mathcal{Y}$.

If $\lambda_0 \in \mathbb{R}^k$ is unconstrained, equation 29 is the classical universal approximation class. If instead one imposes $\lambda_0 \geq \mathbf{0}$ componentwise, the same expressivity is retained because $\tanh$ is odd: for any scalar $a \in \mathbb{R}$, write $a = a^+ - a^-$ with $a^\pm \geq 0$, and note $a \tanh(h) = a^+ \tanh(h) + a^- \tanh(-h)$. Since $-h$ is affine whenever $h$ is affine, negative coefficients can be realized by duplicating hidden units and keeping the corresponding output weights nonnegative. Thus, up to a constant-factor increase in width, the block class contains signed linear combinations of $\tanh$ units.

**Step 2 (uniform approximation of the target energy).** By the universal approximation theorem for single-hidden-layer networks with nonpolynomial activation (e.g., $\tanh$), for any $\delta > 0$ there exist a width $k$ and parameters $\theta$ such that

$$\sup_{(\mathbf{x}, \mathbf{y}) \in \mathcal{X} \times \mathcal{Y}} \big|\mathcal{B}(\mathbf{x}, \mathbf{y}; \theta) - \tau f(\mathbf{x}, \mathbf{y})\big| < \delta. \tag{30}$$

Define $g(\mathbf{x}, \mathbf{y}) := \mathcal{B}(\mathbf{x}, \mathbf{y}; \theta)/\tau$ and $\eta := \delta/\tau$. Then equation 30 is equivalent to

$$\sup_{(\mathbf{x}, \mathbf{y}) \in \mathcal{X} \times \mathcal{Y}} \big|g(\mathbf{x}, \mathbf{y}) - f(\mathbf{x}, \mathbf{y})\big| < \eta. \tag{31}$$

**Step 3 (uniform KL control).** For each $\mathbf{x} \in \mathcal{X}$, define

$$q(\mathbf{y} \mid \mathbf{x}) := \frac{\exp\big(g(\mathbf{x}, \mathbf{y})\big)}{\int_{\mathcal{Y}} \exp\big(g(\mathbf{x}, \mathbf{y}')\big) \, d\mathbf{y}'}. \tag{32}$$

The normalizer in equation 32 is finite because $g$ is continuous and $\mathcal{Y}$ is compact. Let $Z_g(\mathbf{x}) := \int_{\mathcal{Y}} \exp\big(g(\mathbf{x}, \mathbf{y}')\big) \, d\mathbf{y}'$. Since $p^\star(\cdot \mid \mathbf{x})$ is a density, without loss of generality we may normalize the energy so that $\int_{\mathcal{Y}} e^{f(\mathbf{x}, \mathbf{y}')} d\mathbf{y}' = 1$. From equation 31, for all $(\mathbf{x}, \mathbf{y})$,

$$e^{-\eta} \leq \frac{e^{g(\mathbf{x}, \mathbf{y})}}{e^{f(\mathbf{x}, \mathbf{y})}} \leq e^\eta.$$

Integrating over $\mathbf{y} \in \mathcal{Y}$ yields

$$e^{-\eta} \leq Z_g(\mathbf{x}) \leq e^\eta, \qquad \text{hence} \qquad |\log Z_g(\mathbf{x})| \leq \eta, \quad \forall \mathbf{x} \in \mathcal{X}. \tag{33}$$

Moreover,

$$\log \frac{p^\star(\mathbf{y} \mid \mathbf{x})}{q(\mathbf{y} \mid \mathbf{x})} = \log \frac{e^{f(\mathbf{x}, \mathbf{y})}}{e^{g(\mathbf{x}, \mathbf{y})}/Z_g(\mathbf{x})} = \big(f(\mathbf{x}, \mathbf{y}) - g(\mathbf{x}, \mathbf{y})\big) + \log Z_g(\mathbf{x}).$$

Taking expectation under $p^\star(\cdot \mid \mathbf{x})$ and using equation 31 and equation 33 gives

$$\mathrm{KL}\big(p^\star(\cdot \mid \mathbf{x}) \,\big\|\, q(\cdot \mid \mathbf{x})\big) = \mathbb{E}_{p^\star(\cdot \mid \mathbf{x})}[f(\mathbf{x}, \mathbf{Y}) - g(\mathbf{x}, \mathbf{Y})] + \log Z_g(\mathbf{x})$$
$$\leq \eta + \eta = 2\eta, \qquad \forall \mathbf{x} \in \mathcal{X}. \tag{34}$$

**Step 4 (choose $\delta$ and embed into BL).** Choose $\delta := \varepsilon\tau/4$, so that $\eta = \delta/\tau = \varepsilon/4$. Then equation 34 implies

$$\sup_{\mathbf{x} \in \mathcal{X}} \mathrm{KL}\big(p^\star(\cdot \mid \mathbf{x}) \,\big\|\, q(\cdot \mid \mathbf{x})\big) \leq 2\eta = \varepsilon/2 < \varepsilon.$$

Finally, the density $q(\cdot \mid \mathbf{x})$ equals the Gibbs distribution equation 26 with energy $\mathrm{BL}_{\theta^\star}(\mathbf{x}, \mathbf{y}) := \mathcal{B}(\mathbf{x}, \mathbf{y}; \theta)$ (a finite BL architecture containing a single block), and temperature $\tau$. This proves the claim. $\qquad \square$

### E.3 IDENTIFIABLE BEHAVIOR LEARNING (IBL)

#### E.3.1 SETUP AND ASSUMPTION

**Input–output space and data.** Let $\mathcal{X} \subset \mathbb{R}^{d_x}$ and $\mathcal{Y} \subset \mathbb{R}^{d_y}$ be compact sets. Assume the data distribution $P_{X,Y}$ is supported on $\mathcal{X} \times \mathcal{Y}$, and that there exists a point $z_0 = (x_0, y_0)$ in the interior of its support; that is, some open neighborhood of $z_0$ has positive $P_{X,Y}$-measure. All expectations are taken with respect to $P_{X,Y}$ unless otherwise specified.

**Parameter space and polynomial feature maps.** The parameter space factorizes as

$$\Theta := \Theta_U \times \Theta_C \times \Theta_T \times \mathcal{W}_\circ.$$

For $\theta_U \in \Theta_U, \theta_C \in \Theta_C$, and $\theta_T \in \Theta_T$, we define polynomial feature maps

$$p_u : \mathcal{X} \times \mathcal{Y} \to \mathbb{R}^{d_u}, \quad p_c : \mathcal{X} \times \mathcal{Y} \to \mathbb{R}^{d_c}, \quad p_t : \mathcal{X} \times \mathcal{Y} \to \mathbb{R}^{d_t},$$

each of fixed degree and injective in their coefficients (i.e., distinct coefficients yield distinct functions). For a single block, $\theta_U, \theta_C, \theta_T$ correspond to the parameters of the $U$, $C$, and $T$ terms together with their respective external multipliers (e.g., penalty weights $\lambda$). For a deep network composed of multiple blocks, $\theta = (\theta_U, \theta_C, \theta_T)$ denotes the collection of all block-level parameters across the hierarchy, where $\theta_U$ aggregates the parameters of all $U$-terms, $\theta_C$ those of all $C$-terms, and $\theta_T$ those of all $T$-terms (each including their associated multipliers).

The output component $\mathcal{W}_\circ$ corresponds to the affine transformation in the final layer: $\mathcal{W}_\circ = \mathbb{R}^{d'}$ for single-output prediction, and $\mathcal{W}_\circ = \mathbb{R}^{d' \times m}$ for $m$-way classification, where $d'$ is the output dimension induced by the preceding network, whether shallow or deep.

**Identifiable base block.** Let $\lambda_0 \in \mathbb{R}^{d_u}$, $\lambda_1 \in \mathbb{R}^{d_c}$, and $\lambda_2 \in \mathbb{R}^{d_t}$ denote nonnegative weight vectors, treated as learnable parameters. We instantiate the identifiable modular block

$$\mathcal{B}^{\mathrm{id}}(x, y; \theta) = \lambda_0^\top \tanh\big(p_u(x, y)\big) - \lambda_1^\top \mathrm{softplus}\big(p_c(x, y)\big) - \lambda_2^\top \big(p_t(x, y)\big)^{\odot 2}, \quad (35)$$

where $(\cdot)^{\odot 2}$ denotes elementwise squaring. By construction, the $\tanh$ and $\mathrm{softplus}$ heads are strictly monotone in their arguments, while the quadratic head is even.

We assume that each polynomial feature map $\mathbf{p}_\bullet(x, y)$ contains no nonzero monomial independent of $y$; that is, no feature is a pure function of $x$ or a constant. This ensures that $\mathcal{B}^{\mathrm{id}}(x, y)$ is nonconstant in $y$ unless all weights vanish.

**Architectures.** We implement IBL in three architectural forms, each producing a compositional utility function over $(x, y)$.

- IBL(Single): A single block is used as the compositional utility,

$$\mathrm{IBL}(x, y) := \mathcal{B}^{\mathrm{id}}(x, y).$$

- IBL(Shallow): Shallow IBL uses one or two stacked layers of parallel blocks. For instance, a first layer

$$\mathbb{B}_1^{\mathrm{id}}(x, y) := \big[\, \mathcal{B}_{1,1}^{\mathrm{id}}(x, y), \ldots, \mathcal{B}_{1,d_1}^{\mathrm{id}}(x, y) \,\big]^\top \in \mathbb{R}^{d_1}$$

feeds into a bias-free affine map

$$\mathrm{IBL}_{\mathrm{Shallow}}(x, y) := \mathbf{W}_1^\circ \mathbb{B}_1^{\mathrm{id}}(x, y),$$

where $\mathbf{W}_1^\circ \in \mathbb{R}^{m \times d_1}$ for classification and $\mathbf{W}_1^\circ \in \mathbb{R}^{1 \times d_1}$ for scalar output.

- IBL(Deep): Deep IBL extends the construction to depth $L > 2$, recursively defined as

$$\mathrm{IBL}(x, y) := \mathbf{W}_L^\circ \cdot \mathbb{B}_L^{\mathrm{id}}\big(\cdots \mathbb{B}_2^{\mathrm{id}}(\mathbb{B}_1^{\mathrm{id}}(x, y)) \cdots\big),$$

where each $\mathbb{B}_\ell^{\mathrm{id}}$ stacks parallel blocks $\mathcal{B}_{\ell,i}^{\mathrm{id}}(x, y)$, and $\mathbf{W}_L^\circ$ is a bias-free affine transformation. The cases $L = 1$ and $L = 2$ recover the Single and Shallow architectures, respectively.

**Induced conditional model.** Let $\text{IBL}(x, y)$ denote the compositional utility function produced by the chosen architecture (Single, Shallow, or Deep). It induces the conditional Gibbs distribution

$$(\text{Discrete } y \in [m]) \qquad p(y \mid x) = \text{softmax}_y\{\text{IBL}(x, y)\}, \tag{36}$$

$$(\text{Continuous } y) \qquad p(y \mid x) = \frac{\exp\{\text{IBL}(x, y)/\tau\}}{\int_{\mathcal{Y}} \exp\{\text{IBL}(x, \tilde{y})/\tau\} \, d\tilde{y}}, \quad \tau > 0 \text{ fixed.} \tag{37}$$

Here $\tau$ is a fixed temperature parameter. Thus, IBL predicts by defining a compositional utility landscape whose Gibbs distribution governs $y$ given $x$.

**Quotient parameter space.**

**Definition E.1** (Symmetry Quotient Space). *Define the equivalence relation $\sim$ on $\Theta$ as the smallest relation satisfying*

$$\theta_t \sim \theta'_t \quad \Longleftrightarrow \quad p_t^{(i)}(x, y; \theta_t^{(i)})^{\odot 2} = p_t^{(i)}(x, y; \theta_t'^{(i)})^{\odot 2} \quad \text{for all } i \text{ and } (x, y).$$

*The corresponding quotient space is*

$$\bar{\Theta} := \Theta/\sim .$$

*Explanation.* The $T$-component is designed to encode equality constraints, which are symbolically equations. Flipping the overall sign of such a constraint leaves the equation unchanged, so different parameterizations that differ only by sign should be regarded as equivalent.

**Definition E.2** (Scale-Invariant Quotient Space). *Define the equivalence relation $\approx$ on $\bar{\Theta}$ by*

$$\bar{\theta} \approx \bar{\theta}' \quad \Longleftrightarrow \quad \exists\, c > 0 \text{ such that } \mathsf{s}(x, y; \bar{\theta}) = c\, \mathsf{s}(x, y; \bar{\theta}').$$

*The scale-invariant quotient space is then given by*

$$\widetilde{\Theta} := \bar{\Theta}/\approx .$$

*Explanation.* In classification, predictions depend only on relative compositional utility differences between candidate labels. From a technical perspective, quotienting out global shifts or uniform scalings is necessary: without this identification, the cross-entropy loss admits redundant parameterizations that differ only by such transformations. At the same time, this quotient is natural and harmless, since it does not eliminate informative ratios between classes but merely discards absolute levels or scales that play no role in the softmax decision rule.

**Loss Functions.** We adopt a hybrid loss to simultaneously accommodate discrete and continuous outputs. Specifically, cross-entropy (CE) is applied to discrete targets, while denoising score matching (DSM) is applied to continuous targets. Let $\gamma_c, \gamma_d \geq 0$ with $\gamma_c + \gamma_d > 0$. The population risk, defined on the quotient parameter space, is given by

$$\mathcal{M}(\bar{\theta}) = \gamma_d \, \mathbb{E}[-\log p_\theta(Y \mid X)] + \gamma_c \, \mathbb{E}[\mathcal{S}_{\text{DSM}}(\theta; X)], \qquad \theta \in \pi^{-1}(\bar{\theta}), \tag{38}$$

where $\pi$ denotes the canonical projection from the original parameter space onto its quotient.

For continuous outputs $Y \in \mathcal{Y} \subseteq \mathbb{R}^{d_y}$, DSM is implemented by perturbing the target with additive Gaussian noise $\tilde{Y} = Y + \varepsilon$, $\varepsilon \sim \mathcal{N}(0, \sigma^2 I)$, and penalizing the squared discrepancy between the model score and the corresponding denoising score:

$$\mathcal{S}_{\text{DSM}}(\theta; X) = \frac{1}{2\sigma^2} \mathbb{E}_\varepsilon \left[ \left\| \nabla_{\tilde{y}} \log p_\theta(\tilde{y} \mid X) + \tfrac{1}{\sigma^2}(Y - \tilde{Y}) \right\|^2 \,\Big|\, X, Y \right]. \tag{39}$$

In classification-only settings we set $\gamma_c = 0$ (pure CE), while in regression-only settings we set $\gamma_d = 0$ (pure DSM).

For a single observation $Z = (X, Y)$, we define the *per-sample loss* as

$$\ell(\theta; Z) := \gamma_d \left[ -\log p_\theta(Y \mid X) \right] + \gamma_c \, \mathcal{S}_{\text{DSM}}(\theta; X). \tag{40}$$

The empirical criterion then takes the standard $M$-estimation form

$$\hat{Q}_n(\theta) = \frac{1}{n} \sum_{i=1}^n \ell(\theta; Z_i), \qquad Z_i = (X_i, Y_i). \tag{41}$$

**Key Assumptions.**

**Assumption E.1** (Global Atomic Independence and Injectivity). *Let $\bar{\Psi}$ be the atomic parameter quotient.*

1. *Injectivity on the quotient. The map $\bar{\Psi} \to \mathbb{R}^{\mathcal{X} \times \mathcal{Y}}$, $\bar{\psi} \mapsto g_{\bar{\psi}}$, is injective.*

2. *Linear Independence. Atomic linear independence. Any finite collection of pairwise distinct atoms $\{g_{\bar{\psi}_i}\}_{i=1}^{r}$ with $\bar{\psi}_i \in \bar{\Psi}$ is linearly independent in $\mathbb{R}^{\mathcal{X} \times \mathcal{Y}}$.*

3. *Minimality. In all model instances we only consider minimal representations: no duplicate atoms and its corresponding linear coefficient in the mixture is nonzero.*

4. *Canonical ordering. For each model instance, a fixed canonical ordering is imposed on the atom list.*

*Explanation.* Assumption E.1 treats each identifiable block $\mathcal{B}^{\mathrm{id}}$ as an *atomic* building unit and imposes four structural requirements on representations built from these atoms. Together, these four conditions define a non-ambiguous, non-redundant, and canonical algebra of atoms: after quotienting by the natural symmetries, every model constructed from $\mathcal{B}$-blocks admits a unique minimal representation (up to the prescribed equivalences). This structural regularity is the foundation on which identifiability statements are built: it guarantees that observing the model output (or the objective it optimizes) allows one, in principle, to recover the underlying atomic components and their coefficients in the appropriate quotient sense.

*Practical remark.* In practice, these conditions can be encouraged or approximately enforced in two complementary ways. First, the design of atomic classes (choice of polynomial bases, interaction terms, and activation heads) can be chosen so that injectivity and linear independence are more plausible by construction. Second, model selection and post-processing (e.g., pruning atoms with near-zero coefficients, enforcing a deterministic tie-breaking rule for ordering) can be applied after training to realize minimality and canonical ordering. These practical measures make the theoretical assumptions operationally meaningful in empirical applications.

### E.3.2 PROOF OF THEOREMS

**Lemma E.1** (Identifiability of Linear Combinations). *Let $Z$ be a set. For each $j = 1, \ldots, m$, let $\Phi_j$ be a parameter space and define atomic functions*

$$g_\psi := f(\cdot; \phi_j), \qquad \psi = (j, \phi_j) \in \Psi,$$

*where $\Psi := \bigsqcup_{j=1}^{m} \Phi_j$ is the disjoint union. Let $\bar{\Psi}$ be the quotient atomic parameter space, and denote its elements by $\bar{\psi} \in \bar{\Psi}$.*

*Define the quotient parameter space of the model as*

$$\bar{\Xi} := \prod_{j=1}^{m} \big( (\mathbb{R} \setminus \{0\}) \times \bar{\Psi} \big), \qquad \bar{\xi} = ((a_1, \bar{\psi}_1), \ldots, (a_m, \bar{\psi}_m)).$$

*The associated linear combination model is*

$$S_{\bar{\xi}} := \sum_{j=1}^{m} a_j g_{\bar{\psi}_j}.$$

*By virtue of Assumption E.1, the model is identifiable in the quotient parameter space $\bar{\Xi}$: if $S_{\bar{\xi}} \equiv S_{\bar{\xi}'}$ on $Z$, then $\bar{\xi} = \bar{\xi}'$.*

*Proof.* Suppose $S_{\bar{\xi}} \equiv S_{\bar{\xi}'}$ on $Z$, i.e.,

$$\sum_{j=1}^{m} a_j \, g_{(j, \phi_j)} - \sum_{j=1}^{m} a_j' \, g_{(j, \phi_j')} \equiv 0.$$

Let $\mathcal{U}$ be the set of *distinct* atoms in the quotient $\bar{\Psi}$ that appear on either side, and for each $\bar{\psi} \in \mathcal{U}$ let

$$\beta(\bar{\psi}) := \sum_{j:\,[j,\phi_j]=\bar{\psi}} a_j \;-\; \sum_{j':\,[j',\phi'_{j'}]=\bar{\psi}} a'_{j'}$$

be the net coefficient of $g_{\bar{\psi}}$. Then

$$\sum_{\bar{\psi}\in\mathcal{U}} \beta(\bar{\psi})\, g_{\bar{\psi}} \;\equiv\; 0.$$

By the *linear independence* condition (Assumption E.1:2) of pairwise distinct atoms in $\bar{\Psi}$, we must have $\beta(\bar{\psi}) = 0$ for all $\bar{\psi} \in \mathcal{U}$.

Furthermore, by the *Minimality* requirement (Assumption E.1:3), each $\bar{\psi}$ appears exactly once on each side and with nonzero coefficient. Thus the two sides must contain the exact same list of coefficient–atom pairs $\{(a_j, \bar{\psi}_j)\}_{j=1}^m$, and since a canonical ordering is imposed (Assumption E.1:4), it follows that

$$\bar{\xi} = \bar{\xi}'.$$

$\square$

**Theorem E.1** (Identifiability of IBL(Single)). *The IBL(Single) architecture uses the atom set*

$$\big\{\, \tanh(p_{u,i}),\; \mathrm{softplus}(p_{c,i}),\; (p_{t,i})^2 \,:\, i = 1, \dots, d_u;\; i = 1, \dots, d_c;\; i = 1, \dots, d_t \,\big\}.$$

*Under Assumption E.1, the model is identifiable in the quotient space $\bar{\Theta}$: if $\mathcal{B}_\theta^{\mathrm{id}} \equiv \mathcal{B}_{\theta'}^{\mathrm{id}}$ on $\mathcal{X} \times \mathcal{Y}$, then $\theta = \theta'$ in $\bar{\Theta}$.*

*Proof.* Write

$$\mathcal{B}_\theta^{\mathrm{id}} = \sum_{j=1}^m a_j\, f(\cdot; \phi_j), \qquad m := d_u + d_c + d_t,$$

where each $f(\cdot; \phi_j)$ is one of the atoms $\tanh(p_{u,i})$, $\mathrm{softplus}(p_{c,i})$, or $(p_{t,i})^2$, and $a_j$ is the corresponding entry in $(\lambda_0, \lambda_1, \lambda_2)$, with a fixed ordering over all indices.

If $\mathcal{B}_\theta^{\mathrm{id}} \equiv \mathcal{B}_{\theta'}^{\mathrm{id}}$ on $\mathcal{X} \times \mathcal{Y}$, then Lemma E.1 and Assumption E.1 imply that all atoms and coefficients must agree in the quotient atomic space $\bar{\Psi}$. Since the ordering is fixed, this implies $\theta = \theta'$ in $\bar{\Theta}$. $\square$

**Theorem E.2** (Identifiability of IBL(Shallow)). *The IBL(Shallow) architecture uses the atom set*

$$\big\{\, \mathcal{B}_{\theta_{1,j}}^{\mathrm{id}}(x, y) \,\big\}_{j=1}^{d_1},$$

*where each $\mathcal{B}_{\theta_{1,j}}^{\mathrm{id}} : \mathcal{X} \times \mathcal{Y} \to \mathbb{R}$ is a single-block IBL module parametrized by $\theta_{1,j} \in \Theta_1$. The full parameter is denoted*

$$\theta := \big((\theta_{1,1}, \dots, \theta_{1,d_1}),\, \mathbf{W}_1^\circ\big) \in \Theta := (\Theta_1)^{d_1} \times \mathbb{R}^{m \times d_1}.$$

*Under Assumption E.1, the mapping $\theta \mapsto \mathrm{IBL}_{Shallow}$ is identifiable in the quotient space $\bar{\Theta}$: if*

$$\mathrm{IBL}_{Shallow}(x, y; \theta) \equiv \mathrm{IBL}_{Shallow}(x, y; \theta') \quad \text{on } \mathcal{X} \times \mathcal{Y},$$

*then*

$$\theta = \theta' \quad \text{in } \bar{\Theta}.$$

*Proof.* Write the $k$-th output component as a linear combination of atoms:

$$s_\theta^{(k)}(x, y) = \sum_{j=1}^{d_1} w_j^{(k)}\, \mathcal{B}_{\theta_{1,j}}^{\mathrm{id}}(x, y), \qquad k = 1, \dots, m,$$

where $w_j^{(k)}$ denotes the $(k, j)$-th entry of $\mathbf{W}_1^\circ$.

Suppose two parameter tuples $(\mathbf{W}_1^\circ, \{\theta_{1,j}\}_{j=1}^{d_1})$ and $(\mathbf{W}_1^{\circ\,\prime}, \{\theta'_{1,j}\}_{j=1}^{d_1})$ yield identical vector scores on $\mathcal{X} \times \mathcal{Y}$. Then for each $k$, we have $s_\theta^{(k)} \equiv s_{\theta'}^{(k)}$ on $\mathcal{X} \times \mathcal{Y}$.

Fix any $k$. Under Assumption E.1, Lemma E.1 ensures that the coefficient–atom pairs $\{(w_j^{(k)}, \mathcal{B}^{\text{id}}_{\theta_{1,j}})\}_{j=1}^{d_1}$ are uniquely determined (up to equivalence in the quotient $\bar{\Theta}$). In particular, for each $j = 1, \ldots, d_1$, we must have

$$w_j^{(k)} = w_j'^{(k)}, \qquad \mathcal{B}^{\text{id}}_{\theta_{1,j}} \equiv \mathcal{B}^{\text{id}}_{\theta'_{1,j}}.$$

Because this holds for all $k = 1, \ldots, m$, it follows that $\mathbf{W}_1^\circ = \mathbf{W}_1^{\circ}{}'$ and $\theta_{1,j} = \theta'_{1,j}$ in the quotient parameter space for all $j$.

Thus $\theta = \theta'$ in $\bar{\Theta}$, establishing full identifiability under fixed ordering. $\square$

**Theorem E.3** (Identifiability of IBL(Deep)). *Fix integers $L > 2$ and widths $d_1, \ldots, d_{L-1}$. The IBL(Deep) architecture uses the final-layer atom set*

$$\left\{ \mathcal{B}^{\text{id}}_{\vartheta_{L,j}}(x,y) \right\}_{j=1}^{d_L} \subset \mathbb{R}^{\mathcal{X} \times \mathcal{Y}},$$

*where each $\mathcal{B}^{\text{id}}_{\vartheta_{L,j}} : \mathbb{R}^{d_{L-1}} \to \mathbb{R}$ is a scalar-valued block applied to the output of layer $L-1$. Only the first-layer blocks ($\ell = 1$) are IBL(Single) modules as in Theorem E.1. For architectures with skip connections, the final-layer atoms can be extended to include skipped features (e.g., from earlier layers), which are treated as elements of $\left\{ \mathcal{B}^{\text{id}}_{\vartheta_{L,j}}(x,y) \right\}_{j=1}^{d_L}$.*

*The full parameter is*

$$\theta := \left( \{\vartheta_{\ell,j}\}_{\ell=1,j=1}^{L,d_\ell}, \mathbf{W}_{out} \right) \in \Theta := \prod_{\ell=1}^{L} (\Theta_1)^{d_\ell} \times \mathbb{R}^{m \times d_L}.$$

*Under Assumption E.1, the mapping $\theta \mapsto \text{IBL}_{Deep}(x, y; \theta)$ is identifiable in the quotient space $\bar{\Theta}$.*

*Proof.* Under the given architecture, the IBL(Deep) model ultimately takes the form

$$s^{(k)}(x,y) = \sum_{j=1}^{d_L} w_j^{(k)} \mathcal{B}^{\text{id}}_{\vartheta_{L,j}}(x,y), \qquad k = 1, \ldots, m,$$

where each $\mathcal{B}^{\text{id}}_{\vartheta_{L,j}}$ is a scalar-valued function applied to the output of preceding layers. By treating the set $\{\mathcal{B}^{\text{id}}_{\vartheta_{L,j}}(x,y)\}_{j=1}^{d_L}$ as the atom set, we reduce the model to an IBL(Shallow) form:

$$\mathbf{s}(x,y) = \mathbf{W}_{\text{out}} \mathbf{B}_L(x,y).$$

Under Assumption E.1, Theorem E.2 applies, implying that the full parameter $\theta = (\{\vartheta_{\ell,j}\}_{\ell,j}, \mathbf{W}_{\text{out}})$ is identifiable in the quotient space $\bar{\Theta}$. $\square$

**Theorem 2.4** (Identifiability of IBL). *Under Assumption E.1, the architectures IBL(Single), IBL(Shallow), and IBL(Deep) are all identifiable in the quotient space $\bar{\Theta}$.*

*Proof.* Immediate from Theorems E.1, E.2, and E.3. $\square$

**Theorem 2.5** (Loss Identifiability of IBL). *Let $\text{IBL}_\theta(x,y)$ denote an IBL model, and consider the conditional Gibbs distribution*

$$p_\theta(y \mid x) = \frac{\exp\big(\text{IBL}_\theta(x,y)\big)}{\int_{\mathcal{Y}} \exp\big(\text{IBL}_\theta(x,y')\big)\, dy'}.$$

*Define the population risk on the symmetry quotient $\bar{\Theta}$ as in equation 38. Assume that the parameter space $\Theta$ is compact. Then, under Assumption E.1, the following holds:*

*(i) If $\gamma_c > 0$, the risk functional $\mathcal{M}$ admits a unique minimizer in $\bar{\Theta}$. Moreover,*

$$\mathcal{M}(\bar{\theta}_1) = \mathcal{M}(\bar{\theta}_2) \implies \bar{\theta}_1 = \bar{\theta}_2.$$

*(ii) If $\gamma_c = 0$, the risk functional $\mathcal{M}$ admits a unique minimizer in the scale-invariant quotient $\widetilde{\Theta}$. Moreover,*

$$\mathcal{M}(\widetilde{\theta}_1) = \mathcal{M}(\widetilde{\theta}_2) \implies \widetilde{\theta}_1 = \widetilde{\theta}_2.$$

*Proof.* Under Assumption E.1, the IBL architecture is identifiable modulo the symmetry group defined by $\bar{\Theta}$, as established in Theorem 2.4. Let $\theta^\bullet \in \arg\min_{\theta \in \Theta} \mathcal{M}(\theta)$ and set $p^\star(\cdot \mid x) := p_{\theta^\bullet}(\cdot \mid x)$. Since $\Theta$ is compact and the loss $\mathcal{M}$ is continuous, a global minimizer exists. We show that it is unique in the stated quotient.

*Case $\gamma_c > 0$.* At any minimizer we have both $p_\theta(\cdot \mid x) = p^\star(\cdot \mid x)$ and $\nabla_y \log p_\theta(\cdot \mid x) = \nabla_y \log p^\star(\cdot \mid x)$ a.e. Since

$$\nabla_y \log p_\theta(y \mid x) = \nabla_y \mathrm{IBL}_\theta(x, y) - \nabla_y \log Z_\theta(x) = \nabla_y \mathrm{IBL}_\theta(x, y),$$

(the partition function $Z_\theta(x)$ is $y$-independent), score equality yields $\nabla_y\big(\mathrm{IBL}_\theta - \mathrm{IBL}_{\theta^\bullet}\big)(y; x) = 0$ a.e. IBL contains no $y$-independent terms. Therefore,

$$\mathrm{IBL}_\theta(x, y) = \mathrm{IBL}_{\theta^\bullet}(x, y) \quad \text{a.e.}$$

By Theorem 2.4 (identifiability in $\bar{\Theta}$), the minimizer is unique in $\bar{\Theta}$; in particular,

$$\mathcal{M}(\bar{\theta}_1) = \mathcal{M}(\bar{\theta}_2) \implies \bar{\theta}_1 = \bar{\theta}_2.$$

*Case $\gamma_c = 0$.* Here, $\mathcal{M}$ reduces to the cross-entropy risk, which is minimized if and only if $p_\theta(\cdot \mid x) = p^\star(\cdot \mid x)$ almost everywhere. The cross-entropy loss depends on $\mathrm{IBL}_\theta(x, y)$ only through its relative values across $y$, and is invariant under additive shifts and positive rescalings of the compositional utility. Hence, the loss depends only on the equivalence class $\widetilde{\theta} \in \widetilde{\Theta}\}$. As a result,

$$\mathcal{M}(\widetilde{\theta}_1) = \mathcal{M}(\widetilde{\theta}_2) \implies \widetilde{\theta}_1 = \widetilde{\theta}_2.$$

i.e., the minimizer is unique in $\widetilde{\Theta}$.

Hence, the minimizer is unique in the stated quotient space. This completes the proof. $\qquad\square$

**Theorem E.4** (Uniform M-estimation consistency (Newey & McFadden, 1994, Theorem 2.1)). *Let $(\mathcal{A}, d)$ be a compact metric space, and let $\widehat{L}_n : \mathcal{A} \to \mathbb{R}$ be a sequence of random objective functions, with population objective $L : \mathcal{A} \to \mathbb{R}$ such that:*

1. *$L(\alpha)$ is uniquely minimized at $\alpha^\star \in \mathcal{A}$;*

2. *$\mathcal{A}$ is compact;*

3. *$L(\alpha)$ is continuous;*

4. *$\widehat{L}_n(\alpha) \xrightarrow{p} L(\alpha)$ uniformly in $\alpha \in \mathcal{A}$.*

*Then any sequence $\hat{\alpha}_n \in \arg\min_{\alpha \in \mathcal{A}} \widehat{L}_n(\alpha)$ satisfies $\hat{\alpha}_n \xrightarrow{p} \alpha^\star$.*

**Theorem E.5** (Consistency of IBL). *Let $\mathcal{M}$ be the population risk defined in equation 38, and let $\mathcal{M}_n$ denote its empirical analogue. Suppose:*

1. *$\{(X_i, Y_i)\}_{i=1}^n$ are i.i.d. samples;*

2. *$\Theta$ is compact;*

3. *$\theta \mapsto \mathcal{M}(\theta)$ is continuous, and the loss class admits an integrable envelope such that*

$$\sup_{\theta \in \Theta} \big|\mathcal{M}_n(\theta) - \mathcal{M}(\theta)\big| \xrightarrow{p} 0;$$

*Let $\Xi$ denote the relevant quotient space ($\bar{\Theta}$ if $\gamma_c > 0$, $\widetilde{\Theta}$ if $\gamma_c = 0$), and let $\hat{\theta}_n \in \arg\min_{\theta \in \Theta} \mathcal{M}_n(\theta)$ and $\theta^\bullet \in \arg\min_{\theta \in \Theta} \mathcal{M}(\theta)$. Then*

$$\hat{\theta}_n \xrightarrow{p} \theta^\bullet \quad \text{in } \Xi, \qquad \mathcal{M}(\hat{\theta}_n) \xrightarrow{p} \mathcal{M}(\theta^\bullet).$$

*If the model is correctly specified (the data law is realized by some $\theta^\star \in \Theta$), then $\theta^\bullet = \theta^\star$ in $\Xi$, so $\hat{\theta}_n \xrightarrow{p} \theta^\star$.*

*Proof.* Let $\Xi$ denote the relevant quotient space: $\Xi = \bar{\Theta}$ if $\gamma_c > 0$ and $\Xi = \widetilde{\Theta}$ if $\gamma_c = 0$. Let $\pi : \Theta \to \Xi$ be the canonical quotient map. Since $\Theta$ is compact and $\pi$ is continuous and onto, $\Xi$ is compact. By assumption, $\mathcal{M}$ and $\mathcal{M}_n$ are invariant under the corresponding symmetry, hence they factor through $\pi$:

$$\widetilde{\mathcal{M}}(\xi) := \mathcal{M}(\theta), \qquad \widetilde{\mathcal{M}}_n(\xi) := \mathcal{M}_n(\theta) \quad (\text{any } \theta \in \pi^{-1}(\xi)).$$

These are well-defined and continuous on $\Xi$ because $\mathcal{M}$ is continuous on $\Theta$. Moreover,

$$\sup_{\xi \in \Xi} \left| \widetilde{\mathcal{M}}_n(\xi) - \widetilde{\mathcal{M}}(\xi) \right| \leq \sup_{\theta \in \Theta} \left| \mathcal{M}_n(\theta) - \mathcal{M}(\theta) \right| \xrightarrow{p} 0,$$

so uniform convergence in probability holds on $\Xi$.

By Loss Identifiability of IBL (Theorem 2.5), $\widetilde{\mathcal{M}}$ has a unique minimizer $\xi^\bullet \in \Xi$. Let $\hat{\xi}_n \in \arg\min_{\xi \in \Xi} \widetilde{\mathcal{M}}_n(\xi)$ (equivalently, choose $\hat{\theta}_n \in \arg\min_{\theta \in \Theta} \mathcal{M}_n(\theta)$ and set $\hat{\xi}_n = \pi(\hat{\theta}_n)$). Then the conditions of Theorem E.4 hold on the compact metric space $(\Xi, d)$, whence

$$\hat{\xi}_n \xrightarrow{p} \xi^\bullet.$$

Since $\widetilde{\mathcal{M}}$ is continuous on $\Xi$ and $\widetilde{\mathcal{M}}(\hat{\xi}_n) = \mathcal{M}(\hat{\theta}_n)$, $\widetilde{\mathcal{M}}(\xi^\bullet) = \mathcal{M}(\theta^\bullet)$ for any representative $\theta^\bullet \in \pi^{-1}(\xi^\bullet)$, we also obtain

$$\mathcal{M}(\hat{\theta}_n) = \widetilde{\mathcal{M}}(\hat{\xi}_n) \xrightarrow{p} \widetilde{\mathcal{M}}(\xi^\bullet) = \mathcal{M}(\theta^\bullet).$$

If the model is correctly specified (there exists $\theta^\star \in \Theta$ inducing the data law), the strict propriety of the CE/DSM terms implies that the unique minimizer in the quotient is the class of $\theta^\star$; hence $\hat{\theta}_n$ converges in probability to $\theta^\star$ in the corresponding quotient space. $\square$

**Theorem E.6** (Universal Approximation of IBL). *Let $\mathcal{X} \subset \mathbb{R}^d$ and $\mathcal{Y} \subset \mathbb{R}^m$ be compact sets, and let $p^\star(y \mid x)$ be any continuous conditional density such that $p^\star(y \mid x) > 0$ for all $(x, y) \in \mathcal{X} \times \mathcal{Y}$. Then for any $\tau > 0$ and $\varepsilon > 0$, there exists a finite IBL architecture (with some depth and width depending on $\varepsilon$) and a parameter $\theta^\star$ such that the Gibbs distribution*

$$p_\tau(y \mid x; \theta^\star) = \frac{\exp\big(IBL_{\theta^\star}(x, y)/\tau\big)}{\int_{\mathcal{Y}} \exp\big(IBL_{\theta^\star}(x, y')/\tau\big) \mathrm{d}y'} \tag{42}$$

*satisfies*

$$\sup_{x \in \mathcal{X}} \mathrm{KL}\big(p^\star(\cdot \mid x) \,\|\, p_\tau(\cdot \mid x; \theta^\star)\big) < \varepsilon. \tag{43}$$

*Proof.* The argument follows the same construction as in the proof of Theorem 2.3, with only notational modifications due to the IBL parameterization. For brevity, the details are omitted. $\square$

**Lemma E.2** (Sieve Approximation Lemma). *Let $\mathcal{C} : \Theta \to [0, \infty)$ be a complexity measure on the parameter space, and let $(c_n)_{n \geq 1}$ be a nondecreasing sequence with $c_n \uparrow \infty$. Define the sieve*

$$\Theta_n := \{\theta \in \Theta : \mathcal{C}(\theta) \leq c_n\},$$

*and for a fixed data-generating distribution $p^\dagger$, set*

$$\delta_n(p^\dagger) := \inf_{\theta \in \Theta_n} \sup_{x \in \mathcal{X}} \mathrm{KL}\big(p^\dagger(\cdot \mid x) \,\|\, p_\theta(\cdot \mid x)\big).$$

*Then the following are equivalent:*

*1. Sieve universal approximation: For every $\varepsilon > 0$ there exists a constant $C_\varepsilon < \infty$ such that*

$$\inf_{\theta : \mathcal{C}(\theta) \leq C_\varepsilon} \sup_{x \in \mathcal{X}} \mathrm{KL}\big(p^\dagger(\cdot \mid x) \,\|\, p_\theta(\cdot \mid x)\big) < \varepsilon.$$

*2. Vanishing approximation error: $\delta_n(p^\dagger) \downarrow 0$ as $n \to \infty$.*

*Moreover, if each $\Theta_n$ is compact and $\theta \mapsto \sup_x \mathrm{KL}(p^\dagger \| p_\theta)$ is continuous on $\Theta_n$, then the infimum in $\delta_n(p^\dagger)$ is attained for every $n$.*

*Proof.* *(1)* $\Rightarrow$ *(2)*. Fix $\varepsilon > 0$ and let $C_\varepsilon(p^\dagger)$ be as in (i). Since $c_n \uparrow \infty$, there exists $N$ such that $c_n \geq C_\varepsilon(p^\dagger)$ for all $n \geq N$. Hence $\Theta_n \supseteq \{\theta : \mathcal{C}(\theta) \leq C_\varepsilon(p^\dagger)\}$ for all $n \geq N$, and therefore

$$\delta_n(p^\dagger) \;=\; \inf_{\theta \in \Theta_n} \sup_x \mathrm{KL}(p^\dagger \| p_\theta) \;\leq\; \inf_{\theta : \, \mathcal{C}(\theta) \leq C_\varepsilon(p^\dagger)} \sup_x \mathrm{KL}(p^\dagger \| p_\theta) \;<\; \varepsilon,$$

for all $n \geq N$. Since $(\delta_n)$ is nonincreasing in $n$ (because $\Theta_n \uparrow$), it follows that $\delta_n(p^\dagger) \downarrow 0$.

*(2)* $\Rightarrow$ *(1)*. Fix $\varepsilon > 0$. By (ii) choose $N$ such that $\delta_N(p^\dagger) < \varepsilon$. Set $C_\varepsilon(p^\dagger) := c_N$. Then

$$\inf_{\theta : \, \mathcal{C}(\theta) \leq C_\varepsilon(p^\dagger)} \sup_x \mathrm{KL}(p^\dagger \| p_\theta) \;\leq\; \inf_{\theta \in \Theta_N} \sup_x \mathrm{KL}(p^\dagger \| p_\theta) \;=\; \delta_N(p^\dagger) \;<\; \varepsilon,$$

which is (i).

The attainment statement follows immediately from compactness of $\Theta_n$ and continuity of $\theta \mapsto \sup_x \mathrm{KL}(p^\dagger \| p_\theta)$ on $\Theta_n$. $\qquad\square$

**Theorem E.7** (Universal Consistency of IBL). *Consider a parameter space $\Theta$ for a class of IBL models, and let $\mathcal{C} : \Theta \to [0, \infty)$ be a lower semi-continuous complexity measure (e.g., network depth, width, or parameter norm). Let $(c_n)_{n \geq 1}$ be a nondecreasing sequence with $c_n \uparrow \infty$, and define the sieve*

$$\Theta_n := \{\theta \in \Theta : \mathcal{C}(\theta) \leq c_n\}.$$

*Assume:*

1. *The map $\theta \mapsto \sup_x \mathrm{KL}(p^\dagger \| p_\theta)$ is continuous on each compact $\Theta_n$.*

2. *The sequence of empirical minimizers $\{\hat{\theta}_n\}$ is relatively compact in $\bigcup_n \Theta_n$, as ensured by the uniform LLN together with compactness and continuity.*

*Then for any admissible data-generating distribution $p^\dagger$ satisfying the regularity assumptions of Theorem E.6, the IBL posterior sequence $\{p_{\hat{\theta}_n}\}$ satisfies*

$$\sup_{x \in \mathcal{X}} \mathrm{KL}\big(p^\dagger(\cdot \mid x) \,\|\, p_{\hat{\theta}_n}(\cdot \mid x)\big) \xrightarrow{p} 0,$$

*i.e. $\{p_{\hat{\theta}_n}\}$ converges to $p^\dagger$ uniformly in $x$ (in KL).*

*Proof.* Fix an admissible data law $p^\dagger$ (satisfying the regularity of Theorem E.6). For $\theta \in \bigcup_n \Theta_n$ define

$$F(\theta) \;:=\; \sup_{x \in \mathcal{X}} \mathrm{KL}\big(p^\dagger(\cdot \mid x) \,\|\, p_\theta(\cdot \mid x)\big), \qquad \delta_n \;:=\; \inf_{\theta \in \Theta_n} F(\theta).$$

Then Theorem E.6 and Lemma E.2 together imply that $\delta_n \downarrow 0$. By assumption 1, $F$ is continuous on each compact $\Theta_n$.

Let $\hat{\theta}_n \in \arg\min_{\theta \in \Theta_n} \mathcal{M}_n(\theta)$ be any sequence of ERM solutions. We show $F(\hat{\theta}_n) \xrightarrow{p} 0$.

*Step 1 (subsequence reduction and precompactness).* Take an arbitrary subsequence $(\hat{\theta}_{n_k})_k$. By assumption 2 there exists a further subsequence, still denoted $(\hat{\theta}_{n_k})_k$, and a (possibly $k$-dependent) index set $N_k \leq n_k$ with a parameter limit $\theta_\infty \in \Theta_N$ (for some finite $N$) such that $\hat{\theta}_{n_k} \to \theta_\infty$ in probability. Passing to a further subsequence if needed, we may assume $N_k \equiv N$.

*Step 2 (risk domination against $\Theta_N$-approximants).* For each $k$ pick $\theta_k \in \Theta_N$ with $F(\theta_k) \leq \delta_N + 1/k$ (attainment follows from compactness and continuity of $F$ on $\Theta_N$). By the ERM property and uniform LLN on $\Theta_N$,

$$\mathcal{M}(\hat{\theta}_{n_k}) \;\leq\; \mathcal{M}(\theta_k) + o_p(1) \qquad (k \to \infty).$$

Assume (w.l.o.g.) the CE component is present with a positive weight, so that the population risk decomposes as

$$\mathcal{M}(\theta) \;=\; \mathrm{const} + \gamma_d \, \mathbb{E}_X \big[\mathrm{KL}\big(p^\dagger(\cdot \mid X) \,\|\, p_\theta(\cdot \mid X)\big)\big] \;+\; \gamma_c \, \mathcal{L}^{\mathrm{DSM}}(\theta),$$

with $\gamma_d > 0$ (the DSM-only case is handled analogously by replacing KL with Fisher divergence). Using $\mathbb{E}_X[\mathrm{KL}(\cdot\|\cdot)] \leq F(\cdot)$, we obtain

$$\limsup_{k\to\infty} \mathbb{E}_X\Big[\mathrm{KL}\big(p^\dagger(\cdot \mid X) \,\|\, p_{\hat{\theta}_{n_k}}(\cdot \mid X)\big)\Big] \leq \limsup_{k\to\infty} F(\theta_k) \leq \delta_N.$$

Hence, along the subsequence,

$$\mathbb{E}_X\Big[\mathrm{KL}\big(p^\dagger(\cdot \mid X) \,\|\, p_{\hat{\theta}_{n_k}}(\cdot \mid X)\big)\Big] \xrightarrow{p} 0.$$

*Step 3 (identification of the subsequential limit).* By continuity of the model map $\theta \mapsto p_\theta(\cdot \mid x)$ (from Theorem E.6 regularity) and bounded convergence,

$$\mathbb{E}_X\big[\mathrm{KL}\big(p^\dagger(\cdot \mid X) \,\|\, p_{\theta_\infty}(\cdot \mid X)\big)\big] = 0.$$

Thus $f(x) := \mathrm{KL}\big(p^\dagger(\cdot \mid x) \,\|\, p_{\theta_\infty}(\cdot \mid x)\big)$ equals 0 for $P_X$-a.e. $x$. Since $f$ is continuous on compact $\mathcal{X}$ (by the same regularity) and $P_X$ has full support (admissible law), we conclude $f(x) \equiv 0$ on $\mathcal{X}$, i.e.

$$F(\theta_\infty) = \sup_{x\in\mathcal{X}} f(x) = 0.$$

*Step 4 (conclude $F(\hat{\theta}_{n_k}) \to 0$ in probability, hence $F(\hat{\theta}_n) \to 0$ in probability).* By assumption 1 continuity of $F$ on $\Theta_N$ and $\hat{\theta}_{n_k} \to \theta_\infty$ in probability, we have $F(\hat{\theta}_{n_k}) \xrightarrow{p} F(\theta_\infty) = 0$. Since the original subsequence was arbitrary and every subsequence admits a further subsequence with $F(\hat{\theta}_{n_k}) \xrightarrow{p} 0$, the full sequence satisfies $F(\hat{\theta}_n) \xrightarrow{p} 0$.

Therefore,

$$\sup_{x\in\mathcal{X}} \mathrm{KL}\big(p^\dagger(\cdot \mid x) \,\|\, p_{\hat{\theta}_n}(\cdot \mid x)\big) \xrightarrow{p} 0,$$

i.e. $p_{\hat{\theta}_n}(\cdot \mid x) \to p^\dagger(\cdot \mid x)$ uniformly in $x$ in KL. $\qquad\square$

# F    EXPERIMENTAL DETAILS

## F.1    HARDWARE

Most experiments are conducted on a single NVIDIA L40S GPU. A small number of runs are performed on a laptop equipped with an NVIDIA GeForce RTX 2050 GPU and an Intel Core i7–12700H CPU.

## F.2    STANDARD PREDICTION TASKS

**Datasets.**    In the Standard Prediction Task, we use 10 OpenML datasets across diverse application domains. Details are given in Table 2.

**Baseline Models.**    For comparison, we include the following baselines: MLP, Neural Additive Model (NAM) (Agarwal et al., 2020; Kayid et al., 2020), ElasticNet, Random Forest, Stochastic Variational Gaussian Process (SVGP) (Gardner et al., 2018), Logistic Regression, Decision Tree, TabNet (Arik & Pfister, 2021), Polynomial Logistic Regression, and LightGBM (Ke et al., 2017).

**Data preprocessing.**    For all ten datasets, we apply a consistent preprocessing strategy. Ordinal categorical variables are mapped to integer levels to preserve their inherent order. Nominal categorical variables without natural ordering are transformed using one-hot encoding. Continuous variables are standardized to zero mean and unit variance. Each dataset is randomly partitioned into train/validation/test splits with a 7:1:2 ratio.

**Hyperparameter Tuning Protocol.**    We perform hyperparameter optimization for most models using the TPE sampler from the Optuna package (Akiba et al., 2019), with 50 trials per dataset. For each model and dataset, the tuned configuration is evaluated under 8 random seeds.

Table 2: Standard OpenML datasets used in our task. #Features denotes the number of input variables (excluding the target and ID).

| Name | Size | #Features | Task type | Field |
|---|---|---|---|---|
| German Credit | 1,000 | 20 | Binary cls. | Finance |
| Adult Income | 48,842 | 14 | Binary cls. | Economics |
| COMPAS (two-years) | 5,278 | 13 | Binary cls. | Law & Society |
| Bank Marketing | 45,211 | 16 | Binary cls. | Marketing |
| Planning Relax | 182 | 12 | Binary cls. | Psychology |
| EEG Eye State | 14,980 | 14 | Binary cls. | Neuroscience |
| MAGIC Gamma Telescope | 19,020 | 10 | Binary cls. | Physics |
| Electricity | 45,312 | 8 | Binary cls. | Electrical Engineering |
| Wine Quality (Red) | 1,599 | 11 | Multiclass | Chemistry |
| Steel Plates Faults | 1,941 | 27 | Multiclass | Industrial Engineering |

Table 3: Overview of baseline models in the standard prediction task

| Methodological Family | Model Name |
|---|---|
| Neural networks | Standard MLP
Neural Additive Model (NAM)
TabNet |
| Linear regressors | ElasticNet
Logistic Regression
Polynomial Logistic Regression |
| Tree-based models | Random Forest
Decision Tree |
| Gradient boosting methods | LightGBM |
| Bayesian methods | Stochastic Variational Gaussian Process (SVGP) |

**BL Model Hyperparameter Space.** For BL(Single) and BL(Shallow), we optimize cross-entropy loss for classification. Both Adam (Kingma, 2014) and AdamW (Loshchilov & Hutter, 2017) optimizers are considered, and the better-performing variant is reported for each dataset. No data augmentation is applied. Batch sizes are chosen in a dataset-specific manner.

- BL(Single): A unified setting is reported across all experiments:

$$\text{degree}_U = [2], \quad \text{degree}_C = [2, 2, 2], \quad \text{degree}_T = [2, 2],$$

$$\sigma_{\text{params}} = 0.01, \quad \sigma_{\lambda_0} = 0.01, \quad \sigma_{\lambda_1} = 0.01, \quad \sigma_{\lambda_2} = 0.01.$$

Here, $\text{degree}_U$, $\text{degree}_C$, and $\text{degree}_T$ denote the polynomial degrees of the blocks that parameterize $U(x, y)$, $C(x, y)$, and $T(x, y)$, respectively. Lists indicate both the number of blocks and each block's degree: $\text{degree}_U = [2]$ means a single quadratic block for $U$, $\text{degree}_C = [2, 2, 2]$ means three quadratic constraint blocks, and $\text{degree}_T = [2, 2]$ means two quadratic belief blocks. $\sigma_{\text{params}}$ initializes coefficients of all polynomial blocks, while $\sigma_{\lambda_0}$, $\sigma_{\lambda_1}$, and $\sigma_{\lambda_2}$ initialize the UMP weights $(\lambda_0, \lambda_1, \lambda_2)$. The search grid is reported in Table 4.

- BL(Shallow): We use global gradient clipping of 1.0 and an early stopping patience of 20 epochs without validation improvement. Shallow architectures with depth $L \leq 3$ are considered. The search grid is reported in Table 5.

**Baseline Model Hyperparameter Spaces.** For baseline models, we also consider both Adam and AdamW for the neural network–based variants, and report results with the better-performing optimizer on each dataset. Batch sizes are tuned separately for each dataset. The detailed hyperparameter search spaces are summarized in Table 6.

Table 4: Hyperparameter tuning space for BL(Single)

| Model | Parameter | Search space |
|---|---|---|
| BL(Single) | `learning_rate` | $\{1e{-}3, 1e{-}1\}$ |
| | `batch_size` | $\{64, 128, 256, 512\}$ |
| | `max_grad_norm` | $\{1.0, 2.0, 5.0\}$ |

Table 5: Hyperparameter tuning space for BL(Shallow)

| Model | Parameter | Search space |
|---|---|---|
| BL(Shallow) | `learning_rate` | LogUniform$\{5e{-}5, 5e{-}3\}$ |
| | `batch_size` | $\{64, 128, 256, 512\}$ |
| | `n_layers` | UniformInt$\{1, 3\}$ |
| | `n_first_layer` | $\{24, 30, 36, 40\}$ |
| | `n_middle_layer` | $\{8, 6, 4\}$ |
| | `n_last_layer` | $\{2, 4, 6\}$ |
| | `weight_decay` | LogUniform$\{1e{-}4, 1e{-}1\}$ |

Table 6: The hyperparameter tuning space for baseline models used in the standard prediction tasks

| Model | Parameter | Search space |
|---|---|---|
| MLP | `learning_rate` | LogUniform$\{1e{-}5, 1e{-}1\}$ |
| | `batch_size` | $\{32, 64, 128, 256\}$ |
| | `n_layers` | UniformInt$\{2, 4\}$ |
| | `hidden_size` | UniformInt$\{32, 256\}$ |
| | `weight_decay` | LogUniform$\{1e{-}6, 1e{-}2\}$ |
| NAM | `learning_rate` | LogUniform$\{1e{-}3, 1e{-}1\}$ |
| | `batch_size` | $\{128, 256, 512, 1024\}$ |
| | `patience` | UniformInt$\{10, 30\}$ |

| Model | Parameter | Search space |
|---|---|---|
| ElasticNet (SGD) | alpha | LogUniform$\{1e-4, 1e+2\}$ |
| | l1_ratio | Uniform$\{0.0, 1.0\}$ |
| | max_iter | UniformInt$\{100, 2000\}$ |
| | tol | LogUniform$\{1e-6, 1e-2\}$ |
| | fit_intercept | $\{$true, false$\}$ |
| | learning_rate | $\{$optimal, constant, invscaling, adaptive$\}$ |
| | eta0 | LogUniform$\{1e-4, 1e-1\}$ |
| | validation_fraction | Uniform$\{0.05, 0.30\}$ |
| | n_iter_no_change | UniformInt$\{3, 20\}$ |
| PolyLogistic | degree | $\{2, 3\}$ |
| | penalty | $\{\ell_2, \ell_1, $"elasticnet"$\}$ |
| | C | LogUniform$\{1e-3, 1e+2\}$ |
| | l1_ratio | Uniform$\{0.1, 0.9\}$ |
| | solver | $\{$"liblinear", "lbfgs", "newton-cg", "saga"$\}$ |
| | max_iter | UniformInt$\{500, 2000\}$ |
| | tol | LogUniform$\{1e-5, 1e-3\}$ |
| Logistic (ElasticNet) | C | LogUniform$\{1e-3, 1e+2\}$ |
| | l1_ratio | Uniform$\{0.0, 1.0\}$ |
| | max_iter | UniformInt$\{100, 2000\}$ |
| | tol | LogUniform$\{1e-6, 1e-2\}$ |
| | fit_intercept | $\{$true, false$\}$ |
| LogisticRegression | solver | $\{$"liblinear", "lbfgs", "sag"$\}$ |
| | C | LogUniform$\{1e-4, 1e+2\}$ |
| | max_iter | UniformInt$\{100, 2000\}$ |
| | tol | LogUniform$\{1e-6, 1e-2\}$ |
| | fit_intercept | $\{$True, False$\}$ |
| | intercept_scaling | Uniform$\{0.1, 10.0\}$ |
| TabNet | learning_rate | LogUniform$\{1e-4, 3e-2\}$ |
| | batch_size | $\{128, 256, 512, 1024\}$ |
| | virtual_batch_size | $\{64, 128\}$ |
| | n_d=n_a | UniformInt$\{16, 64\}$ |
| | n_steps | UniformInt$\{3, 7\}$ |
| | gamma | Uniform$\{1.2, 1.7\}$ |
| | lambda_sparse | LogUniform$\{1e-6, 1e-3\}$ |
| DecisionTree | criterion | $\{$"gini", "entropy", "log_loss"$\}$ |
| | max_depth | UniformInt$\{3, 20\}$ |
| | min_samples_split | UniformInt$\{2, 20\}$ |
| | min_samples_leaf | UniformInt$\{1, 10\}$ |
| | min_weight_fraction_leaf | Uniform$\{0.0, 0.5\}$ |
| | max_features | $\{$"sqrt", "log2"$\}$ |
| | max_leaf_nodes | UniformInt$\{10, 1000\}$ |
| | min_impurity_decrease | Uniform$\{0.0, 0.1\}$ |
| | ccp_alpha | Uniform$\{0.0, 0.1\}$ |
| GP (SVGP) | kernel | $\{$rbf, matern, rational_quadratic$\}$ |
| | lengthscale | LogUniform$\{0.1, 10.0\}$ |
| | rq_alpha | LogUniform$\{0.1, 5.0\}$ |
| | num_inducing | UniformInt$\{100, 500\}$ |
| | learning_rate | LogUniform$\{1e-2, 5e-1\}$ |
| | training_iters | UniformInt$\{50, 200\}$ |
| RandomForest | n_estimators | UniformInt$\{100, 500\}$ |
| | max_depth | UniformInt$\{3, 30\}$ |
| | max_features | $\{$"sqrt", "log2"$\}$ |
| | min_samples_leaf | UniformInt$\{1, 10\}$ |

| Model | Parameter | Search space |
|---|---|---|
| | min_samples_split | UniformInt{2, 20} |

### F.3 INTERPRETING BL: A CASE STUDY

#### F.3.1 INTERPRETING BL(DEEP): HIGH-LEVEL OVERVIEW

Deeper variants of BL are constructed by stacking multiple BL(Single) modules into hierarchical layers, followed by a final affine transformation. This forms a system of interacting UMPs (each of which can be viewed as an agent), where each internal block $\mathcal{B}$ represents a single interpretable UMP. As shown in Figure 7, first-layer modules correspond to individual UMPs, while the second-layer module performs optimal coordination by aggregating or allocating their outputs. This layered structure offers a compositional interpretation of deeper BL models as systems of interacting, interpretable UMPs.

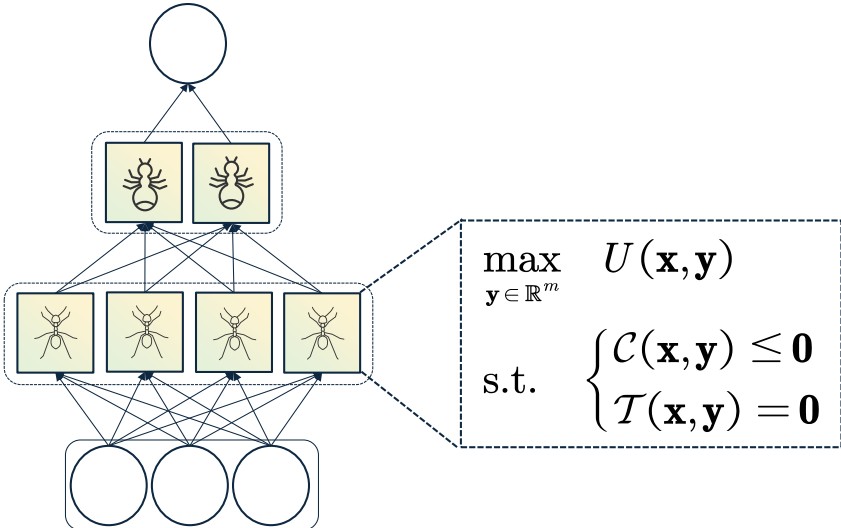

Figure 7: Interpreting deeper BL architectures as hierarchical systems of interacting agents. Each block $\mathcal{B}$ represents an interpretable agent solving its own UMP, while a layer corresponds to a set of heterogeneous agents operating in parallel. The next layer then aggregates and reallocates the negative energies from the previous layer, thereby performing higher-level coordination across agents. This layered organization provides a natural compositional interpretation of deep BL: bottom-layer modules encode local objectives, while upper layers synthesize these into collective outcomes. Analogous structures arise in biological and social systems—for example, in ant colonies, individual ants (first-layer agents) follow simple local rules, yet their collective behavior is coordinated through higher-level interactions (second-layer aggregation), yielding globally efficient resource allocation and task division.

#### F.3.2 CASE STUDY: ADDITIONAL DETAILS

Table 7: Boston Housing dataset variables and descriptions.

| Variable | Description |
|----------|-------------|
| CRIM | Per-capita crime rate by town |
| ZN | Proportion of residential land zoned for lots over 25,000 sq.ft. |
| INDUS | Proportion of non-retail business acres per town |
| CHAS | Charles River dummy variable (=1 if tract bounds river) |
| NOX | Nitric oxide concentration (parts per 10 million) |
| RM | Average number of rooms per dwelling |
| AGE | Proportion of owner-occupied units built prior to 1940 |
| DIS | Weighted distances to five Boston employment centers |
| RAD | Index of accessibility to radial highways |
| TAX | Full-value property-tax rate per \$10,000 |
| PTRATIO | Pupil–teacher ratio by town |
| B | $1000(B_k - 0.63)^2$ where $B_k$ is the proportion of Black residents by town |
| LSTAT | Percentage of lower-status population |
| MEDV | Median value of owner-occupied homes in \$1000s |

Table 8: Semantic roles of blocks in the deep BL architecture.

| Layer | Block | Representative preference |
|-------|-------|---------------------------|
| Layer 1 | Location-Sensitive Buyer | Values river access, transport accessibility, and neighborhood amenities. |
| | Risk-Sensitive Buyer | Averse to local disamenities such as pollution and environmental risk. |
| | Economic-Sensitive Buyer | Sensitive to school quality and neighborhood socio-economic composition. |
| | Zoning-Contrast Buyer | Responds to zoning and land-use patterns that shape local housing supply. |
| | Affordability-Preferring Buyer | Strongly prefers more affordable housing and dislikes high prices. |
| Layer 2 | Integrated Location–Economic Buyer | Jointly evaluates location and socio-economic attributes in an integrated way. |
| | Budget-Conflict Buyer | Exhibits strong preferences for desirable locations but faces binding budget constraints. |
| | Balanced Trade-off Buyer | Jointly considers multiple housing attributes in a balanced manner. |
| Layer 3 | Representative Composite Buyer | Aggregates all lower-level preference components into a representative household. |

Table 9: Each block in the deep BL architecture is aligned with a classic preference mechanism documented in the economics literature.

| Layer / Block | Representative reference |
|---------------|-------------------------|
| Layer 1: Location-Sensitive Buyer | Gibbons & Machin (2005) |
| Layer 1: Risk-Sensitive Buyer | Chay & Greenstone (2005) |
| Layer 1: Economic-Sensitive Buyer | Black (1999) |
| Layer 1: Zoning-Contrast Buyer | Glaeser & Gyourko (2002) |
| Layer 1: Affordability-Preferring Buyer | McFadden (1977) |
| Layer 2: Integrated Location–Economic Buyer | Bayer et al. (2007) |
| Layer 2: Budget-Conflict Buyer | Balseiro et al. (2019) |
| Layer 2: Balanced Trade-off Buyer | Rosen (1974) |

## F.4 Prediction on High-Dimensional Inputs

**Datasets Description and Preprocessing.** For image datasets, we use the official train/test splits of MNIST and Fashion-MNIST: Inputs are converted to single-channel images scaled to $[0, 1]$ and standardized with dataset-specific statistics. No resizing or data augmentation is applied. Training uses shuffled mini-batches of size $64$. For text datasets, we apply the following procedures:

1 Data sources and official splits. We use the official training and test splits for AG News and Yelp Review Polarity without any custom re-partitioning. Both datasets are class-balanced across labels, and we do not perform any resampling.

2 Dataset sizes. AG News: 120,000 training / 7,600 test samples with four balanced classes. Yelp Review Polarity: 560,000 training / 38,000 test samples with two balanced classes.

3 Label mapping. AG News: labels 1–4 are mapped to 0–3. Yelp Review Polarity: labels 1–2 are mapped to 0–1.

4 Text preprocessing and feature representation. All texts are lowercased and tokenized at the word level. The vocabulary is built with unigrams and bigrams, discarding words that appear fewer than two times in the training corpus. The vocabulary size is capped (AG News: 200,000; Yelp: 100,000). We compute TF–IDF weights on the training split and apply the learned weights to the test split. Dimensionality is reduced to 128 latent components using truncated singular value decomposition (SVD). Features are standardized to zero mean and unit variance and finally $\ell_2$-normalized. We fix the random seed for reproducibility and reuse the learned preprocessing components across runs.

**Additional OOD Detection Results.** In addition to accuracy and AUROC, we also report AUPR and FPR@95 for both image and text datasets; the results are shown in Table 10. On image datasets, BL (depth=1) achieves the best overall balance: it ranks first on Fashion-MNIST AUPR and second on Fashion-MNIST FPR@95. On MNIST, it is second in AUPR but underperforms in FPR@95 compared with E-MLP (depth=2). These results suggest that BL yields separable score distributions, particularly on Fashion-MNIST, although its 95% FPR threshold admits more OOD samples than E-MLP at the same recall. On text datasets, OOD detection performance is dataset-dependent: E-MLP performs better on AG News, whereas BL achieves stronger OOD performance on Yelp.

Table 10: OOD AUPR and FPR@95 (%) on image and text datasets. BL and E-MLP are evaluated at depths 1–3 with matched parameter counts, both without skip connections. Top-two per column are blue and red.

| Model | MNIST | | Fashion-MNIST | |
|---|---|---|---|---|
| | AUPR | FPR@95 | AUPR | FPR@95 |
| E-MLP (depth=1) | $89.37 \pm 1.52$ | $35.57 \pm 5.87$ | $91.35 \pm 1.25$ | $28.24 \pm 4.37$ |
| BL (depth=1) | $91.57 \pm 2.39$ | $47.81 \pm 11.29$ | $91.79 \pm 0.90$ | $38.86 \pm 2.57$ |
| E-MLP (depth=2) | $91.52 \pm 1.27$ | $28.89 \pm 2.85$ | $86.19 \pm 2.27$ | $47.72 \pm 4.79$ |
| BL (depth=2) | $91.20 \pm 1.22$ | $52.71 \pm 18.66$ | $89.30 \pm 2.47$ | $42.65 \pm 9.53$ |
| E-MLP (depth=3) | $90.04 \pm 1.89$ | $31.92 \pm 5.76$ | $84.30 \pm 1.50$ | $54.49 \pm 2.74$ |
| BL (depth=3) | $92.36 \pm 2.03$ | $32.32 \pm 5.76$ | $88.41 \pm 4.04$ | $41.19 \pm 13.36$ |
| Model | AG News | | Yelp | |
| | AUPR | FPR@95 | AUPR | FPR@95 |
| E-MLP (depth=1) | $44.52 \pm 15.10$ | $33.82 \pm 4.89$ | $3.31 \pm 1.60$ | $54.21 \pm 2.11$ |
| BL (depth=1) | $18.68 \pm 16.48$ | $42.03 \pm 5.99$ | $12.70 \pm 2.29$ | $40.95 \pm 1.56$ |
| E-MLP (depth=2) | $31.48 \pm 23.94$ | $40.20 \pm 9.82$ | $1.47 \pm 2.31$ | $57.80 \pm 4.88$ |
| BL (depth=2) | $10.76 \pm 15.94$ | $53.71 \pm 9.68$ | $6.73 \pm 2.61$ | $46.54 \pm 1.86$ |
| E-MLP (depth=3) | $51.24 \pm 9.13$ | $32.96 \pm 3.23$ | $3.14 \pm 2.22$ | $55.24 \pm 5.33$ |
| BL (depth=3) | $16.99 \pm 17.12$ | $45.24 \pm 6.11$ | $10.96 \pm 1.10$ | $42.27 \pm 1.94$ |

**Number of Parameters.** To ensure a fair comparison between E-MLP and BL, we match the number of trainable parameters as closely as possible for models with the same depth (see Table 11).

**Running Time.** To evaluate computational cost, we compare the training time of BL and Energy-based MLP across image and text datasets (Table 12). Under comparable parameter budgets, BL generally requires slightly higher training time than E-MLP across datasets. In particular, BL is moderately slower on image datasets and AG News, while exhibiting comparable running time on Yelp.

**Calibration** We report ECE and NLL metrics to assess calibration quality, and the results are presented in Table 13. On image datasets, BL provides substantially better calibration, with BL models occupying the top two positions in each column. On text datasets, calibration performance is broadly comparable, with BL showing slightly lower NLL on Yelp. Overall, these results indicate that BL delivers strong predictive performance together with reliable probability estimates.

Table 11: Number of trainable parameters for E-MLP and BL models across high-dimension datasets.

| Dataset | Model | # Parameters |
|---|---|---|
| MNIST & FashionMNIST | E-MLP (depth=1) | 203,530 |
| | BL (depth=1) | 208,384 |
| | E-MLP (depth=2) | 235,146 |
| | BL (depth=2) | 219,264 |
| | E-MLP (depth=3) | 238,314 |
| | BL (depth=3) | 221,684 |
| AGNews | E-MLP (depth=1) | 136,196 |
| | BL (depth=1) | 149,720 |
| | E-MLP (depth=2) | 386,284 |
| | BL (depth=2) | 397,568 |
| | E-MLP (depth=3) | 230,788 |
| | BL (depth=3) | 224,128 |
| Yelp | E-MLP (depth=1) | 134,146 |
| | BL (depth=1) | 148,960 |
| | E-MLP (depth=2) | 385,770 |
| | BL (depth=2) | 397,312 |
| | E-MLP (depth=3) | 230,530 |
| | BL (depth=3) | 224,000 |

Table 12: Training time (seconds) of BL vs. E-MLP on high-dimensional datasets (mean $\pm$ std).

| Model | MNIST | FashionMNIST | AG News | Yelp |
|---|---|---|---|---|
| E-MLP (depth=1) | $100.59 \pm 0.29$ | $73.57 \pm 1.20$ | $14.69 \pm 0.40$ | $179.37 \pm 0.73$ |
| BL (depth=1) | $110.63 \pm 3.34$ | $96.52 \pm 2.90$ | $17.20 \pm 0.06$ | $181.07 \pm 1.80$ |
| E-MLP (depth=2) | $102.64 \pm 0.26$ | $78.25 \pm 0.28$ | $15.76 \pm 0.06$ | $179.22 \pm 0.66$ |
| BL (depth=2) | $122.85 \pm 3.95$ | $114.43 \pm 3.72$ | $21.78 \pm 0.08$ | $180.38 \pm 1.44$ |
| E-MLP (depth=3) | $104.52 \pm 0.30$ | $85.57 \pm 1.19$ | $16.95 \pm 0.05$ | $178.99 \pm 1.42$ |
| BL (depth=3) | $140.17 \pm 4.42$ | $130.03 \pm 4.96$ | $26.29 \pm 0.24$ | $180.36 \pm 0.91$ |

### F.5 CONSTRAINT ENFORCEMENT TEST: ACHIEVING HIGH-DIMENSIONAL ENERGY CONSERVATION

To evaluate whether the learnable penalty terms in BL are capable of enforcing near-hard constraints under finite temperature, we isolate the penalty mechanism and test it on a high-dimensional energy-conservation constraint. This diagnostic experiment removes the utility term and focuses solely on

Table 13: ECE and NLL on image and text datasets. BL and E-MLP are evaluated at depths 1–3 with matched parameter counts. Top-two per column are blue and red.

| Model | MNIST | | Fashion-MNIST | |
|---|---|---|---|---|
| | ECE | NLL | ECE | NLL |
| E-MLP (depth=1) | $0.02 \pm 0.00$ | $0.20 \pm 0.02$ | $0.08 \pm 0.00$ | $0.74 \pm 0.01$ |
| BL (depth=1) | $0.02 \pm 0.00$ | $0.26 \pm 0.01$ | $0.05 \pm 0.00$ | $0.36 \pm 0.01$ |
| E-MLP (depth=2) | $0.02 \pm 0.00$ | $0.23 \pm 0.02$ | $0.09 \pm 0.00$ | $0.89 \pm 0.03$ |
| BL (depth=2) | $0.02 \pm 0.00$ | $0.16 \pm 0.01$ | $0.07 \pm 0.00$ | $0.44 \pm 0.01$ |
| E-MLP (depth=3) | $0.02 \pm 0.00$ | $0.16 \pm 0.02$ | $0.09 \pm 0.00$ | $0.85 \pm 0.04$ |
| BL (depth=3) | $0.02 \pm 0.00$ | $0.13 \pm 0.02$ | $0.07 \pm 0.00$ | $0.49 \pm 0.02$ |

| Model | AG News | | Yelp | |
|---|---|---|---|---|
| | ECE | NLL | ECE | NLL |
| E-MLP (depth=1) | $0.02 \pm 0.00$ | $0.40 \pm 0.01$ | $0.01 \pm 0.00$ | $0.24 \pm 0.00$ |
| BL (depth=1) | $0.02 \pm 0.00$ | $0.31 \pm 0.01$ | $0.00 \pm 0.00$ | $0.20 \pm 0.00$ |
| E-MLP (depth=2) | $0.02 \pm 0.00$ | $0.42 \pm 0.01$ | $0.00 \pm 0.00$ | $0.25 \pm 0.00$ |
| BL (depth=2) | $0.06 \pm 0.01$ | $0.43 \pm 0.03$ | $0.02 \pm 0.00$ | $0.23 \pm 0.01$ |
| E-MLP (depth=3) | $0.01 \pm 0.00$ | $0.41 \pm 0.02$ | $0.00 \pm 0.00$ | $0.25 \pm 0.01$ |
| BL (depth=3) | $0.05 \pm 0.01$ | $0.39 \pm 0.02$ | $0.02 \pm 0.00$ | $0.22 \pm 0.00$ |

the penalty term, providing a characterization of how the penalty term controls constraint violations as a function of temperature $\tau$ and penalty scale $\lambda$.

**Experiment setup.** We sample $x \in R^{64}$ i.i.d. from a standard Gaussian $x \sim \mathcal{N}(0, I_{64})$ and define a pure penalty compositional utility

$$T(x, y) = \|y\|^2 - \|x\|^2, \qquad \mathrm{BL}(x, y) = -\lambda\, T(x, y)^2,$$

which plays the role of an energy-conservation residual and its quadratic penalty.

We target the Gibbs distribution

$$p(y \mid x) \propto \exp\big(\mathrm{BL}(x, y)/\tau\big)$$

using overdamped Langevin dynamics with step size $\eta = 10^{-4}$:

$$y_{k+1} = y_k + \eta\, \nabla_y BL(x, y_k)/\tau + \sqrt{2\eta\tau}\, \xi_k, \qquad \xi_k \sim \mathcal{N}(0, I_{64}).$$

For each pair $(\lambda, \tau)$ we run $512$ parallel chains, each for $1500$ Langevin steps ($500$ burn-in). We sweep over temperatures $\tau \in \{2.0, 1.0, 0.5, 0.25, 0.1, 0.05, 0.02, 0.01, 0.005\}$ at a fixed penalty $\lambda = 25$, and over penalty weights $\lambda \in \{0, 1, 3, 10, 30, 100, 200, 500\}$ at a fixed temperature $\tau = 0.05$.

For each configuration we record the residual magnitude $|T(x, y)|$ from the final state of every chain. We then report three summary statistics: (i) the mean violation $E[|T(x, y)|]$, (ii) the 95th percentile of $|T(x, y)|$, and (iii) the empirical probability of near-feasible samples. We declare a sample to satisfy the constraint approximately if

$$|T(x, y)| \leq \varepsilon_{\mathrm{tol}} \quad \text{with} \quad \varepsilon_{\mathrm{tol}} = 10^{-1},$$

and estimate $P(|T(x, y)| \leq \varepsilon_{\mathrm{tol}})$ across chains. This tolerance scale is chosen to be small relative to the typical unconstrained residuals, so that the near-feasible regime corresponds to a practically tight energy-conservation constraint.

**Constraint enforcement.** Figure 8 shows that BL achieves near-hard constraint enforcement under finite temperature and penalty scaling. Violations decrease substantially as $\tau$ decreases or $\lambda$ increases. At around $\lambda = 25$ and $\tau = 0.01$, the 64-dimensional energy-conservation constraint is enforced within $10^{-2}$ error. Curves remain mostly smooth and monotone in 64 dimensions, indicating stable Langevin sampling and effective penalty enforcement.

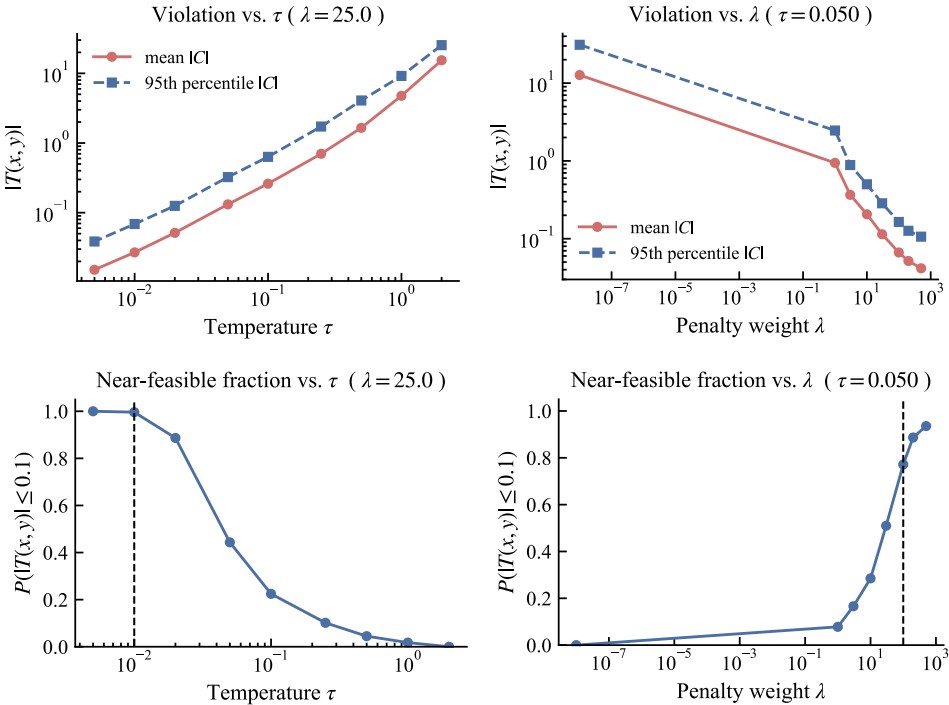

Figure 8: Constraint enforcement test of the BL penalty block on an energy-conservation constraint. The figure reports violation statistics $|T(x, y)|$ when varying the temperature $\tau$ (left side of panel) and the penalty weight $\lambda$ (right side of panel).

## F.6 ESTIMATION RESULTS OF BL ON THE BOSTON HOUSING DATASET

## G USE OF LARGE LANGUAGE MODELS (LLMS)

We used large language models (LLMs) solely for minor grammar correction and polishing of awkward sentences. They were not used in any other part of the research process.

Table 14: Estimated UMP block parameters learned by the BL model (layer = [2, 1]) on the Boston Housing dataset. For each block, $U$ denotes the Utility component, $C$ the Inequality-Constraint component, and $T$ the Equality-Constraint component.

| Variable | Block 11 | | | Block 12 | | |
|---|---|---|---|---|---|---|
| | $U_{11}$ | $C_{11}$ | $T_{11}$ | $U_{12}$ | $C_{12}$ | $T_{12}$ |
| $\lambda$ | 1.003 | 0.997 | 0.999 | 0.997 | 1.003 | 1.000 |
| per capita crime rate (CRIM) | 0.21 | 0.14 | 0.03 | 0.12 | 0.09 | 0.25 |
| residential land proportion (ZN) | 0.23 | -0.04 | -0.27 | 0.25 | 0.00 | 0.09 |
| non-retail business acreage (INDUS) | -0.06 | 0.21 | 0.25 | 0.16 | 0.22 | 0.27 |
| Charles River dummy (CHAS) | 0.25 | 0.04 | -0.24 | -0.12 | -0.20 | -0.23 |
| nitric oxide concentration (NOX) | -0.06 | -0.13 | 0.21 | 0.16 | 0.02 | -0.28 |
| average rooms per dwelling (RM) | 0.06 | 0.07 | 0.05 | 0.05 | -0.19 | -0.22 |
| proportion of older units (AGE) | -0.13 | -0.12 | -0.09 | 0.14 | 0.08 | -0.18 |
| distance to employment centres (DIS) | 0.16 | -0.03 | 0.17 | -0.17 | -0.09 | 0.11 |
| radial highway accessibility (RAD) | 0.24 | -0.11 | 0.04 | -0.28 | 0.09 | 0.10 |
| property tax rate (TAX) | -0.20 | 0.18 | 0.22 | -0.11 | -0.06 | 0.23 |
| low-income population (LSTAT) | 0.05 | -0.12 | -0.09 | 0.23 | -0.16 | -0.19 |
| median home value (MEDV) | 0.21 | -0.08 | 0.07 | 0.08 | -0.17 | 0.15 |
| Constant term (C) | 0.03 | -0.17 | -0.07 | 0.11 | -0.16 | -0.12 |

| Variable | Block 21 | | |
|---|---|---|---|
| | $U_{21}$ | $C_{21}$ | $T_{21}$ |
| $\lambda$ | 1.000 | 1.003 | 0.999 |
| Block 11 output ($b_{1,1}$) | 0.428 | -0.551 | 0.147 |
| Block 12 output ($b_{1,2}$) | -0.168 | -0.356 | -0.178 |
| Constant term (C) | 0.406 | 0.219 | 0.421 |

Table 15: Estimated UMP parameters for the Layer 1 blocks of the BL model (layer = [5, 3, 1]) trained on the Boston Housing dataset. Here, $U$ denotes the Utility component, $C$ the Inequality-Constraint component, and $T$ the Equality-Constraint component.

| Variable | $U_{11}$ | $U_{12}$ | $U_{13}$ | $U_{14}$ | $U_{15}$ |
|---|---|---|---|---|---|
| $\lambda$ | 1.000 | 0.998 | 1.003 | 1.002 | 1.000 |
| per capita crime rate (CRIM) | 0.21 | 0.12 | 0.17 | -0.09 | 0.06 |
| residential land proportion (ZN) | 0.23 | 0.25 | -0.07 | -0.22 | -0.16 |
| non-retail business acreage (INDUS) | -0.06 | 0.16 | 0.16 | 0.23 | -0.14 |
| Charles River dummy (CHAS) | 0.25 | -0.12 | -0.22 | -0.05 | -0.01 |
| nitric oxide concentration (NOX) | -0.06 | 0.16 | -0.14 | 0.24 | 0.16 |
| average rooms per dwelling (RM) | 0.05 | 0.05 | 0.08 | 0.09 | -0.07 |
| proportion of older units (AGE) | -0.13 | 0.14 | 0.06 | -0.23 | -0.15 |
| distance to employment centres (DIS) | 0.16 | -0.17 | -0.07 | 0.19 | -0.10 |
| radial highway accessibility (RAD) | 0.24 | -0.27 | 0.17 | -0.08 | -0.20 |
| property tax rate (TAX) | -0.20 | -0.11 | 0.19 | -0.11 | 0.10 |
| low-income population (LSTAT) | 0.05 | 0.23 | -0.15 | -0.28 | -0.26 |
| median home value (MEDV) | 0.21 | 0.08 | 0.25 | 0.08 | 0.06 |
| Constant term (C) | 0.03 | 0.12 | -0.09 | -0.06 | 0.15 |

| | $C_{11}$ | $C_{12}$ | $C_{13}$ | $C_{14}$ | $C_{15}$ |
|---|---|---|---|---|---|
| $\lambda$ | 0.999 | 1.001 | 1.000 | 0.997 | 1.002 |
| per capita crime rate (CRIM) | 0.13 | 0.09 | -0.10 | 0.11 | 0.05 |
| residential land proportion (ZN) | -0.04 | -0.01 | -0.27 | -0.23 | -0.10 |
| non-retail business acreage (INDUS) | 0.21 | 0.22 | -0.16 | 0.20 | 0.15 |
| Charles River dummy (CHAS) | 0.04 | -0.19 | 0.07 | -0.20 | 0.15 |
| nitric oxide concentration (NOX) | -0.13 | 0.02 | -0.04 | -0.05 | 0.11 |
| average rooms per dwelling (RM) | 0.07 | -0.19 | -0.20 | 0.06 | -0.05 |
| proportion of older units (AGE) | -0.13 | 0.08 | 0.01 | 0.14 | -0.07 |
| distance to employment centres (DIS) | -0.03 | -0.09 | -0.19 | 0.22 | 0.03 |
| radial highway accessibility (RAD) | -0.11 | 0.08 | -0.23 | 0.25 | -0.05 |
| property tax rate (TAX) | 0.18 | -0.06 | -0.15 | -0.22 | -0.08 |
| low-income population (LSTAT) | -0.13 | -0.17 | -0.18 | -0.12 | 0.24 |
| median home value (MEDV) | -0.08 | -0.17 | 0.28 | -0.03 | -0.03 |
| Constant term (C) | -0.17 | -0.16 | 0.05 | -0.21 | -0.06 |

| | $T_{11}$ | $T_{12}$ | $T_{13}$ | $T_{14}$ | $T_{15}$ |
|---|---|---|---|---|---|
| $\lambda$ | 0.999 | 1.002 | 0.999 | 1.004 | 1.001 |
| per capita crime rate (CRIM) | 0.03 | 0.25 | 0.08 | 0.25 | 0.00 |
| residential land proportion (ZN) | -0.27 | 0.10 | -0.26 | -0.20 | -0.02 |
| non-retail business acreage (INDUS) | 0.25 | 0.26 | -0.18 | 0.15 | 0.07 |
| Charles River dummy (CHAS) | -0.23 | -0.23 | -0.09 | 0.10 | 0.08 |
| nitric oxide concentration (NOX) | 0.21 | -0.28 | 0.04 | 0.09 | -0.25 |
| average rooms per dwelling (RM) | 0.05 | -0.21 | -0.24 | -0.15 | -0.10 |
| proportion of older units (AGE) | -0.09 | -0.19 | -0.12 | 0.26 | 0.24 |
| distance to employment centres (DIS) | 0.17 | 0.12 | -0.17 | 0.06 | 0.10 |
| radial highway accessibility (RAD) | 0.04 | 0.10 | 0.00 | 0.04 | -0.01 |
| property tax rate (TAX) | 0.22 | 0.23 | -0.10 | -0.24 | -0.17 |
| low-income population (LSTAT) | -0.09 | -0.19 | -0.19 | -0.04 | -0.25 |
| median home value (MEDV) | 0.08 | 0.15 | -0.19 | -0.13 | -0.09 |
| Constant term (C) | -0.07 | -0.11 | -0.16 | 0.24 | 0.10 |

Table 16: Layer 2 and Layer 3 UMP parameters ($U$, $C$, $T$) for Blocks in the BL model (layer = [5, 3, 1]).

| Variable | Block 21 | | | Block 22 | | | Block 23 | | |
|---|---|---|---|---|---|---|---|---|---|
| | $U_{21}$ | $C_{21}$ | $T_{21}$ | $U_{22}$ | $C_{22}$ | $T_{22}$ | $U_{23}$ | $C_{23}$ | $T_{23}$ |
| $\lambda$ | 1.000 | 1.000 | 1.000 | 0.999 | 1.003 | 1.002 | 1.001 | 1.002 | 0.999 |
| Block 11 output ($b_{1,1}$) | 0.28 | 0.06 | -0.20 | -0.31 | 0.24 | 0.18 | -0.29 | -0.08 | 0.22 |
| Block 12 output ($b_{1,2}$) | 0.21 | -0.11 | -0.09 | -0.44 | 0.12 | -0.22 | 0.15 | -0.22 | 0.20 |
| Block 13 output ($b_{1,3}$) | -0.40 | 0.18 | -0.44 | -0.36 | -0.01 | -0.09 | -0.13 | -0.14 | 0.32 |
| Block 14 output ($b_{1,4}$) | -0.27 | -0.17 | 0.30 | 0.33 | -0.34 | -0.26 | 0.28 | -0.42 | -0.34 |
| Block 15 output ($b_{1,5}$) | -0.07 | -0.29 | 0.34 | 0.22 | -0.38 | -0.08 | -0.14 | 0.25 | 0.32 |
| Constant term (C) | 0.43 | 0.33 | 0.16 | 0.38 | -0.42 | -0.32 | -0.33 | -0.31 | -0.21 |

| Variable | $U_{31}$ | $C_{31}$ | $T_{31}$ |
|---|---|---|---|
| $\lambda$ | 1.002 | 0.998 | 1.000 |
| Block 21 output ($b_{2,1}$) | 0.21 | -0.13 | 0.36 |
| Block 22 output ($b_{2,2}$) | 0.54 | -0.48 | 0.43 |
| Block 23 output ($b_{2,3}$) | -0.08 | 0.28 | 0.55 |
| Constant term (C) | -0.01 | -0.58 | -0.14 |

