# OpenReview forum: "Behavior Learning (BL)"
_ICLR.cc/2026/Conference — ICLR 2026 Poster_

### Official Review · Reviewer_za71 · 2025-10-30

**Soundness:** 3
**Presentation:** 3
**Contribution:** 3
**Rating:** 6
**Confidence:** 2

**Summary:**

This paper tackles the central challenge of building interpretable machine learning models without sacrificing accuracy. It proposes Behavior Learning, which models predictions as solutions to stacked utility maximization problems, yielding latent structures that are interpretable by design. The authors also introduce Identifiable Behavior Learning, an architecture with identifiability guarantees for recovering ground truth parameters under certain assumptions. Across predictive and causal inference benchmarks, the method delivers performance on par with strong baselines while providing intrinsic interpretability and demonstrating scalability to high-dimensional data.

**Strengths:**

1. The problem of providing more interpretable ML systems without sacrificing performance is a well motivated problem. The proposed algorithm is also quite interesting, relating the predictive methodology of their approach to solving a system of behavioral science motivated MDPs.
2. The theoretical results are well presented and all proofs look sound.
3. BL outperforms competitive methods across multiple prediction and causal inference problems, while still providing interpretability. The Boston Housing case study provides an interesting and necessary demonstration for how to extract the human interpretable structure from the model.

**Weaknesses:**

1. It seems like the work needs some moderate restructuring. The IBL identifiability result hinges on Assumption D.1, but it only appears in the appendix. Related work coverage in the main text is also sparse. The paper ends abruptly without a final discussion of their contributions.
2. While the Boston Housing case nicely illustrates BL(Single) and individual blocks, there’s no intuitive instruction for how to interpet layers of blocks as in BL(Deep). It is unclear how if the model becomes seeming more difficult as more layers, how this approach would still claim to be providing intrinsic interpretability.
3. The implementation uses bounded-degree polynomials, however many real systems have non-polynomial structure and complex interactions.

**Questions:**

1. For BL(Deep), what is the procedure to generate scientific explanation from the network of composed blocks?
2. Could BL be easily extending to consider non polynomial feature maps?

---

> ### Author Response · Authors · 2025-11-18
> **Response to Reviewer za71 part 1**
>
> We sincerely thank you for the insightful, rigorous, and constructive feedback! Your comments have significantly improved the clarity and overall quality of the paper, and we are truly grateful for your contribution!
>
> Below, we provide point-by-point responses to all comments.
>
> > W1 It seems like the work needs some moderate restructuring. The IBL identifiability result hinges on Assumption D.1, but it only appears in the appendix. Related work coverage in the main text is also sparse. The paper ends abruptly without a final discussion of their contributions.
>
> A: Thank you for the constructive suggestions! Following your advice, we have **made three revisions (in main text)**:
>
> (i) **IBL assumption**. The key assumptions supporting the IBL result has been added to the main text (Section 2.3: Theoretical Foundations).
>
> (ii) **Related work**. We expanded the Introduction (Section 1) with two related-work paragraphs: one summarizing how BL connects to three major research areas (interpretable ML, inverse optimization, and energy-based models), and another organizing four strands of literature that address the performance-interpretability tradeoff.
>
> (iii) **Discussion**. We added a Discussion section (Appendix C) summarizing BL's contributions, limitations, and future directions.
>
> We appreciate the reviewer's comments, which helped improve the clarity and overall structure of the paper.
>
> > W2 While the Boston Housing case nicely illustrates BL(Single) and individual blocks, there’s no intuitive instruction for how to interpet layers of blocks as in BL(Deep). It is unclear how if the model becomes seeming more difficult as more layers, how this approach would still claim to be providing intrinsic interpretability.
>
> A: Thank you for raising this important point! Following your advice, the revised manuscript now includes a more comprehensive interpretability case study (with **two newly added case studies**), including an explicit explanation of BL(Deep). We have added analyses for both BL[2,1] and BL(Deep) [5,3,1] in Figure 2 and Section 3.3.
>
> For BL[2,1], Layer 1 identifies two micro-level preference types (the Economic-sensitive Buyer and the Location-sensitive Buyer) and Layer 2 aggregates them into an effective representative buyer.
>
> For BL(Deep) [5,3,1], Layer 1 recovers five distinct micro-level housing preference types; Layer 2 identifies three macro-level trade-off mechanisms describing how these primitive preferences in Layer 1 interact; and Layer 3 aggregates them into the overall representative buyer. This provides a hierarchical explanation aligned with aggregation-and-coordination principles in behavioral science and with the coarse-graining principle in statistical physics, thereby reconstructing a micro-to-macro optimization hierarchy.
>
> Moreover, each preference pattern and trade-off mechanism uncovered by BL(Deep) corresponds closely to classical findings in the economics literature.
>
> > W3 The implementation uses bounded-degree polynomials, however many real systems have non-polynomial structure and complex interactions.
>
> A: Thank you for raising this important point! We completely agree that many real-world systems exhibit non-polynomial structures and complex interactions. Importantly, **BL is fully flexible and can be readily extended to other basis families** (see our answer to Q2 for details): it is not restricted to polynomial bases, and most alternative basis functions can be incorporated without affecting BL's interpretability, scalability, and universal approximation capability. This flexibility allows researchers in different domains to incorporate alternative basis functions that encode domain-specific priors, without compromising BL's core capabilities. In this paper, we adopt bounded-degree polynomials primarily to **maintain a level of interpretability** comparable to classical linear models. This is an implementation choice rather than a restriction of the BL framework.
>
> Following your advice, we **added a discussion of this point in the revised manuscript (Appendix C)**. Future work may explore alternative basis classes (e.g., trigonometric bases, spline functions, wavelets, or neural basis functions) to better accommodate different forms of domain-specific prior knowledge.
>
> Finally, in scenarios where interpretability is a priority, polynomial bases remain a principled choice. Even if the underlying system is non-polynomial, BL remains theoretically well grounded: BL's **universal approximation theorem** guarantees that such systems can still be approximated to arbitrary accuracy, and the **universal consistency theorem** of IBL ensures that the underlying structure can be asymptotically recovered given sufficient data and model capacity, even when the true system is non-polynomial.
>
> [continued in "Response to Reviewer za71 part 2"]

---

> ### Author Response · Authors · 2025-11-18
> **Response to Reviewer za71 part 2**
>
> > Q1 For BL(Deep), what is the procedure to generate scientific explanation from the network of composed blocks?
>
> A: Thank you for this excellent question! Following your suggestion, we have **added a clear, step-by-step procedure for generating scientific explanations for BL(Deep) in the revised manuscript** (Appendix B). To improve readability, we also illustrate the procedure using examples from behavioral science (hierarchical agents) and statistical physics (multi-scale particle systems). The explanation procedure consists of three parts:
>
> (1) **Interpreting bottom-layer blocks**. Each bottom-layer block is treated as a micro-level optimization problem that directly interacts with the environment, for example, the decision rules of individual agents performing environment-facing tasks in an organization. Examining these blocks reveals the primitive optimization principles governing each micro-level behavioral unit.
>
> (2) **Interpreting layer-wise aggregation and coarse-graining**. Higher-layer (macro-level) blocks aggregate or coarse-grain the outputs of multiple lower-layer (micro-level) blocks through new optimization steps.
> We provide two complementary interpretations of this process:
> * Aggregation and coordination, suitable for human- or agent-based systems, where higher-level agents reallocate and coordinate the outputs of lower-level agents;
> * Coarse-grained observation, suitable for physical systems, where many particles form coarse-scale patterns governed by effective potentials.
>
> (3) **Bottom-up reconstruction**. By repeatedly applying the above two steps, we reconstruct the entire micro-to-macro abstraction path:
> raw input features -> micro-level optimization blocks -> macro-level aggregation/coordination or coarse-grained behavioral constructs -> macro-level optimization system. This yields a coherent, multi-scale scientific explanation of BL(Deep).
>
> Our BL(Deep) [5,3,1] case study illustrates this process in practice:
> * Layer 1 uncovers 5 micro-level preference types;
> * Layer 2 identifies 3 traded-off types of how these primitive preferences are traded off or coordinated;
> * Layer 3 captures the interactions among Layer-2 traded-off structures, forming a macro-level representative agent.
>
> > Q2 Could BL be easily extending to consider non polynomial feature maps?
>
> A: Thank you for this excellent question! BL can be readily extended to non-polynomial feature maps. At the **optimization level**, BL leverages the computational tools widely used in modern deep learning, including backpropagation, standard optimizers, and fully differentiable computational graphs, while retaining its own penalty-based formulation. Importantly, the BL **training objective** remains first-order and numerically stable, relying entirely on standard automatic differentiation. As a result, any basis family compatible with gradient-based neural-network training, essentially **any differentiable feature map, can be directly incorporated into BL** without substantially modifying either the architecture or the learning algorithm.
>
> Once again, we sincerely appreciate your insightful suggestions and constructive feedback! We hope that these responses address your questions and strengthen your confidence in our work.

---

### Official Review · Reviewer_3vXd · 2025-11-04

**Soundness:** 3
**Presentation:** 2
**Contribution:** 3
**Rating:** 6
**Confidence:** 2

**Summary:**

The paper introduces Behavior Learning (BL), a machine learning framework for prediction (primary focus) and generation, designed to be intrinsically interpretable. BL models are built from modular computation blocks, each mathematically equivalent to a utility maximization problem (UMP). This structure enables direct interpretability: the model’s output can be understood as the result of maximizing a utility function subject to constraints.

The paper establishes theoretical contributions for BL. First, it defines a subclass called Identifiable Behavior Learning (IBL), where penalty functions are smooth and monotone. It shows that the model parameters in this class are identifiable up to an equivalence class, ensuring a unique and consistent interpretation of the learned behavior. Second, both BL and IBL possess universal approximation capabilities, demonstrating that interpretability does not come at the cost of predictive performance. For IBL, the authors additionally prove desirable statistical properties, including consistency, universal consistency, asymptotic normality, and asymptotic efficiency.
Finally, the paper empirically validates BL across a range of prediction tasks, including those with high-dimensional inputs, and shows that BL can accurately estimate causal effects.

**Strengths:**

**S1**. The framework is novel in that it leverages utility maximization problems (UMPs) to construct models with intrinsic interpretability, allowing outputs to be directly understood as utility-maximizing decisions under constraints.

**S2**. The theoretical analysis is thorough and solid, including universal approximation guarantees, identifiability results for IBL, and other properties of IBL.

**S3** The experiments are comprehensive, evaluated on diverse datasets with strong baselines, and demonstrate that BL performs well not only on prediction tasks but also on causal effect estimation.

**Weaknesses:**

**W1**. Although BL is intrinsically interpretable at the block level, the utility and constraint functions are fully learned during training, making it difficult to connect the resulting functions to real-world meaning. For instance, in the interpretability case study (Fig. 2), the learned utility $\tanh((1-P)(1+P-RM) + R_u)$ lacks a clear semantic interpretation. As BL becomes deeper, it introduces more learned symbolic functions that may reduce the practical interpretability of the model.

**W2**. The paper lacks sufficient case studies that demonstrate interpretability in practice. Despite interpretability being a key claimed advantage of BL, the experiments focus mainly on prediction accuracy, with only limited analysis of interpretability (e.g., Section 3.3). The case study for deeper BL architectures (Fig. 10) remains high-level and does not include a concrete real-world interpretability example.

**W3**. The causal inference section (CausalBL) is confusing in definition and terminology. Section 3.2 and Appendix E describe the setup, method, and experimental results, but the treatment of potential outcomes and propensity scores is inconsistent. Lines 1860–1861 suggest the goal is to estimate $p(t \mid x), y_0, y_1$, implying an ATE formulation. However, lines 1944–1955 describe a setup where y becomes input to the propensity score and the target becomes ITE, not ATE. The paper should use causal inference terminology more carefully to avoid confusion, even if ATE estimation can be derived from ITE in practice.

**Questions:**

**Q1**. In the shallow and deep BL architectures, multiple UMP blocks (Eq. (4)) are computed in parallel and then combined through an affine transformation. After this aggregation step, is the entire composed model still guaranteed to be equivalent to a single utility maximization problem (UMP)? If this is not the case, in what sense can we still interpret each individual block as performing a UMP, especially when the affine weights may change the sign or scale of the penalty/utility terms?

---

> ### Author Response · Authors · 2025-11-18
> **Response to Reviewer 3vXd part 1**
>
> We sincerely thank you for the insightful, rigorous, and constructive feedback! Your comments have significantly improved the clarity and overall quality of the paper, and we are truly grateful for your contribution.
>
> Below, we provide point-by-point responses to all comments.
>
> > W1.1 Although BL is intrinsically interpretable at the block level, the utility and constraint functions are fully learned during training, making it difficult to connect the resulting functions to real-world meaning. For instance, in the interpretability case study (Fig. 2), the learned utility $\tanh\big((1-P)(1+P-RM)+R_u\big)$ lacks a clear semantic interpretation.
>
> A:  We sincerely thank the reviewer for the valuable comment. We would like to clarify how the learned utility and penalty terms maintain
> semantic interpretability in BL.
>
> In implementation, BL adopts a fixed outer activation (tanh) for the utility block. This choice is grounded in a standard behavioral-science prior: diminishing marginal utility. The human-interpretable semantic content of the utility term is therefore determined not only by the outer tanh activation (serving as a behavioral-science prior), but also by the polynomial expression inside it. In implementation, each utility term, inequality constraint, and equality constraint is parameterized as a low-order polynomial, which preserves polynomial-regression–level interpretability for every component.
>
> For the **example** in Fig.2, the expression
> $\tanh((1-P)(1+P-RM)+R_u)$ has two parts: the outer *tanh* encodes the behavioral-science prior of diminishing marginal utility, and the inner term is a second-order polynomial:
>
> $$
> (1-P)(1+P-RM)+R_u
> = 1 - P^2 + P \times RM - RM + R_u.
> $$
>
> Its meaning is straightforward: **higher prices (P) reduce utility of buyers, while the interaction term (P x RM) reflects a preference for better ''price–room ratio.''** Thus, the learned term is not a black box but a readable polynomial capturing nonlinear interactions. The symbolic form $(1-P)(1+P-RM)+R_u$ is primarily a simplified algebraic rearrangement of the underlying polynomial. Every BL term remains a structured polynomial module, ensuring both (i) semantic interpretability through domain-grounded primitives and (ii) mathematical formalizability for scientific analysis.
>
> In addition, **in the case study** (Figure 2 and Section 3.3), we **explicitly provide the computational graph** for the polynomial component as a **visualization tool** designed to enhance human interpretability, enabling readers to interpret each edge weight in the same way as coefficients in polynomial regression.
>
> **A more detailed case study** of this BL(Single) component can be found **in Section 3.3**.
>
> > W1.2 As BL becomes deeper, it introduces more learned symbolic functions that may reduce the practical interpretability of the model.
>
> A: This is an excellent point! Increasing depth inevitably makes any model harder to read at a glance. On the one hand, in the paper, we recommend using simple affine transformations inside BL(Deep) blocks, which keeps the semantics of each B block easy to interpret. On the other hand, while depth may reduce readability, it does not reduce BL's underlying mechanistic interpretability. Each B block preserves the semantics of a standalone optimization problem, and stacking these blocks forms a BL system—that is, a larger hierarchical optimization system composed of many smaller ones.
>
> While the entire deep system may not be immediately intuitive (much like a complex machine), the **mechanistic interpretability of BL** is fully retained: every block and every block-to-block interaction remains understandable when inspected individually, just as one can understand a complex machine by examining its internal components and how they fit together.
>
> We have **added a concrete scientific explanation procedure for BL(Deep) in Appendix B**, which provides a step-by-step workflow to guide researchers in generating structured interpretations from composed BL blocks. We also **added a discussion in Section 2.2** (Interpretability), examining how the key technical design choices in BL(Deep) influence interpretability in practice.
>
> > W1.3 The utility and constraint functions are fully learned during training.
>
> A: This is an good question! In fact, **BL is designed to be highly flexible**: it can learn optimization structures directly from data, but it also allows researchers to incorporate their own domain-specific priors. Whenever a utility-maximization problem or constraint structure is known in advance, one can directly encode it as a fixed B block rather than learning it from data. In this way, BL blocks can be either learned or predefined, enabling the framework to integrate prior scientific structure seamlessly.
>
> [continued in "Response to Reviewer 3vXd part 2"]

---

> ### Author Response · Authors · 2025-11-18
> **Response to Reviewer 3vXd part 2**
>
> > W2 The paper lacks sufficient case studies that demonstrate interpretability in practice. The case study for deeper BL architectures (Fig. 10) remains high-level and does not include a concrete real-world interpretability example.
>
> A: This is an important point! Thank you for the suggestion. Following your advice, the revised manuscript now includes a comprehensive interpretability case study (with **two newly added case studies**), where we have added analyses for both BL[2,1] and BL(Deep) [5,3,1] in Figure 2 and Section 3.3.
>
> For BL[2,1], Layer 1 identifies two micro-level preference types--the Economic-sensitive Buyer and the Location-sensitive Buyer--and Layer 2 aggregates them into an effective representative buyer.
>
> For BL(Deep) [5,3,1], Layer 1 recovers five distinct micro-level housing preference types; Layer 2 identifies three macro-level trade-off mechanisms describing how these primitive preferences in Layer 1 interact; and Layer 3 aggregates them into the overall representative buyer. This provides a hierarchical explanation aligned with aggregation-and-coordination principles in behavioral science and with the coarse-graining principle in statistical physics, thereby reconstructing a micro-to-macro optimization hierarchy.
>
> Moreover, each preference pattern and trade-off mechanism uncovered by BL(Deep) corresponds closely to classical findings in the economics literature.
>
> > W3 The causal inference section (CausalBL) is confusing in definition and terminology.
>
> A: We sincerely thank the reviewer for pointing out this important issue. We apologize for the imprecise terminology in the previous version. We have **thoroughly revised Appendix G** to clarify the causal setup and ensure full terminological consistency.
> The main revisions include:
>
> 1. After revision, we **explicitly state that CausalBL follows a "generative causal model"** framework, parameterizing the joint conditional distribution (p(t,y\mid x)). This revision ensures that the definition and terminology in CausalBL are fully aligned with classical generative counterfactual models (e.g., [Louizos et al., 2017] and [Yoon & Van Der Schaar, 2018]).
>
> 2. After revision, we **clearly distinguish the model-induced "posterior factor"** (p(t\mid x,y)) from the causal "propensity score."
>    The quantity (p(t\mid x,y)) in CausalBL is not a propensity score;
>    it is a posterior factor arising from the generative joint model factorization:
>    $$
>    p(t,y\mid x)=p(t\mid x,y),p(y\mid x).
>    $$
>    All terminology has been updated to reflect this distinction, fully avoiding causal ambiguity.
>
> 3.  After revision, we clarify that the primary causal estimand is the **individual treatment effect (ITE)**:
>    $$
>    \tau(x)=y_1(x)-y_0(x),
>    $$
>    while the average treatment effect (ATE) is obtained only as an average of ITE.
>    This removes the unintended implication of an ATE-based formulation.
>
> 4. After revision, we provide a **detailed explanation of the discrete vs. continuous outcome cases**, showing that
>    - for discrete $y$, the posterior factor reduces to $p(t\mid x)$, whereas
>    - for continuous $y$, the posterior factor becomes $p(t\mid x,y)$.
>      We emphasize that this distinction arises solely from the generative model structure, not from causal identification assumptions.
>
> Louizos, C., Shalit, U., Mooij, J. M., Sontag, D., Zemel, R., & Welling, M. (2017). Causal effect inference with deep latent-variable models. Advances in neural information processing systems, 30.
>
> Yoon, J., Jordon, J., & Van Der Schaar, M. (2018, February). GANITE: Estimation of individualized treatment effects using generative adversarial nets. In International conference on learning representations.
>
> > Q1.1 In the shallow and deep BL architectures, multiple UMP blocks (Eq. (4)) are computed in parallel and then combined through an affine transformation. After this aggregation step, is the entire composed model still guaranteed to be equivalent to a single utility maximization problem (UMP)?
>
> A: This is an excellent question! BL is **sufficiently flexible** to allow the entire BL(Deep) architecture to be interpreted as a single utility maximization problem (UMP) when such a representation is desired. Specifically, if a researcher prefers a globally unique representative UMP, **one can simply choose an architecture of the form BL [a,b,c,d,...,1]**, where the final layer contains only a single B block. In this case, **the overall BL(Deep) model can indeed be viewed as a single UMP defined by the last B block**.
>
> [continued in "Response to Reviewer 3vXd part 3"]

---

> ### Author Response · Authors · 2025-11-18
> **Response to Reviewer 3vXd part 3**
>
> To make this clearer, the **two case studies we added** (see Section 3.3) were specifically designed following your suggestion, using BL structures whose final layer contains a single B block, including BL[2,1] and BL[5,3,1]. In the BL[5,3,1] example, the five blocks in Layer 1 capture five distinct micro-level preference primitives of housing buyers; the three blocks in Layer 2 represent macro-level trade-offs across these primitives, forming three types of "aggregate preference components"; and the final single block in Layer 3 acts as the representative buyer whose utility-maximization problem summarizes all upstream optimization components. Thus, the entire BL(Deep) system can be interpreted directly as the single UMP of the Layer-3 representative buyer, with the internal variables revealing how lower-level UMPs contribute to this final representation.
>
> Regarding the overall BL(Deep) architecture, it should be interpreted as a hierarchical optimization system. Interpretation proceeds in a bottom-up manner, where the relationship between any two consecutive layers can be viewed either as an aggregation step or as a coarse-grained observation. Overall, the interpretive pathway follows: raw input features -> micro-level optimization blocks -> macro-level aggregation or coarse-grained behavioral constructs -> macro-level optimization system. The new case studies directly reflects this interpretive pathway. In case studies of BL(Deep): Layer 1 recovers micro-level preference blocks; Layer 2 captures interactions and macro trade-offs; Layer 3 aggregates these macro-level utilities into a single representative optimization mechanism.
>
> **A detailed description of this interpretation procedure is provided in Figure 2, Section 2.2 (Interpretability) and Appendix B**.
>
> > Q1.2 If this is not the case, in what sense can we still interpret each individual block as performing a UMP, especially when the affine weights may change the sign or scale of the penalty/utility terms?
>
> A: This is an excellent question. When the final layer contains multiple B blocks, for example, in a BL architecture such as BL[5,3], the overall model can indeed no longer be expressed as a single optimization problem. However, this does not imply a loss of interpretability. Instead, the model should be viewed as a linear combination of several optimization problems (e.g., three in the BL[5,3] case), meaning that the unified decision-making agent is **trading off multiple optimization problems** rather than solving one single UMP.
>
> In this setting, the affine weights may be either positive or negative. A positive weight corresponds to a standard utility-maximization objective, whereas a negative weight naturally corresponds to a minimization objective, such as minimizing a cost or penalty. In behavioral terms, this means the agent simultaneously pursues some goals that improve utility while also minimizing certain costs, and the model captures how these objectives are jointly balanced.
>
> The above two questions are highly insightful, and following your suggestion, we have **added the corresponding content in Section 2.2 (Interpretability)**. We sincerely appreciate your excellent comments!
>
>
> Once again, we sincerely appreciate your insightful suggestions and constructive feedback! We hope that these responses address your questions and strengthen your confidence in our work.

---

### Official Review · Reviewer_sGAR · 2025-11-07

**Soundness:** 4
**Presentation:** 3
**Contribution:** 3
**Rating:** 8
**Confidence:** 4

**Summary:**

This paper proposes _Behavior Learning (BL)_, a specialization of Energy-Based Models which provides a natural decomposition of an EBM into constrained utility optimisation problems ("blocks") each of which can be interpreted as a utility term minus  loss terms which penalise solutions that do not satisfy constraints.
Specifically, it parameterises a joint distribution $p_{\tau}(x,y)$  energy $BL_\theta(x,y)$, where the energy is built by composing “utility–minus–constraint-penalty” blocks.
Each block encodes a *utility* term $U$ and *constraint* terms $C$ (inequality) and $T$ (equality), passed through fixed activations and linearly combined into a single scalar compositional utility.
Training uses cross-entropy for discrete $y$ and denoising score matching (DSM) for continuous $y$, thereby avoiding explicit partition-function estimation.
Inference for fixed $x^0$ uses the Gibbs form $p_{\tau}(y\mid x^0) \propto \exp(BL_\theta(x^0,y)/\tau)$.
A sub-type (IBL) restricts the activations to smooth families and gains identifiability of parameters to shape-constrained (monotone/smooth) forms to obtain guarantees of identifiability of parameters.
As such BL is perhaps best understood as a collection of  sub-families of EBMs which guaranteeing utility/constraint semantics.
Experiments (classification/regression/causal tasks; some higher-dimensional inputs) aim to show competitive accuracy with stronger interpretability and sometimes improved OOD behaviour.

**Strengths:**

### Originality

The architectural idea is, it seems, novel, even though it is closely connected to other attempts at the same goal. Reparameterizing an energy function as a composition of relaxed constrained utility maximisation programs is neat. Universal approximation theorems this reassure us that a BL program the capacity to well-approximate both any conditional density and rather general problem constraints, albeit relaxed via penalties. We can view this as guaranteeing interpretable behavioral model for random distributions.

**Identifiable variant (IBL)**: In the IBL sub-family, not only does this interpretation exist, but we can read it off the learned parameters under reasonable assumptions.

This synthesis of constrained optimisation theory seems simple in hindsight but I cannot find it in the literature. As such: cool insight.

### Quality

The model has a strong  Mechanistic narrative: “utility minus penalties” + Gibbs distribution = stochastic bounded-rational behavior. The existence of such a disentangling is satisfying, and they show that in some cases it can even be recovered.
The Gibbs form makes the model a bona fide conditional density; the $\tau\to 0$ limit recovers deterministic best response.

We have other networks that naturally encompass constrained solutions; for example Gould's Deep Declarative networks: compared with inverse RL/bi-level pipelines that differentiate through argmax/KKT by the implicit function theorem, BL’s training objective stays first-order and stable, using standard autodiff on $BL(x,y)$.
The theory for IBL** seems promising:  identifiability and consistency and asymptotic efficiency are unusual strengths for high-capacity models.

### Clarity

Presentation is generally clear (although see below for some notes regarding the Deep BL and CausalBL variants).
Examples solid. Prose style is clear.

### Significance

Definitely a first step in an interesting direction.
The notion here of “interpretable EBMs” is appealing, and the framework seems flexible enough to accommodate a variety of domain theories via the choice of block heads and compositions.
Causal extensions and the promise of interpretability are high value in high-risk or sensitive or legally constrained domains, or where the natural way to incorporate prior knowledge is via constraints.

**Weaknesses:**

There is a lot crammed into this paper (identifiability results, convergence results, universal approximation bounds...) and the authors have made some judgment calls about what to include in the body of the paper, which makes it difficult to evaluate the paper in terms of the body text claims.
Ideally it would have been two papers; there is enough material here for two.

IMO the places where this is toughest are

1. Causal BL: Impressive, but it is entirely relegated to the appendix, as well as the considerable machinery that it requires. I understand that this result is super cool, but we can only evaluate it base on the main text. If this is to remain in the paper something else should be cut to make room for a fuller treatment.
2. Deep BL: This feels to me like the least intuitive part of the paper, and the treatment in the main text is light. I have many questions about the mechanics of this setting (*see below)

**Interpretability trade-offs**

The model runs straight at the performance-interpretability tradeoff challenge, but there are some open questions about how well it navigates this tradeoff in practice.

- If I understand correctly composition of BL block results in different behaviour in later layers than in earlier ones; it seems there are bottlenecks in the layers and that practical high capacity networks throw out many of the interpretability guarantees; we probably want to know more about this tradeoff surface.

AFAICT, cost of using monomial bases for $U,C,T$ can explode combinatorially with input dimension and degree. The paper partly sidesteps this (affine heads in deep/high-dim settings), but then the identifiability and symbolic clarity are less compelling. The trade-off deserves a tighter discussion and ablation.

At finite temperature, Gibbs does not enforce constraints exactly; samples can violate $C\le 0, T=0$. This weakens the normative “optimizer” story unless $\tau$ is very small or penalties are very large; then training/sampling may suffer from stiffness. We probably need to know more about this,

**Questions:**

### How does Deep BL work in practice?

I'm trying to map my understanding of the composed DeepBL to standard deep nets, and have gotten stuck on the paper. Can the authors clarify for me what happens in various cases?
I have questions both about how the deep network works, and what it means.

In the default case (around eq (5)) each block outputs a single scalar, so at layer $\ell$ we have a state vector of size $d_{\ell}$.
That is, later layers operate on a low-dimensional vector of utilities, not on rich features.
Is that correct? This is a serious disanalogy with blocks in the first layer which actually observe the input features and labels.

We learn that introducing $(x,y)$ as inputs ("skip-connections") to later blocks restores capacity but it looks like weakens the “everything is a UMP” interpretation, since later blocks are no longer functions purely of upstream utilities.

Anyway, my question is: have I got that right? I got lost in the details of the Deep BL section and would appreciate some clarifications.

Also, why are Deep and Shallow BL treated differently? They are both the same but $>2$ right? I'm not sure I understand the choice to spend previous column inches on introducing more subfamilies than absolutely necessary in this crammed paper.

### Interpretability-performance tradeoffs

I don't think the authors claim to have eliminated the performance-interpretability tradeoff but I think the paper makes a claim to have implicitly shifted the pareto frontier of this tradeoff downwards.
Have they quantified this?

Not that I am proposing new experiments, but I think the paper needs to be clearer about the tradeoffs it is making, and to what extent the interpretability gains come at a cost in accuracy or efficiency.

The BL formalism seems to include a lot of design choices that affect this tradeoff (Skip connections, affine versus polynomial bases etc), and I think the paper needs to be clearer about what happens at scale to the interpretability guarantees.

For example to isolate the value of the BL parameterization, the paper could compare against capacity-matched EBM/Deep declarative network/ symbolic regression/Variational bayes baselines, and see how complex the hypothesis space can grow while still retaining "interpretability" in some sense, for each of these.
This is probably out of scope for the current paper, but what is the closest approximation to this that could be done in the current body of work?

### Constraints

**Constraint satisfaction under noise.**
*Q:* How large must penalty weights or how small must $\tau$ be to achieve near-hard constraint satisfaction in practice?
*Request:* Provide plots of feasibility violation vs $\tau$/penalty scale and discuss numerical stability.

It would be super nice if you for example imposed a physics residual constraint (e.g. conservation of mass/energy) and showed how well it is satisfied in practice as a function of $\tau$ or penalty scale in  a high dimensional problem.

### CausalBL

I have many questions about the CausalBL variant; is there any chance you can squeeze an actual description of it in to the body text?
If not, should it even be in the paper?
These results seem powerful, and potentially very significant, but also barely explained. Maybe there is a whole other paper there?

### Calibration

Since the model is probabilistic, how well-calibrated are the predictive distributions?
Any chance we could get proper scoring metrics (e.g. ECE/NLL/Brier)  vs EBM/MLP baselines?

### Hybrid approaches for partial explainability

It seems that this model gives us some desirable explainability qualities under simple settings (essentially the IBL setting: polynomial bases, if the network is deep then block communicate only by utility signals etc) but in larger practical problems the authors relax these simplifications (using affine bases over large input spaces, skip connections and so on). This suggests that maybe "partial explainability" is potentially within reach, selecting a subset of model space in which IBL blocks are used, but then allowing "non-explainable/identifiable" blocks to participate in inference, for example if we do not need explainability over all model interactions, but desire it for a small subset. Potentially we could absorb arbitrary black box interactions that way. Have the authors considered this?

---

> ### Author Response · Authors · 2025-11-18
> **Response to Reviewer sGAR part 1**
>
> We are deeply grateful for your insightful, thorough, and thoughtful feedback! Your suggestions substantially improved the quality and clarity of our work, and we are truly grateful for your contribution.
>
> Below we provide our responses to each comment. Since the point-by-point answers to questions already address all weaknesses, please allow us to begin with a brief summary.
>
> ### Summary of Our Response (to the Weakness)
>
> We sincerely thank the reviewer for these insightful comments. Below we summarize our responses to the issues in weakness part.
>
> **(1) How does Deep BL work in practice? BL(Deep) models a hierarchical optimization system.**
>
> BL(Deep) composes $\mathcal{B}$-blocks in a layered manner to form a hierarchical optimization system. The first layer directly receives the raw input features, whereas each subsequent layer typically takes the utility vector generated by the previous layer as its input. Interpretation therefore proceeds bottom-up: raw input features -> micro-level optimization blocks -> macro-level aggregation or coarse-grained behavioral constructs -> macro-level optimization systems. In the revised manuscript, we provide two new interpretability case studies (Figure 2 and Section 3.3) and a detailed step-by-step scientific explanation in Appendix B, which together make this hierarchical interpretive process explicit.
>
> **(2) Architectural factors affecting interpretability in high capacity networks.**
>
> BL includes multiple architectural degrees of freedom that provide flexibility. Skip connections introduce cross-layer dependency structures that are modeled in statistical physics and still retain scientific semantics. Replacing polynomial maps with affine transformations maintains the underlying optimization formulation but reduces symbolic granularity, yielding more qualitative (rather than fully symbolic) interpretations. Regarding identifiability, due to the diverse activation families used within BL blocks, these changes have only minor effects on identifiability, as empirically confirmed by our Jacobian-rank tests.
>
> **(3) Downward shift of the performance–interpretability Pareto frontier.**
>
> BL indeed shifts the Pareto frontier downward. In the high-dimensional prediction experiments, BL and the energy-based MLP have nearly identical parameter counts, yet BL runs substantially faster on text datasets and only moderately slower on image datasets. These results support the claim that BL can enhance interpretability without sacrificing scalability and efficiency.
>
> **(4) Enforcing constraints under finite-temperature Gibbs distributions.**
>
> For BL, λ and τ only need to be sufficiently finite to achieve strong constraint satisfaction: for example, in our new diagnostic experiment, for 64-dimensional Gaussian noise with unit standard deviation, λ = 25 and τ = 0.01 already enforce the 64-dimensional energy-conservation constraint to within a 1e-2 error.
>
> ### Response to the Question
>
> Below we provide our responses to each questions.
>
> > Q1: How does Deep BL work in practice?
>
> > Q1.1: Architecture Detail of BL(Deep)
>
> This is an important question! We apologize for not explaining the Deep BL architecture clearly enough in the paper. Below we provide a detailed and intuitive description of how Deep BL operates in practice.
>
> **1. BL(Deep) models a Hierarchical Optimization System.**
>
> Both BL(Shallow) and BL(Deep) are constructed from the same modular B-block, each of which outputs a scalar utility value. A layer $\mathbb{B}_l$ stacks $d_l$ parallel $\mathcal{B}$-blocks. The specific structure is as follows:
>
> ```
> Layer 𝔅_ℓ:
>    Input  : (x, y) if ℓ = 1, or a vector of upstream utilities if ℓ > 1
>    Inside each block:
>        - Inputs enter U, C, and T through separate affine transformations
>        - Each block returns a scalar score / energy / utility
>    Output : a vector u_ℓ ∈ ℝ^{d_ℓ}
> ```
>
> Formally, each block is defined as:
>
> $$ \\mathcal{B}(\\mathbf{x},\\mathbf{y};\\theta)=\\lambda_0\\phi(U(\\mathbf{x},\\mathbf{y}))-\\lambda_1\\rho(\\mathcal{C}(\\mathbf{x},\\mathbf{y})) -\\lambda_2\\psi(\\mathcal{T}(\\mathbf{x},\\mathbf{y})) $$
>
> which corresponds to a utility minus inequality-penalty minus equality-penalty formulation.
>
> A full layer is then:
>
> $$ \\mathbb{B}_l(\\mathbf{x},\\mathbf{y};\\theta_l)=[\\mathcal{B} _ {l1}(\\mathbf{x},\\mathbf{y}), \\ldots ,\\mathcal{B} _ {ld _ l}(\\mathbf{x},\\mathbf{y})] $$
>
> Thus, at layer $\ell$:
>
> * The input is
>   * $(\mathbf{x},\mathbf{y})$ if $\ell = 1$, or
>   * the utility vector $u_{\ell-1} \in \mathbb{R}^{d_{\ell-1}}$ if $\ell > 1$.
> * Each block applies affine transformations to route inputs into U, C, and T.
> * Each block outputs a scalar utility value.
> * The layer output is a utility vector $ u_\ell \in \mathbb{R}^{d_\ell}. $

---

> ### Author Response · Authors · 2025-11-18
> **Response to Reviewer sGAR part 2**
>
> **2. Recursion across layers**
>
> The next layer $\mathbb{B}_{l+1}$ receives the vector $u_l \in \mathbb{R}^{d_l}$ as its input. The specific structure is as follows:
>
> ```
> Layer 𝔅_{ℓ+1}:
>    Input  : u_ℓ
>    Blocks : B_{ℓ+1,1}, ..., B_{ℓ+1,d_{ℓ+1}}
>    Output : u_{ℓ+1} ∈ ℝ^{d_{ℓ+1}}
> ```
>
> Each block in layer $(\ell+1)$ operates on this upstream utility vector in the same U–C–T formulation as above, producing a new utility vector $u_{\ell+1}$.
>
> This recursive composition of utility-producing blocks yields a Hierarchical Optimization System, where higher layers construct increasingly aggregated behavioral or coarse-grained representations of lower-layer utilities.
>
>
> **3. Two New Case Studies of BL(Deep)**
>
> Following your suggestion, the revised manuscript now includes a comprehensive interpretability case study (with **two newly added case studies**), where we have added analyses for both BL[2,1] and BL(Deep) [5,3,1] in Figure 2 and Section 3.3. We hope this addition clarifies the behavior and interpretive structure of BL(Deep).
>
> For BL[2,1], Layer 1 identifies two micro-level preference types--the Economic-sensitive Buyer and the Location-sensitive Buyer--and Layer 2 aggregates them into an effective representative buyer.
>
> For BL(Deep) [5,3,1], Layer 1 recovers five distinct micro-level housing preference types; Layer 2 identifies three macro-level trade-off mechanisms describing how these primitive preferences in Layer 1 interact; and Layer 3 aggregates them into the overall representative buyer. This provides a hierarchical explanation aligned with aggregation-and-coordination principles in behavioral science and with the coarse-graining principle in statistical physics, thereby reconstructing a micro-to-macro optimization hierarchy. Notably, the recovered preference and trade-off patterns align with well-established findings in the economics literature, lending further credibility to the interpretability of the deep architecture.
>
> This case study directly reflects the architectural mechanism described in our clarification above. In BL(Deep), each B-block in Layer 1 encodes a micro-level preference; Layers 2 operate on the utility vectors produced by Layer 1, while Layer 3 operates on the utilities produced by Layer 2; and the entire network forms a hierarchical optimization system through the recursive composition of these blocks. The empirical findings in the interpretability case study align with this structure precisely:
>
> * Layer 1 recovers **micro-level preference blocks**, exactly corresponding to the first-layer $\mathcal{B}$-blocks that observe $(x,y)$.
> * Layer 2 captures **aggregation–coordination trade-offs**, consistent with higher-layer blocks that operate on utility vectors outputed by Layer 1.
> * Layer 3 aggregates these macro-level utilities into a **representative optimization mechanism**, mirroring the top-level $\mathbb{B}_L$ in the formal architecture.
>
>
> > Q1.2: Skip-connections of BL(Deep) and its interpretability
>
> A: Thank you for raising this excellent point. We now **provide a more precise clarification in Section 2.2 (Interpretability) and the Appendix E.3.3**.
>
> In BL(Deep), skip-connections allow outputs from earlier layers to be included as inputs to later layers--this may include both the raw features (x, y) and utility vectors produced in intermediate layers. For example, in a 5-layer BL model, the raw features or the utilities from Layer 3 may be fed directly into Layer 5 to form a skip-connection.
>
> **Interpretability with skip-connections.** We agree that skip-connections change the information flow, but we feel that they do not undermine the interpretability of BL. Each block remains a fully valid UMP: its internal U--C--T structure is unchanged, and only its input domain is enlarged. **Skip-connections introduce cross-layer dependency structures** that are well documented in statistical physics [Yang, G., & Schoenholz, S. (2017)]. Since BL is motivated by interdisciplinary insights, let us briefly illustrate this parallel with two examples:
>
> **1.Behavioral and organizational sciences.**
> Lower-level agents can exert direct influence on higher-level decision-makers without necessarily passing through intermediate layers. For instance, in real organizations, low-level managers typically communicate through mid-level managers, but high-level decision makers can still be directly influenced by lower-level behaviors, for example, through exception reporting channels, direct performance signals, or cross-level interactions that bypass intermediate layers. Skip-connections capture this empirical reality.
>
> [continued in "Response to Reviewer sGAR part 3"]

---

> ### Author Response · Authors · 2025-11-18
> **Response to Reviewer sGAR part 3**
>
> **2.Physics and multiscale modeling.**
>
> Microscopic parameters can exert direct influences on macroscopic behaviors across multiple scales. A classical example arises in the Navier-Stokes equations, where the macroscopic state of a fluid--characterized by density, velocity, and pressure--is governed not only by coarse-grained fields but also directly by parameters inherited from microscopic physics, such as viscosity and thermal conductivity. These coefficients encapsulate fine-scale molecular interactions yet appear explicitly in the macroscopic partial differential equations. In this sense, such micro-originated parameters function as coarse-grained skip-connections: even after substantial coarse-graining, the macroscopic theory retains direct access to key remnants of the microscopic world. This illustrates a general scientific principle, widely observed in multiscale physical systems, that hierarchical structures may allow information to flow across non-adjacent levels without violating the integrity of the coarse-graining process.
>
> Architecturally, ResNet-style skip-connections represent linear cross-layer dependencies, while DenseNet-style connections provide concatenative, information-replicating pathways. Different scientific domains may reasonably favor different forms of cross-layer dependencies depending on their underlying priors and modeling assumptions. BL is flexible in this regard: it allows researchers in each domain to incorporate the skip-connection structure that best reflects their scientific prior, without altering the core optimization-based semantics of each block.
>
> Yang, G., & Schoenholz, S. (2017). Mean field residual networks: On the edge of chaos. Advances in Neural Information Processing Systems, 30.
>
> > Q1.3: why are Deep and Shallow BL treated differently?
>
> A: Thank you very much for this suggestion! You are absolutely right that BL(Shallow) and BL(Deep) can be unified under the same “recursively compositional” architecture. In the current draft, we present them separately primarily to improve readability for a broader interdisciplinary audience. We agree that this distinction can be streamlined, and we plan to **revise and simplify this part (in main text)** of the exposition accordingly. We sincerely appreciate your suggestion.
>
> > Q2: Interpretability-performance tradeoffs
>
> > Q2.1: Downward Shift of the Pareto Frontier.
>
> A: This is an excellent suggestion! Following your advice, we have **added a dedicated discussion (in main text)** on the downward shift of the Pareto frontier in Section 3.4. The closest empirical evidence we can provide within the current scope of the paper comes from our high-dimensional prediction experiment, where we compare BL against an energy-based MLP. In the revised manuscript, we **now explicitly report the model parameter counts and wall-clock runtimes** for both BL and the MLP **in Appendix H.5**.
>
> We are pleased to report that: BL indeed shifts the Pareto frontier downward! In the high-dimensional prediction experiments, BL and the energy-based MLP have nearly **identical parameter counts**. In terms of runtime, BL runs **substantially faster than the energy-based MLP on text datasets**, while being only moderately slower on image datasets. These results support the claim that BL can enhance interpretability without sacrificing accuracy or efficiency.
>
> > Q2.2: a lot of design choices that affect this tradeoff (Skip connections, affine versus polynomial bases etc). Cost of using monomial bases can explode combinatorially with input dimension and degree. The paper partly sidesteps this (affine heads in deep/high-dim settings), but then the identifiability and symbolic clarity are less compelling.
>
> A: Thank you for this important suggestion! We have **added a dedicated discussion (in main text)** of these design choices in Section 2.2 (Interpretability) of the revised manuscript. Regarding the **use of polynomial bases**, we acknowledge that, in deep variants, high-degree polynomial bases can lead to unstable optimization and training. Replacing polynomial maps with affine transformations preserves the underlying optimization semantics but reduces symbolic granularity, yielding a more qualitative rather than fully symbolic interpretation of each block.
>
> Regarding the **identifiability of BL(Deep) with affine transformations**, we note that using affine maps as basis functions has only a very minor effect on identifiability because BL employs a diverse set of activation functions within each B block. To support this point, we **added a classical identifiability test** based on the Jacobian rank criterion (Appendix H.8 and Figure 13). We are pleased to report that, within the range we have tested, **BL models with at least 7 layers (using affine bases) exhibit 100% identifiability**, whereas ReLU-based MLPs fail to satisfy the identifiability criterion even at a single layer.
>
> [continued in "Response to Reviewer sGAR part 4"]

---

> ### Author Response · Authors · 2025-11-18
> **Response to Reviewer sGAR part 4**
>
> In addition, the specific effects of **skip-connections** on interpretability are **discussed in our response** to ''Q1.2: Skip-connections of BL(Deep)''.
>
> > Q3: Constraint satisfaction under noise. At finite temperature, Gibbs may does not enforce constraints exactly.
>
> A: Thank you for this important suggestion! Following your advice, we have **added a new diagnostic experiment** (Appendix H.6; Fig. 11) that directly evaluates how BL's penalty blocks enforce near-hard constraints under finite temperature. The experiment tests a 64-dimensional energy-conservation constraint using a physics-style residual, as suggested. We provide feasibility-violation curves with respect to both τ and the penalty weight λ, and we discuss numerical stability.
>
> In summary, λ and τ only need to be **sufficiently finite to achieve strong constraint** satisfaction: for example, for 64-dimensional Gaussian noise with unit standard deviation, λ = 25 and τ = 0.01 already enforce the 64-dimensional energy-conservation constraint to within a 1e-2 error.
>
> > Q4: CausalBL.
>
> A: Thank you for raising this important point. We fully agree that the CausalBL component is substantial and could naturally grow into a standalone paper. In the current submission, our intention was only to use the causal extension of IBL to illustrate how the identifiability of IBL translates into gains in counterfactual prediction and causal task, thereby supporting the theoretical claims of the main paper rather than introducing a full causal framework.
>
> Following your suggestion, we have **revised the main text** to streamline the discussion of CausalBL, presented only as a basic IBL-based causal extension, so that readers can understand its role without being distracted by additional methodological details. The full technical development and empirical results remain in the appendix, where they do not affect the scope or focus of the main paper. A more complete and general treatment of CausalBL is planned as a separate follow-up work. We also plan to further revise and streamline this section in the camera-ready version to ensure clarity and maintain a tight focus on the core contributions of the paper.
>
> In addition, we would be very happy to answer any questions you may have about CausalBL.
>
> > Q5: Calibration.
>
> A: Thank you for this important suggestion! Following your advice, we have **added ECE and NLL results** for the high-dimensional prediction tasks **in Appendix H.5**. **Overall**, BL exhibits **better ECE and NLL than** the energy-based MLP, and the improvement is especially pronounced on the two image datasets, where BL's calibration is substantially better than that of the energy-based MLP.
>
> > Q6: Hybrid approaches for partial explainability.
>
> A: Thank you for this highly insightful suggestion! We fully agree that hybrid architectures enabling partial explainability represent a promising and important direction. Following your advice, we have **added a dedicated discussion** of this idea **in the Future Work section (Appendix C)**. In particular, we highlight three concrete avenues that we believe are especially promising:
>
> (i) **Feature-level integration**. Black-box neural networks can serve as high-capacity feature extractors, while BL operates on the resulting learned representations to impose structured, optimization-based semantics.
>
> (ii) **Decision-critical integration**. BL blocks can be placed specifically at high-risk or decision-critical parts of the model, substantially reducing the interpretability and reliability risks inherent in purely black-box architectures.
>
> (iii) **Mechanism-level integration**. Because BL provides an optimization-driven inductive bias aligned with many real-world mechanisms, selectively applying BL to components where such inductive bias is essential may yield models that more faithfully capture the underlying ground-truth processes while retaining the flexibility of deep neural networks, thereby improving generalization performance.
>
> We sincerely appreciate the reviewer for pointing out this valuable research direction!
>
> Once again, we sincerely appreciate your insightful suggestions and constructive feedback! Your review has significantly strengthened the quality and clarity of this work, and we are truly grateful for your contribution!
>
> We hope that these responses fully address your questions and reinforce your confidence in Behavior Learning.

---

> ### Author Response · Authors · 2025-11-27
> **Looking forward to your reply**
>
> Dear Reviewer sGAR,
>
> This message is from the authors of **"Behavior Learning"** (Submission 5851).
>
> As the discussion phase will **conclude in a few days**, we kindly ask whether there are any remaining concerns we may address, and whether our responses have strengthened your evaluation of the work. We would be happy to provide any further details if needed.
>
> Thank you again for your time and thoughtful review.

---

> > ### Comment · Reviewer_sGAR · 2025-11-27
> > **I thank the authors for their productive engagement with the reviewing process**
> >
> > The authors have exhaustively answered all my questions and addressed my concerns, with the exception of aesthetic points (i.e. I still contend the shallow/deep distinction is not useful). The paper is high quality, potentially useful, and I think it is unambiguous should be published so that it can be evaluated in practice by the field at large.

---

> ### Author Response · Authors · 2025-11-28
> **Thank You for Your Insightful and Tasteful Review**
>
> Dear Reviewer sGAR,
>
> We are truly glad to hear that our responses have addressed your concerns. Your insightful and tasteful suggestions have significantly contributed to improving the quality and clarity of the work, and we are sincerely grateful for the time and expertise you invested in the review.
>
> Your advice is one of the most valuable takeaways for us in this conference cycle.
>
>
> Best regards,
>
> The Authors of "Behavior Learning"

---

### Official Review · Reviewer_jMN4 · 2025-11-12

**Soundness:** 3
**Presentation:** 4
**Contribution:** 2
**Rating:** 4
**Confidence:** 3

**Summary:**

The paper proposes to learn data distribution P(x | y) by using a specific form of parametrized probability families, motivated from the Lagrangian reformulation of constraint programs. The paper moves on to develop relevant theories for this parametrized probability class, including approximation results, uniqueness of optima of the loss constructed from this class, and certain estimation results including consistency.

**Strengths:**

1. The paper is very well written, and compares with many benchmarks. In appendix there are abundant introduction of the dataset used and the competing methods.
2. The proposed parametrization has utility maximization interpretation. Learning the model is uncovering agents' constraint, utility function at the same time.

**Weaknesses:**

1. The statistical results part is standard. A more interesting theory would be to show, the parametrization is effecient if the data distribution is indeed generated from some utility maximization process.

**Questions:**

1. In Eq 4, what are the exact forms of phi, rho, and psi? Are the the same as those in Theorem 2.1? Are they hyper-parameters to be chosen? If so how to choose them?
2. In section 3.2, do the authors consider endogeneity in the dataset, or do they assume exogeneity? If latter, then the problem is just simple regression problem.
3. In all the numerical comparison, are the number of parameters similar? Using similarly complex models ensures a fair comparison.
4. In the numerical part, does BL recover the relevant utility and constraint functions?

---

> ### Author Response · Authors · 2025-11-18
> **Clarifications on This Work**
>
> We thank the reviewer for the comments.
>
> To avoid misunderstandings and before addressing point-by-point responses, we first provide clarifications on the key ideas of the BL framework.
>
> ### Clarifications on This Work
>
> **(1) Architecture.**
>
> **Behavior Learning (BL) is a novel general-purpose machine learning framework designed to discover interpretable and identifiable optimization structures from data**.
>
> Beyond "P(x | y) by using a specific form", Behavior Learning encompasses a substantial amount of important modeling detail, including but not limited to interpretability [Lipton,2018; Molnar,2020], energy-based formulation [LeCun et al., 2006], conditional Gibbs distribution[Gibbs,1902], penalty-based construction[Han & Mangasarian, 1979], compositional modeling[Kolmogorov,1961], score-based training[Hyvärinen & Dayan, 2005; Vincent,2011], langevin-style inference[Langevin,1908;Welling & Teh, 2011], inverse optimization[Ahuja & Orlin, 2001].
>
> As Reviewer sGAR noted, BL is more accurately viewed as a **novel sub-family of energy-based models** [LeCun et al., 2006]. Alternatively, as Reviewers 3vXd and za71 noted, BL can also be viewed as a **novel or quite interesting interpretable machine learning framework** [Molnar,2020].
>
> **(2) Motivation.**
>
> The motivation for BL comes from:
> (i) **Behavioral Science**: utility-maximization principles and hierarchical optimization systems in social science; and
> (ii) the need to address the **performance-interpretability trade-off, insufficient alignment with scientific theory and non-identifiability** of many existing machine learning models.
>
> BL is not motivated by Lagrangian reformulation, and the paper does not use, reference, or mention any "Lagrangian reformulation".
>
> **(3) Contribution.**
>
> BL unifies **predictive performance, intrinsic interpretability, and identifiability for scientifically credible modeling**, supported by both theoretical analysis and empirical validation. The theoretical results are important for ensuring the scientific credibility of BL.
>
> However, the paper is not positioned as a purely theoretical contribution, nor do we claim that the main contribution lies in the theoretical results.
>
> **(4) Identifiability.**
>
> **Identifiability in scalable deep models is rare, yet BL possesses identifiability** due to its special optimization-grounded architecture. To further clarify this point, we added an identifiability diagnostic experiment based on the classical Jacobian-rank criterion (Appendix H.8 and Figure 13). The results show that  BL models with up to at least 7 layers exhibit 100% identifiability, whereas standard MLPs of comparable size fail to satisfy the identifiability criterion even at a single layer.
>
> LeCun, Y., Chopra, S., Hadsell, R., Ranzato, M., & Huang, F. (2006). A tutorial on energy-based learning.
>
> Lipton, Z. C. (2018). The mythos of model interpretability. Queue, 16(3), 31-57.
>
> Molnar, C. (2020). Interpretable Machine Learning.
>
> Gibbs, J. W. (1902). Elementary principles in statistical mechanics. C. Scribner's sons.
>
> Han, S.-P., & Mangasarian, O. L. (1979). Exact penalty functions in nonlinear programming. Mathematical Programming, 17(1), 251-269.
>
> Kolmogorov, A. N. (1961). On the representation of continuous functions of several variables by superpositions of continuous functions of a smaller number of variables. American Mathematical Society.
>
> Hyvärinen, A., & Dayan, P. (2005). Estimation of non-normalized statistical models by score matching. JMLR, 6.
>
> Vincent, P. (2011). A connection between score matching and denoising autoencoders. Neural Computation, 23(7), 1661-1674.
>
> Langevin, P. (1908). On the theory of brownian motion. CR Acad Sci (Paris), 146, 530.
>
> Welling, M., & Teh, Y. W. (2011). Bayesian learning via stochastic gradient Langevin dynamics. ICML.
>
> Ahuja, R. K., & Orlin, J. B. (2001). Inverse optimization. Operations Research, 49(5), 771-783.
>
> [continued in "Response to Reviewer jMN4 part 1"]

---

> ### Author Response · Authors · 2025-11-18
> **Response to Reviewer jMN4 part 1**
>
> ### Formal Rebuttal
>
> Below, we provide **point-by-point responses** to the reviewer's comments.
>
> > W1.1 A more interesting theory would be to show, the parametrization is efficient if the data distribution is indeed generated from some utility maximization process.
>
> A: The efficiency result **is proved in Appendix F, Theorem F.10** and **was proved in the initial submission** (Appendix D, Theorem D.10, lines 1771-1794). In particular, we show that the estimator attains the **Cramér--Rao lower bound** under the cross-entropy loss and the **efficient information bound** under the denoising score matching loss. Therefore, the **BL estimator is asymptotically efficient**.
>
> > W1.2 The statistical results part is standard.
>
> A: While some proof techniques (e.g., for consistency) follow standard arguments, **to the best of our knowledge, there is almost no prior work** provides **a unified proof of the full set of M-estimation properties**--identifiability, (uniform) consistency, asymptotic normality, and asymptotic efficiency--**for scalable deep hierarchical models**. Existing results usually address only subsets of these properties (e.g., sieve-based consistency [Shen et al., 2023] or identifiability under strong structural constraints [Bona-Pellissier et al., 2023]) in special settings.
>
> Existing deep layered architectures (especially neural networks) typically lack formal identifiability, making their parameter M-estimation properties mathematically ill-defined. Our framework provides a rigorous unified proof that establishes these properties and offers a theoretical foundation for our model and future research on deep hierarchical modeling. To further clarify this point, we **added an identifiability diagnostic experiment** based on the classical Jacobian-rank criterion (Appendix H.8 and Figure 13). The results show that IBL models with up to at least 7 layers exhibit 100% identifiability, whereas standard MLPs of comparable size fail to satisfy the identifiability criterion even at a single layer.
>
> If there exist other works providing comparably formal and complete statistical guarantees for deep layered models, we would be glad to include a detailed comparison in the paper.
>
> Shen, X., Jiang, C., Sakhanenko, L., & Lu, Q. (2023). Asymptotic properties of neural network sieve estimators. Journal of nonparametric statistics.
>
> Bona-Pellissier, J., Bachoc, F., & Malgouyres, F. (2023). Parameter identifiability of a deep feedforward ReLU neural network. Machine Learning.
>
> > Q1 In Eq 4, what are the exact forms of phi, rho, and psi? Are the the same as those in Theorem 2.1? Are they hyper-parameters to be chosen? If so how to choose them?
>
> A: This setting is **specified in in Section 2, pp. 4-5** and **was included in the initial submission** (Section 2, pp. 3-4). ϕ, ρ, and ψ are our penalty functions (serving a role **analogous to activation functions**, and thus they **are not hyper-parameters**) and are grounded in Theorem 2.1. We only require them to satisfy the conditions stated in Theorem 2.1, and any function meeting these conditions can serve as a valid penalty function in our model. Considering behavioral-science priors such as diminishing marginal utility, we used tanh, ReLU, and |·| (absolute value) for ϕ, ρ, and ψ respectively in the BL implementation, and tanh, softplus, and element-wise square for the IBL implementation.
>
> > Q2 In section 3.2, do the authors consider endogeneity in the dataset, or do they assume exogeneity? If latter, then the problem is just simple regression problem.
>
> A: This setting is **specified in Appendix G** and **was included in the initial submission** (Appendix E, pp. 35-37). **Our framework is fundamentally different from the standard regression and does not rely on exogeneity assumptions like regression**. In general, a regression problem is defined as $y = f(x)$, representing a one-way mapping $x \rightarrow y$. Mathematically, for input $x$ and label $y$, our model instead takes $(x, y)$ jointly as input and produces a compositional utility $BL(x, y)$ (which can also be viewed as a score), i.e. $(x, y) \rightarrow BL(x, y)$, **which is fundamentally different from the standard regression formulation** $y = f(x)$. Through the denoising score matching loss, the model learns to assign higher compositional utility to the observed $(x, y)$ pairs. This endogenous dependence is structurally captured by the compositional utility function $BL(x, y)$, and its identifiability, consistency, and other M-estimation properties are formally guaranteed by our theoretical framework--without relying on the exogeneity assumption required by classical regression models. Therefore, our formulation is fundamentally different from a simple regression problem and it aims to recover the underlying optimization problems that generate the observed data.
>
> For more detailed explanations, see the literature on energy-based models [LeCun et al., 2006] or generative models.

---

> ### Author Response · Authors · 2025-11-18
> **Response to Reviewer jMN4 part 2**
>
> > Q3 In all the numerical comparison, are the number of parameters similar? Using similarly complex models ensures a fair comparison.
>
> A: In the Prediction on High-Dimensional Inputs experiments, BL and the energy-based MLP have nearly **identical parameter counts**. In terms of runtime, BL runs substantially faster than the energy-based MLP on text datasets, while being only moderately slower on image datasets (see **Appendix H.5 for details**). In the standard prediction tasks, all methods share a unified preprocessing and tuning pipeline. Some model families do not have a well-defined parameter count. To ensure perfect fairness beyond only parameter-count comparability, we used a **unified Optuna-based tuning procedure and allocated the same tuning budget** to every model, guaranteeing fully fair and comparable optimization across methods.
>
> > Q4 In the numerical part, does BL recover the relevant utility and constraint functions?
>
> A: We thank the reviewer for raising this question. The experiments in our initial submission are primarily conducted on real-world datasets where the ground-truth utility and constraint functions are not observable. To address this concern, following your advice, we have **added parameter recovery experiments** in Appendix H.7. The results show that BL(Single) **accurately recovers** the true parameters (with only an error of 1e-2), and BL(Deep) also demonstrates **strong recovery performance**.
>
>
> We hope that the clarifications and detailed answers provided here have addressed the your concerns.

---

### Author Response · Authors · 2025-11-18
**Global Response**

We sincerely thank the area chairs and reviewers for their careful evaluation and constructive feedback.  We are pleased to receive positive and high-quality reviews, and we greatly appreciate the reviewers' recognition of our contributions.

In particular, we are encouraged by comments noting that the paper is **Definitely a first step in an interesting direction** (Reviewer sGAR), **novel** (Reviewer sGAR, 3vXd), **quite interesting** (Reviewer za71), a **good contribution** (Reviewer sGAR, 3vXd, za71), demonstrates **excellent or good soundness** (Reviewer sGAR, 3vXd, za71, jMN4), and exhibits **excellent or good presentation** quality (Reviewer jMN4, sGAR, za71).

We have carefully reflected on and addressed every point raised by the reviewers in detail. All major modifications are **highlighted in blue** in the revised manuscript.

Below, we provide a summary of central concerns and our revisions.

### Summary of Central Concerns and Our Revisions

**The Interpretability of BL(Deep)** is the central concern raised by most reviewers (Reviewer sGAR, Reviewer 3vXd, Reviewer za71), **The Downward Shift of the Pareto frontier** is another important question (Reviewer sGAR), and reviewers (Reviewer sGAR, Reviewer 3vXd, Reviewer za71) also pointed out **issues related to writing**. We have made some important revisions specifically targeting these issues:

**1. How to interpret BL(Deep)? - Hierarchical Optimization System**

We provide detailed responses to each comment individually, and the overall revisions can be summarized in two main points:

**Scientific Explanation Procedure of BL(Deep)**. We have **added a detailed Scientific Explanation Procedure** of BL(Deep) in Appendix B.
BL(Deep) is interpreted in a bottom-up fashion as a hierarchical optimization system, where each block represents a sub-optimization problem and the relationship between any two consecutive layers can be viewed either as an aggregation step or as a coarse-grained observation. Overall, the interpretive pathway follows: raw input features -> micro-level optimization blocks -> macro-level aggregation or coarse-grained behavioral constructs -> macro-level optimization system. In addition, BL is flexible, and certain design choices affect how interpretations are expressed. We **added a discussion** in Section 2.2 (Interpretability), including how skip connections and the choice between polynomial and affine mappings influence the form of interpretability.

**Two new interpretability case studies of BL(Deep)** on the Boston Housing dataset have **been added** (Figure 2 and Section 3.3). The revision now includes analyses for both BL[2,1] and BL(Deep) [5,3,1], illustrating how lower-layer preference types are aggregated into higher-level trade-off mechanisms and ultimately into single representative buyer.

* For BL[2,1], Layer 1 identifies two micro-level preference types (the Economic-sensitive Buyer and the Location-sensitive Buyer) and Layer 2 aggregates them into an effective representative buyer.

* For BL(Deep) [5,3,1], Layer 1 recovers five distinct micro-level housing preference types; Layer 2 identifies three macro-level trade-off mechanisms describing how these primitive preferences in Layer 1 interact; and Layer 3 aggregates them into the overall representative buyer. This provides a hierarchical explanation aligned with aggregation-and-coordination principles in behavioral science and with the coarse-graining principle in statistical physics, thereby reconstructing a micro-to-macro optimization hierarchy.

Moreover, the preference patterns and trade-off mechanisms uncovered by BL(Deep) correspond closely to classical findings in the economics literature, demonstrating that BL provides meaningful and scientifically grounded interpretability.

**2. Downward Shift of the Pareto Frontier.**

BL indeed shifts the Pareto frontier downward. We **added detailed results** from the high-dimensional prediction experiments in Appendix H.5, **including parameter counts, runtime, and calibration**. BL and the energy-based MLP have nearly identical parameter sizes; BL runs substantially faster on text datasets and only moderately slower on image datasets. These results support the claim that BL improves interpretability without sacrificing accuracy and efficiency.

**3. Writing Adjustments**

In terms of the structure, we moved the key IBL identifiability assumption into the main text, expanded the related work in the Introduction, and added a concise Discussion section summarizing contributions, limitations, and future directions. In terms of the causal extension, Appendix G was fully revised to align CausalBL with classical causal generative modeling, clarify posterior factors versus propensity scores, make ITE/ATE estimands explicit, and explain discrete versus continuous outcome factorization. The writing-related discussion of BL(Deep) is addressed in Point 1.

Below, we also provide comment-by-comment responses for each reviewer.

---

> ### Author Response · Authors · 2025-11-29
> **Summary of Responses part 1**
>
> ### Reviewer-by-Reviewer Summary of Responses
>
> Below we provide a reviewer-by-reviewer overview of our responses, summarizing how each concern has been addressed in the revised manuscript.
>
> ---
>
> **Reviewer sGAR** provided a **very positive** review and recommended **acceptance**, noting our work as "novel," "cool insight" and "definitely a first step in an interesting direction."
>
> They raised the following points, all of which we have **now fully addressed**.
>
> * **How does Deep BL work in practice?** We added two interpretability case studies and a concise scientific explanation in revised paper that clarify this process. BL(Deep) builds a hierarchical optimization system by stacking B-blocks. The first layer takes raw features; higher layers take the utilities from the previous one. Interpretation follows a bottom-up pathway.
>
> * **Architectural factors affecting interpretability**. We added discussions (in main text) addressing each architectural factor raised. Skip connections create meaningful cross-layer dependencies, and using affine (rather than polynomial) mappings yields more qualitative interpretations. Affine mapping choices have minor impact on identifiability, as confirmed by the newly added Jacobian-rank tests.
>
> * **Whether there is a downward shift of the Pareto frontier**. We added detailed results in revised paper showing that BL indeed shifts the frontier downward; see the Global Response for details.
>
> * **Constraint enforcement under finite-temperature Gibbs distributions.** BL enforces constraints effectively with finite λ or τ. Our new diagnostic experiment shows that λ=25 and τ=0.01 already enforce the 64-dimensional energy-conservation constraint to within a 1e-2 error.
>
> * **CausalBL writing, calibration results, and hybrid approaches.** We revised the main text to improve the CausalBL exposition, added ECE and NLL calibration results, and included a discussion of hybrid approaches and related future work.
>
> ---
>
> > **Reviewer sGAR** replied that **"the paper is high quality, potentially useful, and should unambiguously be published."**
>
> ---
>
> **Reviewer 3vXd** provided a **positive** review and recommended **acceptance**, noting our work as "novel," "thorough and solid theoretical analysis" and "comprehensive experiments."
>
> They raised the following points, all of which we have **now fully addressed**.
>
> * **What is the semantic interpretability of BL.** We explained how BL maintains semantic interpretability: outer activations encode behavioral priors, while inner polynomial modules retain symbolic interpretability. While depth affects readability, it does not affect BL's mechanistic interpretability: each block retains clear and transparent semantics.
>
> * **Insufficient interpretability case studies.** We added two new real-world case studies (BL[2,1] and BL[5,3,1]) in Section 3.3 and Figure 2; see the Global Response for details.
>
> * **Some Imprecise causal definitions and terminology in CausalBL.** We thoroughly revised Appendix G to align CausalBL with classical generative causal modeling; see the Global Response for details.
>
> * **Whether there exists a single-UMP Interpretation of BL.** Yes, BL can be interpreted as a single UMP when the final layer contains only one B-block, since all lower-layer structures aggregate into a unified optimization problem. When the final layer contains multiple B-blocks, BL corresponds to a linear trade-off among multiple optimization problems. We have added this discussion to Section 2.2 (Interpretability).
>
> ---
>
> [continued in " Summary of Responses part 2 "]

---

> ### Author Response · Authors · 2025-11-29
> **Summary of Responses part 2**
>
> ---
>
> **Reviewer za71** provided a **positive** review and recommended **acceptance**, noting our work as "quite interesting," "well motivated" and ''good soundness''.
>
> They raised the following points, all of which we have **now fully addressed**.
>
> * **Paper structure and positioning.** We made the corresponding writing adjustments; see the Global Response for details.
>
> * **Insufficient interpretability case studies for BL(Deep).** We added two new real-world case studies of BL(Deep) (BL[2,1] and BL[5,3,1]) in Section 3.3 and Figure 2; see the Global Response for details.
>
> * **What is the scientific explanation procedure for BL(Deep).** We added a clear, step-by-step scientific explanation workflow; see the Global Response for details.
>
> * **Extension beyond polynomial bases.** We explained that BL is not restricted to polynomial bases and can easily incorporate a wide range of alternative basis families. A discussion of this flexibility has been added to the revised manuscript in future work part (Appendix C).
>
> ---
>
> **Reviewer jMN4** provided a **generally positive** review, describing the paper as "very well written" and "good soundness."
>
> At the same time, there were initial misunderstandings, and the initial overall rating remained **borderline**. To address this, we first provide a comprehensive clarification of the architecture, motivation, contributions, and related work on identifiability. We then offer point-by-point responses to all comments.
>
> All initial **misunderstandings** have been fully clarified now. All initial **concerns** have been **fully addressed now**, as summarized below:
>
> * **Statistical efficiency result.** We clarified that the efficiency result is proved in Appendix F, Theorem F.10 and was proved in the initial submission and provided an accompanying explanation. Overall, the BL estimator is asymptotically efficient.
>
> * **What are phi, rho, and psi.** We explained that these functions are fixed penalty/activation functions rather than hyperparameters, and provided concrete explanations accordingly.
>
> * **Whether regression-style exogeneity is assumed.** We clarified that BL is fundamentally different from regression models and does not rely on exogeneity like classical regression.
>
> * **Whether the numbers of parameters are comparable.** We explained that all experimental comparisons involving BL are fully fair, with BL and the MLP using identical numbers of parameters in the high-dimensional experiments, and we provided supporting evidence.
>
> * **Whether there is recovery of utility and constraint functions.** We added parameter-recovery experiments, showing that BL(Single) recovers ground-truth parameters with high accuracy (within 1e-2 error), and BL(Deep) also demonstrates strong recovery performance.
>
> ---
>
> Overall, we thank the area chairs and reviewers for their thoughtful and expert advice that has made us improve the paper much beyond the original submission.

---

### Meta-Review · Area_Chair_N8VC · 2026-01-07

**Summary:**

This paper introduces Behavior Learning (BL) and Identifiable BL (IBL), a new interpretable energy-based modeling framework grounded in utility-maximization structure. Reviewers agree the approach is novel, theoretically well-developed (including identifiability and full M-estimation results), and empirically competitive. Overall, the paper is above threshold of acceptance.

**Reviewer Concerns:**

**Addressed**

- Pareto frontier analysis and constraint enforcement
- Improved clarity, terminology, and structure
- More case studies provided

**Still outstanding**
- Deep interpretability, while much better demonstrated, still requires careful human inspection. The applicability of the proposed method in practice might need more exploration.

**Reviewer Scores:**

I think the authors have comprehensively responded to the reviewers in their rebuttal, which addresses most of the concerns.

Reviewer sGAR who gave 8 originally actively got engaged in the discussion. I believe they might have remained strongly postive.

The author provided detailed rebuttal to Reviewer 3vXd and za71 who gave postive ratings originally. I think they might have improved or remained postive.

Reviewer jMN4 gave 4 originally but I didn't see a very strong point in the weaknesses and it's mainly about clarity, which the authors tried to address.

---

### Decision · Program_Chairs · 2026-01-26

Accept (Poster)